# Self-assembly of pericentriolar material in interphase cells lacking centrioles

Fangrui Chen[1], Jingchao Wu[1], Malina K Iwanski[1], Daphne Jurriens[1], Arianna Sandron[1], Milena Pasolli[1], Gianmarco Puma[1], Jannes Z Kromhout[1], Chao Yang[1], Wilco Nijenhuis[1,2], Lukas C Kapitein[1,2], Florian Berger[1], Anna Akhmanova[1]*

[1]Cell Biology, Neurobiology and Biophysics, Department of Biology, Faculty of Science, Utrecht University, Utrecht, Netherlands; [2]Center for Living Technologies, Eindhoven-Wageningen-Utrecht Alliance, Utrecht, Netherlands

**Abstract** The major microtubule-organizing center (MTOC) in animal cells, the centrosome, comprises a pair of centrioles surrounded by pericentriolar material (PCM), which nucleates and anchors microtubules. Centrosome assembly depends on PCM binding to centrioles, PCM self-association and dynein-mediated PCM transport, but the self-assembly properties of PCM components in interphase cells are poorly understood. Here, we used experiments and modeling to study centriole-independent features of interphase PCM assembly. We showed that when centrioles are lost due to PLK4 depletion or inhibition, dynein-based transport and self-clustering of PCM proteins are sufficient to form a single compact MTOC, which generates a dense radial microtubule array. Interphase self-assembly of PCM components depends on γ-tubulin, pericentrin, CDK5RAP2 and ninein, but not NEDD1, CEP152, or CEP192. Formation of a compact acentriolar MTOC is inhibited by AKAP450-dependent PCM recruitment to the Golgi or by randomly organized CAMSAP2-stabilized microtubules, which keep PCM mobile and prevent its coalescence. Linking of CAMSAP2 to a minus-end-directed motor leads to the formation of an MTOC, but MTOC compaction requires cooperation with pericentrin-containing self-clustering PCM. Our data reveal that interphase PCM contains a set of components that can self-assemble into a compact structure and organize microtubules, but PCM self-organization is sensitive to motor- and microtubule-based rearrangement.

## Editor's evaluation

Microtubules are organized by microtubule organizing centers (MTOCs) such as the centrosome, which is composed of two centrioles surrounded by pericentriolar material (PCM). Despite a century of investigation, the mechanisms by which the centrosome organizes microtubules remains incompletely understood. Here, using genetic and pharmacological manipulations, as well as computer simulations, Chen and colleagues generate interphase cells with centriole-less PCM to investigate mechanisms by which PCM proteins cluster and nucleate and anchor microtubules. This manuscript will be of interest to cell biologists studying microtubule organization.

## Introduction

The centrosome is the major microtubule organizing center (MTOC) in animal cells. It consists of two centrioles surrounded by pericentriolar material (PCM) (reviewed in *Conduit et al., 2015*; *Paz and Luders, 2017*). Major PCM components are microtubule-nucleating and anchoring proteins, which can associate with centrioles and with each other. For a long time it was thought that the PCM is amorphous, but super-resolution microscopy studies have shown that it has a distinct organization, with

*For correspondence: a.akhmanova@uu.nl

some proteins likely attached to the centriole wall and others organized around them (*Fu and Glover, 2012*; *Lawo et al., 2012*; *Mennella et al., 2014*; *Mennella et al., 2012*). This distinct organization is more obvious in interphase than in mitosis, when the microtubule-organizing capacity of the centrosome increases due to enhanced PCM recruitment. Many PCM components are known to oligomerize and interact with each other, and recent work suggested that the phase separation of interacting PCM components might contribute to centrosome assembly during mitosis (*Raff, 2019*; *Woodruff et al., 2017*). This idea is underscored by data showing that various cell-type-specific assemblies of PCM components can form clusters that are able to nucleate and organize microtubules and serve as MTOCs in the absence of centrioles, particularly during the formation of mitotic spindle poles (*Balestra et al., 2021*; *Chinen et al., 2021*; *Gartenmann et al., 2020*; *Meitinger et al., 2020*; *Watanabe et al., 2020*; *Yeow et al., 2020*). Furthermore, an important centrosome component, cytoplasmic dynein, is a motor that can bind to different PCM proteins and transport them to the centrosome-anchored microtubule ends, where these PCM proteins can nucleate and anchor additional microtubules, thus generating a positive feedback loop in centrosome assembly (*Balczon et al., 1999*; *Burakov et al., 2008*; *Purohit et al., 1999*; *Redwine et al., 2017*). Dynein and its mitotic binding partner NuMA also strongly participate in the formation of mitotic and meiotic spindle poles (*Chinen et al., 2020*; *Khodjakov et al., 2000*; *Kolano et al., 2012*). The relative importance of different molecular pathways of PCM assembly varies between cell systems and phases of the cell cycle and has not been explored systematically in interphase cells.

Here, we set out to investigate the centriole-independent self-assembly properties of interphase PCM. These properties, such as the ability of PCM to cluster or form molecular condensates, nucleate and anchor microtubules and move with motor proteins, are relevant because in most differentiated cell types, centrosome function is suppressed, and some PCM components form acentrosomal MTOCs (*Muroyama and Lechler, 2017*; *Sallee and Feldman, 2021*). During mitotic exit, when mitotic kinases are inactivated, PCM complexes can be removed from the centrosomes as 'fragments' or 'packets' (*Magescas et al., 2019*; *Rusan and Wadsworth, 2005*), indicating that they maintain some degree of self-association. In other cases, complexes of PCM proteins may fully disassemble and then assemble at other locations, but their properties will likely still determine the composition and localization of acentrosomal MTOCs. To study centriole-independent function and dynamics of PCM proteins, we removed centrioles using the PLK4 kinase inhibitor centrinone B which blocks centriole duplication (*Wong et al., 2015*). Importantly, in most commonly studied cultured cell lines, such as fibroblasts, epithelial, endothelial or cancer cells, microtubule networks are dense, and the centrosome is not the only MTOC. In such cells, non-centrosomal microtubule minus ends are often stabilized by the members of CAMSAP family (*Jiang et al., 2014*; *Meng et al., 2008*; *Tanaka et al., 2012*), and the Golgi apparatus serves as a second MTOC which nucleates and anchors a very significant proportion of microtubules (*Efimov et al., 2007*; *Rios, 2014*; *Wu et al., 2016*; *Zhu and Kaverina, 2013*). If centrosomes are lost because centriole duplication is blocked by inhibiting PLK4 or depleting another essential centriole component, Golgi-dependent microtubule organization becomes predominant (*Gavilan et al., 2018*; *Martin et al., 2018*; *Wu et al., 2016*). The ability of the Golgi complex to serve as an MTOC critically depends on the Golgi adaptor AKAP450, which recruits several PCM components that nucleate microtubules including the γ-tubulin ring complex (γ-TuRC), CDK5RAP2 and pericentrin (*Gavilan et al., 2018*; *Rivero et al., 2009*; *Wu et al., 2016*). Moreover, AKAP450 also tethers microtubule minus ends stabilized by CAMSAP2 to the Golgi membranes (*Gavilan et al., 2018*; *Rivero et al., 2009*; *Wu et al., 2016*). In the absence of AKAP450, the Golgi ribbon is maintained, but neither PCM components nor CAMSAP-stabilized microtubule minus ends can be attached to the Golgi membranes and they are instead dispersed in cytoplasm, leading to a randomly organized microtubule network (*Gavilan et al., 2018*; *Rivero et al., 2009*; *Wu et al., 2016*). These data seem to suggest that centrioles and/or Golgi membranes are essential to assemble PCM into an MTOC in interphase. However, this notion appears to be inaccurate: in our previous study in RPE1 cells, we observed that a single compact acentriolar MTOC (caMTOC) can still form after centriole loss in AKAP450 knockout cells, if the stabilization of free minus ends in these cells is disabled by knocking out CAMSAP2 (*Wu et al., 2016*).

We used this observation as a starting point to investigate which properties of PCM components allow them to self-assemble in interphase mammalian cells and how the presence of non-centrosomal microtubules affects this process. AKAP450 knockout cells provided a system to study assembly of

PCM proteins in the absence of competition with the Golgi-associated MTOC. caMTOC formation in AKAP450 knockout cells required microtubules and depended on dynein, which brought together small PCM clusters with attached minus ends. Experiments and modeling showed CAMSAP2-mediated minus-end stabilization strongly perturbed PCM coalescence, because in the absence of AKAP450, randomly oriented CAMSAP2-stabilized microtubules supported PCM motility and prevented PCM clustering. In the absence of CAMSAP2, caMTOCs did form, but were often cylindrical rather than spherical in shape and contained a subset of the major centrosome components. γ-tubulin, pericentrin, CDK5RAP2 and ninein were necessary for the formation of caMTOCs, whereas some other major PCM proteins, namely CEP192, CEP152 and NEDD1, were neither enriched in these structures nor required for their formation, indicating that not all PCM components associate with each other in the absence of centrioles and that interphase MTOC function does not strictly require these three proteins. A single caMTOC containing PCM components could also form in the presence of CAMSAP2 when this protein was directly linked to a microtubule minus-end-directed motor. Importantly, in the absence of pericentrin, minus-end-directed transport of CAMSAP2-stabilized minus ends organized these ends into a ring, indicating that self-associating PCM is required for the formation of a caMTOC. This conclusion was supported by modeling. Taken together, our data show that a subset of interphase PCM components can self-assemble and efficiently nucleate and tether microtubules, but PCM clustering is sensitive to microtubule- and motor-dependent rearrangements. These properties of interphase PCM may also be involved in the transition from a centrosomal to a non-centrosomal microtubule network, as typically occurs during cell differentiation.

## Results

### Assembly of microtubule-dependent caMTOCs in AKAP450/CAMSAP2 KO cells lacking PLK4 activity

To study centriole-independent PCM organization and dynamics in interphase cells, we induced centriole loss in RPE1 cells by treating them for 11 days with the PLK4 inhibitor centrinone B (*Wong et al., 2015*; *Figure 1A and B*). In wild type (WT) cells, PCM (detected with antibodies against pericentrin) relocalized to the Golgi apparatus, and the microtubule array reorganized around the Golgi membranes, as described previously (*Gavilan et al., 2018*; *Wu et al., 2016*; *Figure 1C and D*). In AKAP450 knockout cells, centriole loss led to the appearance of strongly dispersed PCM clusters, which could no longer bind to the Golgi, and a highly disorganized microtubule system, consistent with published work (*Gavilan et al., 2018*; *Wu et al., 2016*; *Figure 1D and E*). In contrast, in AKAP450/CAMSAP2 double knockout cells, a single caMTOC with microtubules focused around it was observed (*Figure 1D–F*). Formation of a single caMTOC was also observed in AKAP450 knockout cells transiently depleted of CAMSAP2 by siRNA transfection (*Figure 1—figure supplement 1A, B*). As an alternative approach to block the ability of the Golgi apparatus to recruit PCM, we treated the cells with Brefeldin A, which disrupts the Golgi (*Klausner et al., 1992*). As expected, Brefeldin A by itself had no effect on centrosome integrity but led to the dispersal of the Golgi marker GM130 (*Figure 1—figure supplement 1C-E*). In centrinone-treated wild-type cells, PCM was dispersed after Golgi disruption, and the same was true for the majority of CAMSAP2 knockout cells (*Figure 1—figure supplement 1C-F*). However, in 12% of CAMSAP2 knockout cells treated with both centrinone B and Brefeldin A, we observed PCM compaction (*Figure 1—figure supplement 1D-F*), similar to that seen in AKAP450/CAMSAP2 double knockout cells. The efficiency of caMTOC formation was low likely because dispersed AKAP450 could still recruit PCM to some extent or because the duration of the Brefeldin A treatment (2 hr) was too short to allow for PCM compaction in most cells (longer Brefeldin A treatments were not performed due to potential cell toxicity). The acentriolar PCM can thus form a compact structure if both the Golgi MTOC and CAMSAP2-mediated minus-end stabilization are disabled.

Unlike centrosomes, which always have a spherical shape, caMTOCs in AKAP450/CAMSAP2 knockout cells were cylindrical in ~35% of the cells, whereas in the remaining cells that lacked centrioles based on staining for centrin, MTOCs had a round shape (~38% of the cells); the rest of the centrin-negative cells either had dispersed PCM clusters (~7%) or no detectable PCM clusters (~11%) (*Figure 1—figure supplement 2A, B*). In contrast, ~72% of acentriolar AKAP450 knockout cells had dispersed PCM, while caMTOCs were very rare (*Figure 1E*, *Figure 1—figure supplement 2A, B*).

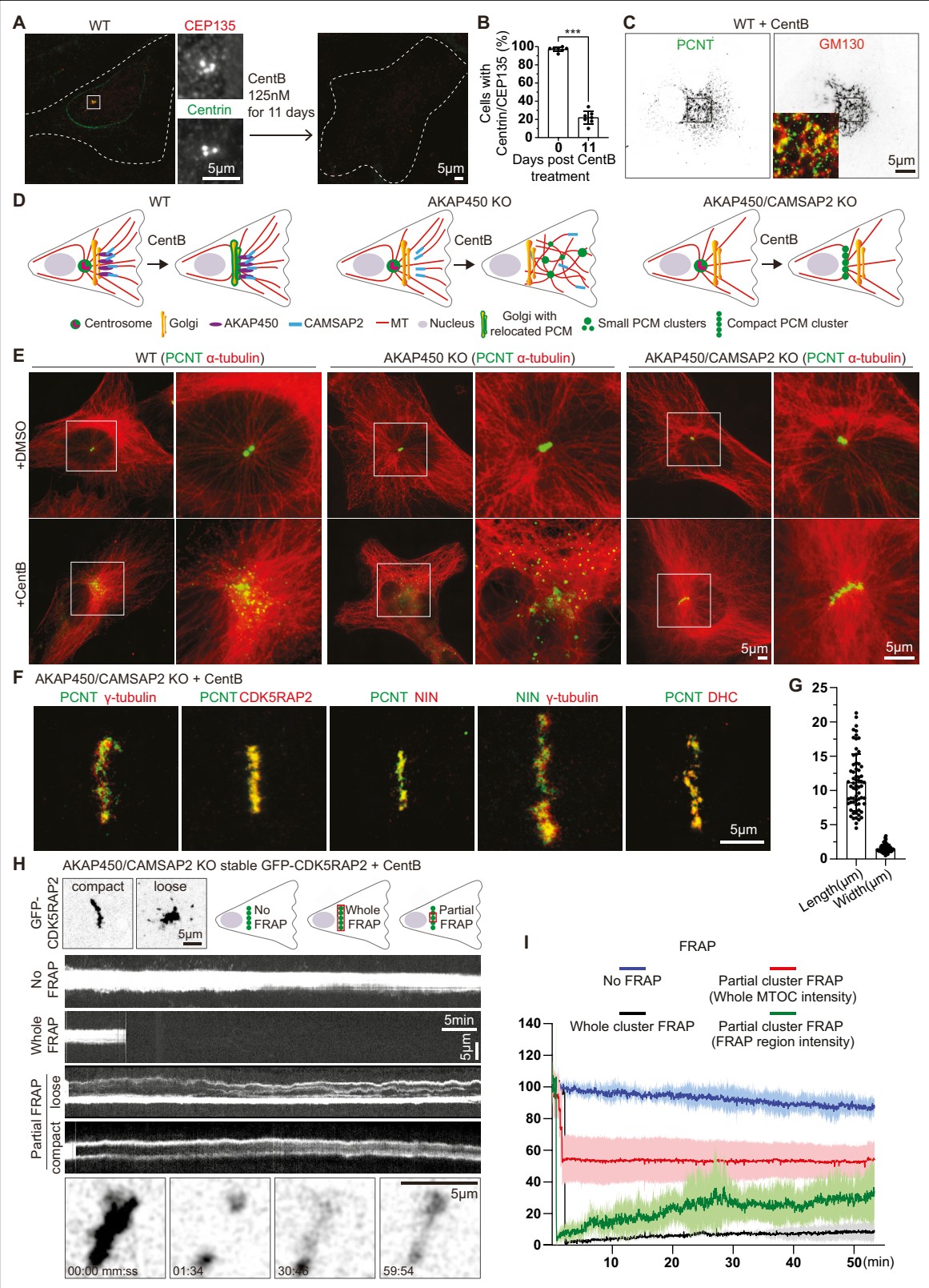

**Figure 1.** Formation and characterization of caMTOCs in AKAP450/CAMSAP2 knockout cells. (**A**) Immunofluorescence images of control or centrinone-treated wild type (WT) RPE1 cells stained for centrioles (CEP135, red; centrin, green). The zooms of the boxed area show the centrioles stained with the indicated markers. (**B**) Quantification shows the percentage of cells with centrioles before and after the centrinone treatment. 350 cells (n=7 fields of view) analyzed for each measurement in three independent experiments. The statistical significance was determined by unpaired two-tailed Mann-

*Figure 1 continued on next page*

*Figure 1 continued*

Whitney test in Prism 9.1 (***p<0.001). Values represent mean ± SD. (C) Immunofluorescence images of centrinone-treated WT RPE1 cells stained for pericentrin (PCNT, green) and the Golgi marker GM130 (red). Inset shows the merged image of the boxed area. (D) Diagrams of the microtubule organization in WT and knockout (KO) cells used. (E) Immunofluorescence images of control and centrinone-treated WT and knockout RPE1 cell lines stained for pericentrin (green) and microtubules (α-tubulin, red). Enlargements on the right show the boxed areas. (F) Immunofluorescence images of centrinone-treated AKAP450/CAMSAP2 knockout RPE1 cells stained for different PCM components as indicated and imaged by STED microscopy. (G) Quantification of the length and width of cylindrical PCM clusters. n=65 cells analyzed in three independent experiments. Values represent mean ± SD. (H) (Top left) Two frames of time-lapse images of centrinone-treated AKAP450/CAMSAP2 knockout RPE1 cells stably expressing GFP-CDK5RAP2 prior to FRAP experiments. (Top right) Schemes show regions of caMTOC where photobleaching was performed. (Middle) Kymographs illustrating fluorescence of unbleached caMTOC (No FRAP), fully photobleached caMTOC (Whole FRAP) and partially photobleached caMTOC (Partial FRAP). (Bottom) Time-lapse images illustrating partial FRAP of a caMTOC. Time is min: s. (I) Normalized fluorescence intensity as a function of time. The blue line shows averaged intensity traces of unbleached caMTOCs (No FRAP), the black line shows averaged intensity traces of fully photobleached caMTOCs (Whole FRAP), the red line shows averaged intensity traces of whole caMTOC that were partially photobleached (whole caMTOC intensity, Partial FRAP) and the green line shows averaged intensity traces of the photobleached region of the partially photobleached caMTOC (FRAP region intensity, Partial FRAP). n=3 for No FRAP, 3 for Whole FRAP, 5 for Partial FRAP (whole caMTOC intensity) and 5 for Partial FRAP (FRAP region intensity); time-lapse images of ~1600 timepoints with 2 s interval were analyzed for each measurement. Values are mean ± SD.

The online version of this article includes the following source data and figure supplement(s) for figure 1:

**Source data 1.** An Excel sheet with numerical data on the quantifications shown in panels B, G, and I.

**Figure supplement 1.** Characterization of caMTOCs in wild type, AKAP450 knockout and CAMSAP2 knockout cells.

**Figure supplement 1—source data 1.** An Excel sheet with numerical data on the quantifications shown in panels B and F.

**Figure supplement 2.** Characterization of caMTOCs in AKAP450/CAMSAP2 knockout cells.

**Figure supplement 2—source data 1.** An Excel sheet with numerical data on the quantifications shown in panels B and D.

**Figure supplement 3.** PCM dynamics visualized with GFP-CDK5RAP2.

Analysis by Stimulated Emission Depletion (STED) microscopy revealed that cylindrical caMTOCs in AKAP450/CAMSAP2 knockout cells consisted of small clusters of PCM components, including pericentrin, CDK5RAP2, γ-tubulin, ninein and dynein heavy chain (*Figure 1F*). caMTOCs with a clearly elongated shape had an average length of ~11 µm and an average width of ~1.5 µm (*Figure 1G*).

To study PCM dynamics in caMTOCs, we generated cell lines stably expressing the PCM component CDK5RAP2 tagged with GFP. In control, untreated cells, GFP-CDK5RAP2 was localized to the centrosome and the Golgi apparatus as expected (*Figure 1—figure supplement 3A*). In centrinone-treated AKAP450/CAMSAP2 knockout cells, it was strongly enriched within caMTOCs and sometimes also present in small motile clusters around a caMTOC (*Figure 1H*, *Figure 1—figure supplement 3B*). Fluorescence recovery after photobleaching (FRAP) assays showed that when the whole caMTOC was bleached, the recovery was very slow and incomplete (*Figure 1H, I*). If only a part of a caMTOC was photobleached, the dynamics of recovery showed cell-to-cell variability. Highly condensed caMTOCs displayed a slow redistribution of GFP-CDK5RAP2 signal, suggesting that their components are largely immobile and do not rearrange. In more loosely organized caMTOCs, some rearrangement of small PCM clusters was observed (*Figure 1H, I*); however, the recovery was still far from complete. These data indicate that caMTOCs display variable degrees of compaction and are composed of PCM clusters that display limited exchange of GFP-CDK5RAP2 with the cytoplasmic pool, possibly because most of the GFP-CDK5RAP2 is accumulated within the caMTOC.

Next, we investigated whether centriole loss induced by means other than pharmacological PLK4 inhibition could also cause the formation of a single caMTOC in AKAP450/CAMSAP2 knockout cells. To achieve efficient protein depletion in RPE1 cells, they were transfected with siRNAs twice (on day 0 and day 2), treated with thymidine starting from day 4 to block cell cycle progression and fixed and stained on day 5 or day 7 (*Figure 1—figure supplement 2C*). Depletion of PLK4 using siRNAs caused the appearance of round or cylindrical caMTOCs, similar to those observed after PLK4 inhibition with centrinone B, indicating that catalytically inactive PLK4 had no scaffolding role within these structures (*Figure 1—figure supplement 2C-F*). The percentage of cells with caMTOCs increased over time (*Figure 1—figure supplement 2D*), possibly due to the gradual depletion of PLK4. In contrast, depletion of CPAP, an essential centriole biogenesis factor (*Kohlmaier et al., 2009*; *Schmidt et al., 2009*; *Tang et al., 2009*), which also led to centriole loss, was much less efficient in inducing caMTOCs, and cylindrical caMTOCs were never observed (*Figure 1—figure supplement 2C-E*). After CPAP depletion, cells in which pericentrin formed dispersed clusters or no visible clusters predominated (~67%,

*Figure 1—figure supplement 2D, E*). Treatment of CPAP-depleted AKAP450/CAMSAP2 knockout cells with centrinone B for 1 day promoted the assembly of round or cylindrical caMTOCs, and the proportion of such cells increased to ~55% after 3 days of centrinone B treatment (*Figure 1—figure supplement 2C-G*). We also tested whether the inhibition of PLK1, a kinase that is known to be a major regulator of PCM self-assembly in mitosis (*Haren et al., 2009*; *Joukov et al., 2014*; *Lee and Rhee, 2011*), has an effect on the formation of caMTOCs by treating cells with BI2536 (a highly selective and potent inhibitor of PLK1), but found this not to be the case (*Figure 1—figure supplement 2C,D*). We conclude that PCM can assemble into a single stable MTOC in a centriole-independent manner if PLK4 is either inactivated or depleted and the two major pathways of microtubule nucleation and minus-end stabilization dependent on the Golgi membranes and CAMSAP2 are disabled.

## Composition of caMTOCs and their effect on microtubule organization

PCM is composed of numerous proteins that can bind to each other and interact with microtubules, and we next set out to investigate which PCM components can self-assemble in the absence of centrioles. We first stained centrinone-treated AKAP450/CAMSAP2 knockout cells with antibodies against different centrosome and centriole markers and microtubule-associated proteins (MAPs). As indicated above, the abundant PCM components pericentrin, CDK5RAP2, ninein and γ-tubulin colocalized within caMTOCs (*Figure 1F*). In contrast, three other major PCM proteins, CEP152, CEP192 and NEDD1, could not be detected in these structures although they were present in centrosomes of AKAP450/CAMSAP2 knockout RPE1 cells that were not treated with centrinone B (*Figure 2A–C*) and were also expressed in centrinone-treated cells (*Figure 2—figure supplement 1A*). We then individually depleted all these proteins in centrinone-treated AKAP450/CAMSAP2 knockout cells using siRNAs. After the depletion of pericentrin, no clusters of other PCM components could be detected (*Figure 2D, J and K*, *Figure 2—figure supplement 1B*). To confirm this result, we also attempted to knock out pericentrin in AKAP450/CAMSAP2 knockout cells, but such cells were not viable, likely because centrosome defects in these cells caused prolonged mitosis and p53-dependent G1 arrest (*Fong et al., 2016*; *Lambrus et al., 2016*; *Meitinger et al., 2016*). However, we were able to knock out pericentrin in cells lacking AKAP450, CAMSAP2 and p53 (*Figure 2—figure supplement 2*), confirming that the loss of p53 makes the cells more tolerant to centrosome absence and allows them to divide even in the presence of centrinone B (*Figure 2—figure supplement 3*, *Video 1*). Similar to pericentrin-depleted cells, these quadruple knockout cells were unable to form a single caMTOC when treated with centrinone B (*Figure 2E and J*). In these acentriolar quadruple knockout cells, CDK5RAP2, γ-tubulin and cytoplasmic dynein displayed no clustering, while ninein and PCM1, a centriolar satellite protein that localizes closely around the centrosome in normal cells (*Prosser and Pelletier, 2020*), formed small clusters distributed throughout the cytoplasm (*Figure 2—figure supplement 4A*). Loss of pericentrin had no effect on the expression of CDK5RAP2, γ-tubulin and ninein (*Figure 2—figure supplement 4B*), indicating that clustering by pericentrin affects the organization but not the stability of these PCM components.

The depletion of CDK5RAP2, γ-tubulin or ninein in centrinone-treated AKAP450/CAMSAP2 knockout cells did not prevent the formation of small pericentrin clusters, but these failed to coalesce into a single caMTOC (*Figure 2F–H and J*, *Figure 2—figure supplement 1B*). In contrast, the depletion of CEP152, CEP192, or NEDD1 had no effect on the formation of caMTOCs (*Figure 2A–C and J*, *Figure 2—figure supplement 1B*), in agreement with the fact these proteins could not be detected within these structures. caMTOCs contained several centriole biogenesis factors, including CPAP, CP110, and CEP120, but lacked centrin and CEP135; however, the depletion of different centriolar proteins did not affect caMTOC formation (*Figure 2K*, *Figure 2—figure supplement 1C*). Within caMTOCs, we also detected a component of the HAUS complex (HAUS2), the centrosomal protein CEP170, dynein, dynactin, CLASP1/2, CLIP-115, CLIP-170, chTOG, KIF2A, and KIF1C (*Figure 2K*, *Figure 2—figure supplement 1C*). We tested the importance of some of these proteins for caMTOC formation by siRNA-mediated depletion (see *Figure 2K* for an overview), but among the tested proteins, only cytoplasmic dynein appeared essential for this process. In dynein-depleted cells, no clusters of pericentrin or other PCM components could be detected after centrinone B treatment (*Figure 2I–K*). It is important to note, however, that because we used siRNAs to reduce protein expression, we cannot exclude that the residual levels of some of the investigated proteins were sufficient to support caMTOC formation. Because we detected several microtubule plus-end tracking proteins

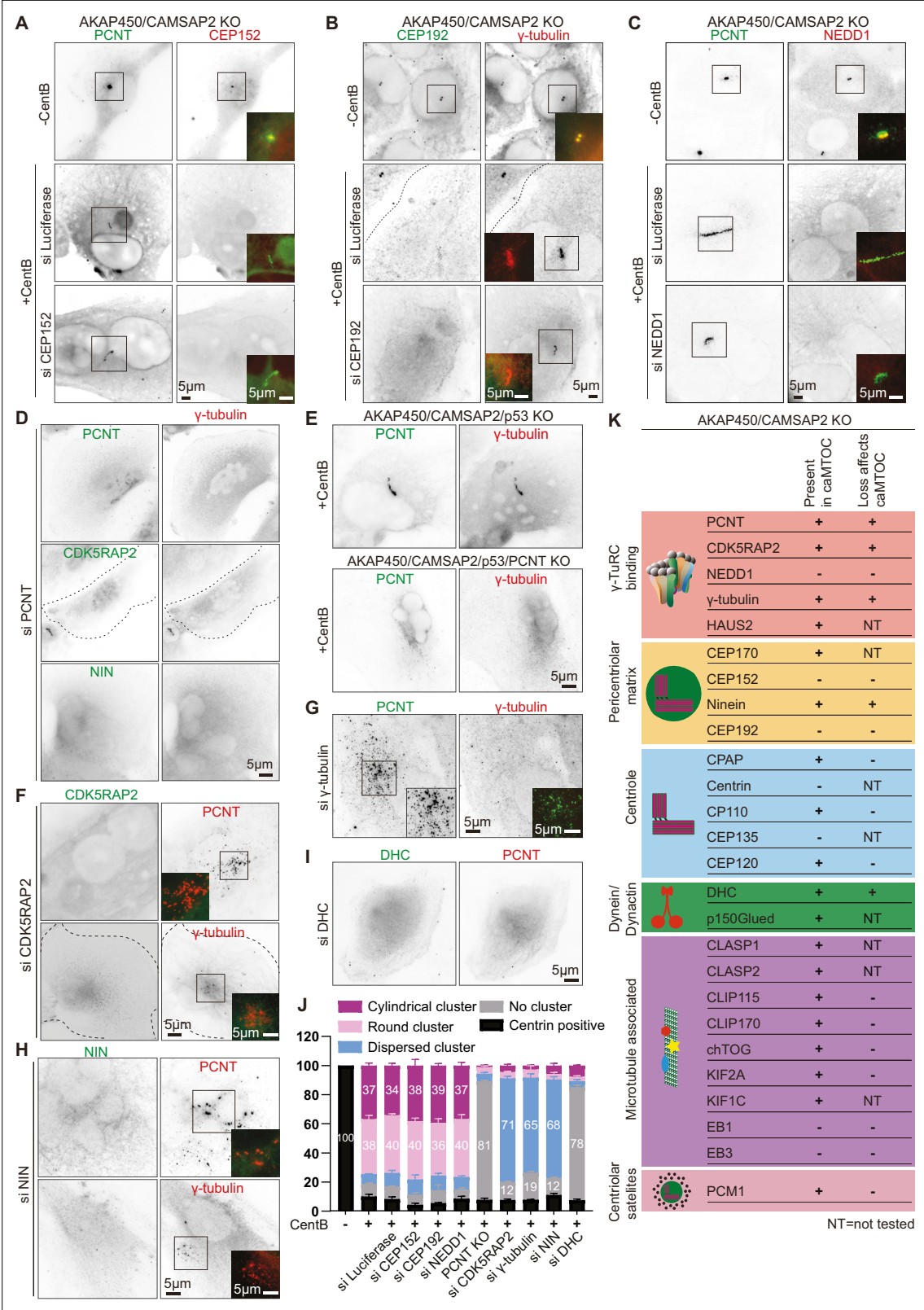

**Figure 2.** Molecular composition of caMTOCs in AKAP450/CAMSAP2 knockout cells. All the cells used in this figure are AKAP450/CAMSAP2 knockout cells, except for panel E as indicated. (**A–C**) Immunofluorescence images of control or centrinone-treated AKAP450/CAMSAP2 knockout RPE1 cells stained for and depleted of the indicated proteins. (**D, F–I**) Immunofluorescence images of centrinone-treated AKAP450/CAMSAP2 knockout RPE1 cells stained for and depleted of the indicated proteins. (**E**) Immunofluorescence images of centrinone-treated AKAP450/CAMSAP2/p53 knockout

*Figure 2 continued on next page*

*Figure 2 continued*

and AKAP450/CAMSAP2/p53/pericentrin knockout RPE1 cells stained as indicated. In panels A-I, insets show enlargements of the merged channels of the boxed areas, and dashed lines indicate cell edges. (J) Quantification of the main PCM organization types, as described in *Figure 1—figure supplement 2A*, for cells prepared as described in panel A-C, E-I. Numbers on the histogram show the percentages. 1293 (-CentB), 1547(+CentB), 2021(siLuci), 1822(siCEP152), 1345(siCEP192), 1161(siNEDD1), 2302 (AKAP450/CAMSAP2/p53/PCNT knockout (PCNT KO)), 2264(siCDK5RAP2), 2510(siγ-tubulin), 2408(siNIN) and 2526(siDHC) cells were analyzed for each measurement in three independent experiments (n=3). Values represent mean ± SD. (K) Summarizing table of PCM localization and the depletion effects on caMTOC formation in AKAP450/CAMSAP2 knockout RPE1 cells. NT, not tested.

The online version of this article includes the following source data and figure supplement(s) for figure 2:

**Source data 1.** An Excel sheet with numerical data on the quantifications shown in panel J.

**Figure supplement 1.** Characterization of PCM components localizing to caMTOCs in AKAP450/CAMSAP2 knockout cells.

**Figure supplement 1—source data 1.** Full raw unedited western blots shown in panels A and B.

**Figure supplement 2.** Generation of the AKAP450/CAMSAP2/p53/pericentrin knockout RPE1 cell line.

**Figure supplement 2—source data 1.** Full raw unedited western blots shown in panels F and K.

**Figure supplement 3.** Characterization of cell division in centrinone-treated AKAP450/CAMSAP2/p53 knockout cells.

**Figure supplement 3—source data 1.** An Excel sheet with numerical data on the quantifications shown in panel D.

**Figure supplement 4.** Characterization of PCM organization in AKAP450/CAMSAP2/p53/pericentrin knockout cells.

**Figure supplement 4—source data 1.** Full raw unedited western blots shown in panel B.

**Figure supplement 5.** Effects of the depletion or knockout of EB1 and EB3 on caMTOC formation in AKAP450/CAMSAP2 knockout cells.

**Figure supplement 5—source data 1.** Full raw unedited western blots shown in panel B.

(+TIPs) in the caMTOCs, such as CLIP-170, CLASP1/2 and the large subunit of dynactin, p150Glued, we also tested for the presence of the core components of the +TIP machinery, EB1 and EB3, but found that they were not enriched within the caMTOCs (*Figure 2K*, *Figure 2—figure supplement 5A*). Using the previously described cells that lack EB3, CAMSAP2 and the C-terminal partner-binding half of EB1 (*Yang et al., 2017*), we generated a knockout cell line that also lacks AKAP450 and found that caMTOCs could still form in these cells (*Figure 2K*, *Figure 2—figure supplement 5B, C*). We conclude that a subset of PCM components binds to each other in the absence of centrioles, and in AKAP450/CAMSAP2 knockout cells, these proteins form caMTOCs that recruit a number of additional PCM proteins and MAPs normally present in interphase centrosomes.

Despite containing only a subset of centrosomal proteins, caMTOCs strongly affected the organization and density of the microtubule network in acentriolar cells: microtubules were focused around caMTOCs if present and disorganized in cells lacking caMTOCs. The strongest loss of microtubule density was observed in cells lacking pericentrin, dynein or γ-tubulin, while milder phenotypes were observed in cells lacking CDK5RAP2 or ninein (*Figure 3A, B and E*). To further characterize microtubule organization after the loss of these proteins, we analyzed the proportion of the radial and non-radial microtubules. Whereas control cells (AKAP450/CAMSAP2 knockout cells treated with centrinone B and control siRNA) formed radial microtubule networks with ~12% non-radial microtubules, acentriolar cells lacking pericentrin or cytoplasmic dynein had ~46% non-radial microtubules, and the depletion of CDK5RAP2, ninein and γ-tubulin led to an intermediate phenotype with 25–30% non-radial microtubules (*Figure 3C, D and E*). An acentriolar PCM assembly containing pericentrin, CDK5RAP2, ninein, γ-tubulin and dynein is thus sufficient to form a radial microtubule array, similar to a centrosome, and PCM clustering promotes dense microtubule organization.

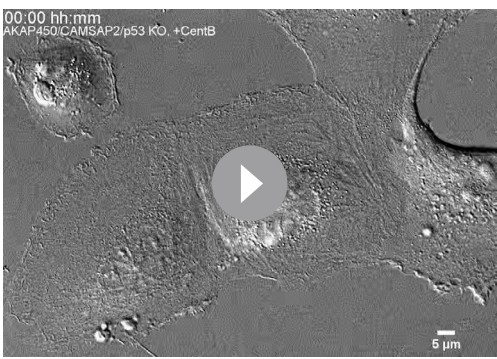

**Video 1.** Cell cycle progression in acentriolar AKAP450/CAMSAP2/p53 knockout cells. Cell cycle progression visualized by phase-contrast imaging of centrinone-treated AKAP450/CAMSAP2/p53 knockout RPE1 cells. The cells were imaged for ~15 hr with 1 min interval. Time is hh:mm.

https://elifesciences.org/articles/77892/figures#video1

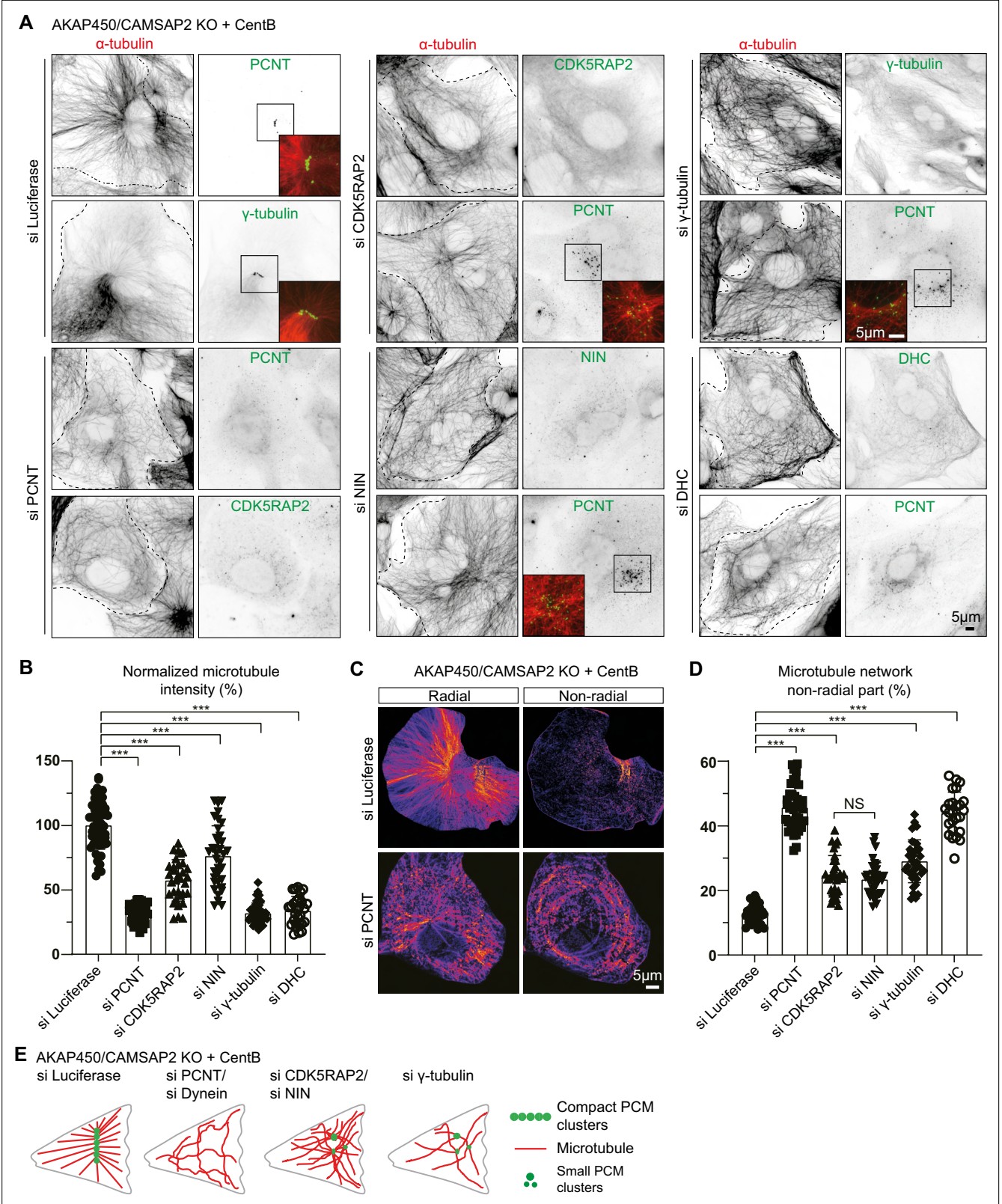

**Figure 3.** Microtubule organization in acentriolar cells missing different caMTOC components. (**A**) Immunofluorescence images of centrinone-treated AKAP450/CAMSAP2 knockout RPE1 cells depleted of the indicated proteins and stained for microtubules (α-tubulin, red) and different PCM proteins (green). Insets show enlargements of the merged channels of the boxed areas and dashed lines show cell boundaries. (**B**) Quantification of the normalized overall microtubule intensity for the indicated conditions. The number of cells analyzed in three independent experiments: n=56 (siLuci),

*Figure 3 continued on next page*

*Figure 3 continued*

45 (siPCNT), 33 (siCDK5RAP2), 36 (siNinein), 43 (siγ-tubulin), and 28 (siDHC). The statistical significance was determined by unpaired two-tailed Mann-Whitney test in Prism 9.1 (***$P<0.001$). Values represent mean ± SD. (**C**) Microtubule images were split into a radial and non-radial components (heat maps) based on microtubule orientation in relation to the PCM clusters or the brightest point, as described in Materials and methods. (**D**) Quantification of the proportion of the non-radial microtubules shown in panel C (see Materials and methods for details). The number of cells analyzed for each measurement in three independent experiments: n=25 (siLuci), 43 (siPCNT), 32 (siCDK5RAP2), 34 (siNinein), 37 (siγ-tubulin), and 25 (siDHC). The statistical significance was determined by unpaired two-tailed Mann-Whitney test in Prism 9.1 (***$p<0.001$). Values represent mean ± SD. (**E**) Diagram illustrating the distribution of PCM clusters and microtubule organization upon the depletion of the indicated proteins in centrinone-treated AKAP450/CAMSAP2 knockout cells.

The online version of this article includes the following source data for figure 3:

**Source data 1.** An Excel sheet with numerical data on the quantifications shown in panel B and D.

## Dynamics of caMTOC disassembly

To test whether the formation and maintenance of caMTOCs depends on microtubules, we depolymerized them by treating cells with nocodazole at 37 °C and found that caMTOCs fragmented into small clusters upon nocodazole treatment and reassembled into a single structure after nocodazole washout (*Figure 4A–C*). Because we found that caMTOC formation is dynein-dependent, we also included the dynein inhibitor dynapyrazole A in these experiments (*Steinman et al., 2017*). We confirmed that treatment with dynapyrazole A for 3 hr had no effect on dynein expression (*Figure 4D*) and found that the addition of this drug before nocodazole treatment prevented the disassembly of caMTOCs, whereas the treatment of cells with dynapyrazole A during nocodazole washout strongly inhibited caMTOC re-assembly (*Figure 4A–C*). These data indicate that both microtubule-dependent dispersal and coalescence of PCM clusters into caMTOCs are driven by dynein activity.

We next studied PCM dynamics using stably expressed GFP-CDK5RAP2 as a marker in live cells where microtubules were labeled with SiR-tubulin. GFP-CDK5RAP2 was mostly immobile within caMTOCs before nocodazole treatment (*Figure 1—figure supplement 3B*, *Video 2*). After a few minutes of nocodazole treatment, when the microtubule density was significantly reduced, small PCM clusters started to move out of the caMTOC and undergo rapid directional motility with speeds of up to 2 µm/s, which is within the range characteristic for cytoplasmic dynein (*Schlager et al., 2014*; *Figure 4E and F*, *Figure 4—figure supplement 1A*, *Video 2*). Once microtubules were completely disassembled, the movement of GFP-CDK5RAP2-positive clusters stopped, indicating that it is microtubule-dependent but occurs only when the microtubule network is partially depolymerized. Since cluster dispersal toward the cell periphery could be blocked by a dynein inhibitor, and since cytoplasmic dynein is a minus-end-directed motor, these data indicate that during microtubule disassembly by nocodazole at 37 °C, there is a transient stage when PCM clusters interact with only a few microtubules, some of which have their minus-ends facing outwards, and these microtubules serve as tracks for PCM transport. To support this idea, we used motor-PAINT, a technique that employs nanometric tracking of purified kinesin motors on the extracted cytoskeleton of fixed cells to super-resolve microtubules and determine their orientation (*Tas et al., 2017*). Using this approach, we determined microtubule orientations in centrinone-treated AKAP450/CAMSAP2 knockout cells and in cells that were also treated with nocodazole for 15 min to induce partial microtubule disassembly (*Figure 4G*, *Figure 4—figure supplement 1B*). We found that the cells contained a significant number of minus-end-out microtubules, and their proportion increased during early stages of nocodazole treatment, possibly because minus-end-out microtubules are more stable (*Figure 4G*, for example, minus-end-out microtubules constituted ~23% of the total microtubule length determined from kinesin-1 trajectories in the untreated cell and ~46% in the nocodazole-treated cell). These microtubules could serve as tracks for outward movement of PCM, causing the disassembly of caMTOC when the overall microtubule density around the caMTOC was strongly reduced (*Figure 4H*). These data suggest that the dense network of PCM-anchored microtubule minus-ends around a caMTOC allows for its compaction via dynein-mediated forces, but that dynein can pull the caMTOC apart when microtubules are disorganized.

To further confirm that caMTOC disassembly is an active microtubule-dependent process, we also depolymerized microtubules by a combination of cold (4 °C) and nocodazole treatment. When all microtubules were depolymerized, caMTOCs did not fall apart, even when the cells were subsequently warmed to 37 °C in the presence of nocodazole, so that microtubules could not re-grow

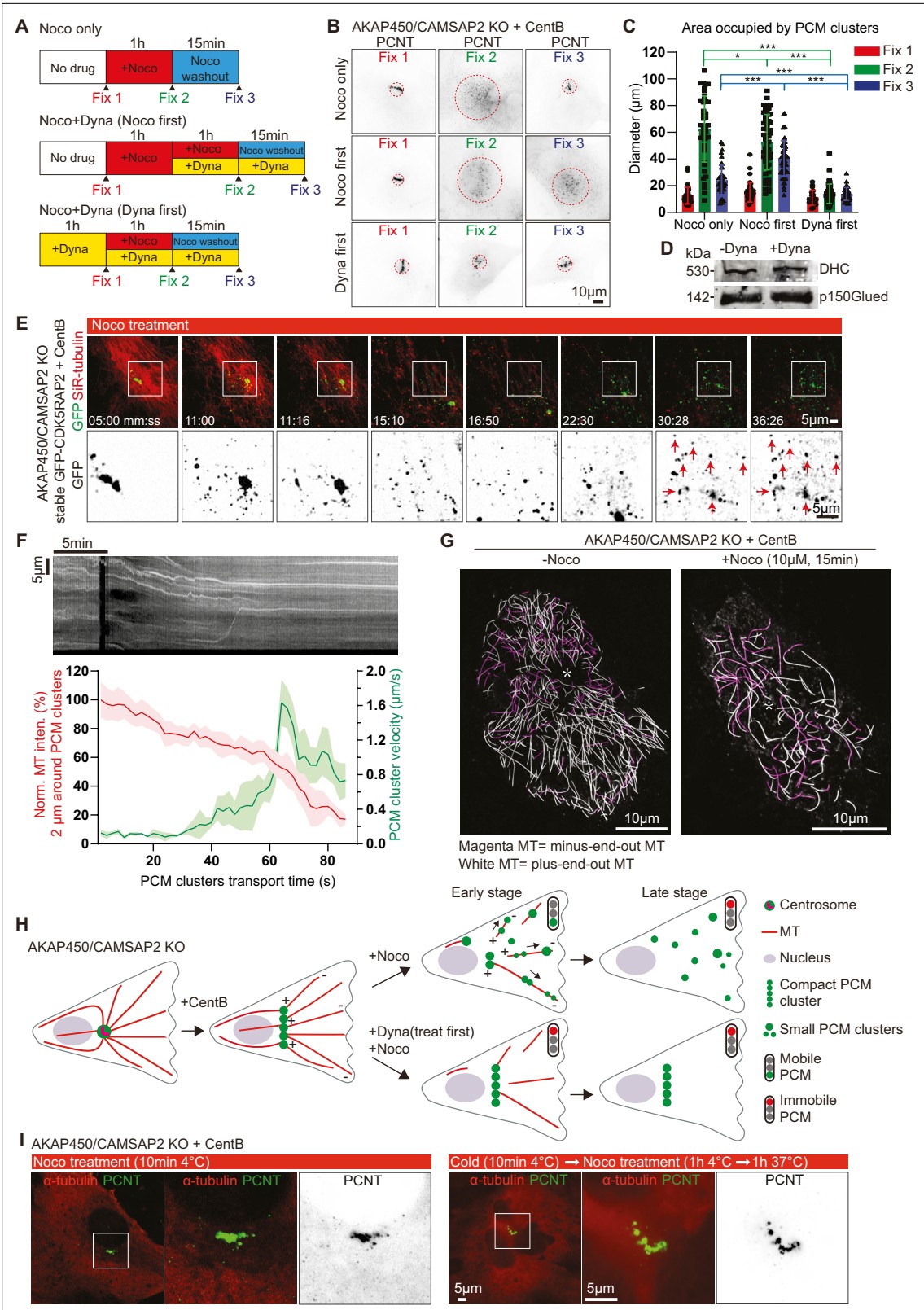

**Figure 4.** Microtubule- and dynein-dependent disassembly of caMTOCs. (**A**) Diagram illustrating different order of cell treatments with nocodazole (Noco) and/or dynapyrazole A (Dyna, 5 μM) and the time points when the cells were fixed. (**B**) Immunofluorescence staining of centrinone-treated AKAP450/CAMSAP2 knockout cells treated as shown in panel A. Dashed red circles represent the areas occupied by PCM clusters in each condition. (**C**) Quantification of the area occupied by PCM clusters in each condition, as shown in panels A and B. n=35–53 cells analyzed for each measurement

*Figure 4 continued on next page*

*Figure 4 continued*

in three independent experiments. The statistical significance was determined by unpaired two-tailed Mann-Whitney test in Prism 9.1 (not significant (NS), p<0.12; *p<0.033; ***p<0.001). Values are represented as mean ± SD. (**D**) Western blot showing that 3 hr treatment with dynapyrazole A does not affect the expression of the endogenous dynein heavy chain and the dynactin large subunit p150Glued. (**E**) Time-lapse images of centrinone-treated AKAP450/CAMSAP2 knockout RPE1 cells stably expressing GFP-CDK5RAP2. Microtubules were visualized by treating cells with 100 nM SiR-tubulin overnight. Red arrows show the immobilized PCM clusters at indicated timepoints. Time is min: s. Time-lapse images of the same cell prior to the nocodazole treatment were shown in *Figure 1—figure supplement 3B*. (**F**) (Top) Kymograph illustrating the motility of PCM clusters during microtubule disassembly with nocodazole. (Bottom) Measurements of the normalized microtubule (SiR-tubulin) fluorescence intensity (red plot, left Y-axis) and the instantaneous velocity of GFP-CDK5RAP2 clusters (green plot, right Y-axis) during the movement of GFP-CDK5RAP2 clusters away from caMTOC. Microtubule density around each PCM cluster was determined by measuring mean fluorescence intensity of SiR-tubulin in a circular area with a 2 μm radius centered on the PCM cluster and normalizing it to the mean fluorescence intensity of 20 images prior to nocodazole addition (set as 100%). The moment when a PCM cluster started to move out of the caMTOC was set as the initial time point (0 min) for this cluster, and the subsequent PCM cluster motion velocity and the relative local microtubule density of 43 time points were calculated and averaged. n=12 clusters were analyzed in each condition. Values are represented as mean ± SD. (**G**) Motor-PAINT images of centrinone-treated AKAP450/CAMSAP2 knockout RPE1 cells before and after nocodazole treatment. Plus-end-out microtubules are shown in white whereas minus-end-out microtubules are shown in magenta. Asterisks represent the putative position of caMTOC. (**H**) Summarizing diagram illustrating microtubule organization and the motility of GFP-CDK5RAP2-positive PCM clusters during nocodazole treatment and dynapyrazole A (treat first) and nocodazole co-treatment. (**I**) Immunofluorescence images of centrinone-treated AKAP450/CAMSAP2 knockout RPE1 cells stained for microtubules (α-tubulin, red) and PCM (PCNT, green). Enlargements show the merged and single channels of the boxed areas.

The online version of this article includes the following source data and figure supplement(s) for figure 4:

**Source data 1.** An Excel sheet with numerical data on the quantifications shown in panels C and F.

**Source data 2.** Full raw unedited western blots shown in panel D.

**Figure supplement 1.** PCM dynamics during nocodazole treatment in centrinone-treated AKAP450/CAMSAP2 knockout cells.

(*Figure 4I*). However, we noticed that in these conditions, the continuity and cylindrical organization of the PCM cluster were often perturbed. This raised the possibility that the elongated arrangement of PCM components within caMTOCs is microtubule-driven. Indeed, when cells were subjected to cold (4 °C) treatment in the absence of nocodazole, most of microtubules depolymerized, but some short cold-stable microtubules remained associated with the caMTOC (*Figure 4—figure supplement 1C*). These data indicate that PCM self-assembly in the absence of centrioles is microtubule-dependent, and microtubules are involved in shaping the PCM cluster. Once assembled, the PCM cluster is quite stable, unless microtubule organization is altered and dynein-driven microtubule-based transport pulls it apart.

## Dynamics of caMTOC assembly

When nocodazole-mediated microtubule disassembly was carried out at 4 °C, the caMTOC remained intact and after nocodazole washout it served as the major microtubule nucleation site, similar to the centrosome in untreated wild-type cells (*Figure 5—figure supplement 1A*). However, when nocodazole-mediated disassembly of the caMTOC was carried out at 37 °C, the cluster fell apart and reassembled upon nocodazole washout (*Figure 4A and B*), providing a way to study the dynamics of PCM self-assembly and the roles of different PCM components during this process. Small PCM clusters positive for pericentrin, CDK5RAP2, γ-tubulin and the centriolar satellite protein PCM1 that co-localized with the plus-ends of microtubules (labeled with EB1) could be detected 30 s

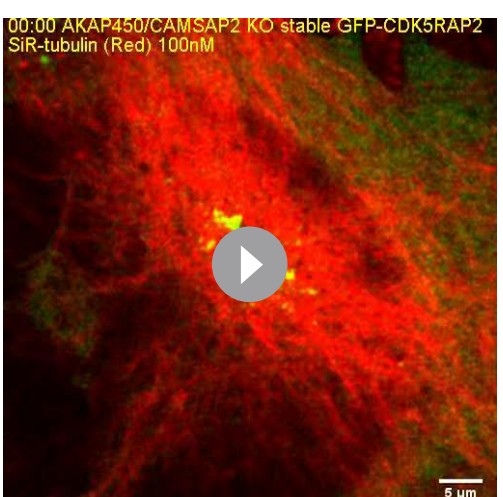

**Video 2.** caMTOC disassembly during nocodazole treatment of acentriolar AKAP450/CAMSAP2 knockout cells. PCM dynamics visualized by stable expression of GFP-CDK5RAP2 (green) in centrinone-treated AKAP450/CAMSAP2 knockout RPE1 cells. Microtubules were labeled overnight with 100 nM SiR-tubulin (red). The cell was imaged for ~3.5 min (100 frames, 2 s interval) prior to the addition of 10 μM nocodazole. Time is min: s.
https://elifesciences.org/articles/77892/figures#video2

after nocodazole washout; these PCM clusters and nascent microtubules did not colocalize with the Golgi membranes (*Figure 5A*, *Figure 5—figure supplement 1B*). Ninein was not detected within the clusters at this early stage of microtubule regrowth but could be found 2 min after nocodazole washout. In contrast, no clusters of CEP192 or NEDD1 were observed even 10 min after nocodazole washout (*Figure 5A*, *Figure 5—figure supplement 1B, C*). The depletion of pericentrin, CDK5RAP2 and γ-tubulin strongly inhibited microtubule nucleation in these conditions, whereas the depletion of dynein heavy chain or ninein had a milder effect (*Figure 5B and C*). Live cell imaging with GFP-CDK5RAP2 and SiR-tubulin showed that CDK5RAP2 clusters with attached microtubules coalesced by undergoing microtubule-based movements (*Figure 5D*), and measurements in cells fixed at different time points after nocodazole washout showed that a partly radial microtubule system emerged already 2 min after nocodazole washout (*Figure 5—figure supplement 1D*). Reassembly of a single caMTOC in the central part of the cell occurred within ~15 min after nocodazole washout, though it was less compact than in cells that were not treated with nocodazole (*Figure 5D–G*). Depletion of pericentrin, γ-tubulin and dynein heavy chain strongly inhibited the reformation of a radial microtubule network during nocodazole washout, whereas the effect of depleting CDK5RAP2 and ninein was less strong (*Figure 5—figure supplement 1D-F*). Live cell imaging of acentriolar AKAP450/CAMSAP2 knockout RPE1 cells stably expressing GFP-CDK5RAP2 showed that when pericentrin was depleted, CDK5RAP2 clusters were not detectable, and the microtubule network, both before nocodazole treatment and after nocodazole washout, was disorganized (*Figure 5—figure supplement 2*, *Video 3*). Taken together, our data show that pericentrin and γ-tubulin form microtubule-nucleating and anchoring units, which are clustered by the self-association of pericentrin and assembled into larger structures by dynein-based transport. CDK5RAP2 contributes to microtubule nucleation efficiency, whereas ninein appears to act somewhat later and contributes to the formation of a compact PCM cluster and a radial microtubule network. Importantly, all these proteins can cluster in the absence of centrioles, and together they can efficiently nucleate and anchor microtubules.

## The role of CAMSAP2-stabilized minus ends in defining microtubule network geometry

The results of nocodazole treatment and washout suggested that PCM can self-assemble into a caMTOC which nucleates and anchors microtubules, but this structure is sensitive to microtubule organization. This observation prompted us to investigate in more detail how the microtubules that are not anchored at PCM clusters affect PCM organization in steady state conditions. An abundant population of stable minus ends that do not attach to PCM is decorated by CAMSAP2. In centrinone-treated wild-type cells, CAMSAP2-bound microtubule minus ends were anchored at the Golgi (*Wu et al., 2016*; *Figure 1D*), whereas in centrinone-treated AKAP450 knockout cells they were distributed randomly (*Figures 1D and 6A*). Live imaging showed that CAMSAP2-decorated minus ends displayed only limited motility on the scale of hours and thus formed a relatively stationary, disorganized microtubule network (*Video 4*). Live imaging of GFP-CDK5RAP2 together with SiR-tubulin in these cells demonstrated that small PCM clusters were distributed throughout the cytoplasm (*Figure 6B*). These clusters moved along microtubules and encountered each other, but the direction of the movements was random and the clusters did not coalesce into a single structure (*Figure 6B*, *Figure 6—figure supplement 1A*, *Video 5*). Treatment with nocodazole and subsequent nocodazole washout confirmed that the motility of GFP-CDK5RAP2 clusters in centrinone-treated AKAP450 knockout cells was microtubule-dependent, and that these clusters could nucleate microtubules and move together with microtubule minus ends, but did not converge into a single caMTOC (*Figure 6—figure supplement 1A-C*, *Video 5*). Treatment with dynapyrazole A strongly inhibited the movements of small PCM clusters (*Figure 6—figure supplement 1D, E*, *Video 6*), indicating that they are dynein-driven. After the depletion of pericentrin, GFP-CDK5RAP2 became completely diffuse, and it failed to form clusters during nocodazole treatment or washout (*Figure 6—figure supplement 2*, *Video 7*), indicating that clustering of GFP-CDK5RAP2 in AKAP450 knockout cells is pericentrin-dependent. Based on these data, we conclude that in AKAP450 knockout cells, pericentrin still forms PCM clusters that can nucleate microtubules and can be moved by dynein along other microtubules, similar to what occurs in wild-type cells. However, in the absence of AKAP450, CAMSAP2-stabilized microtubules form a disorganized network, which imposes a random motility pattern on pericentrin-dependent

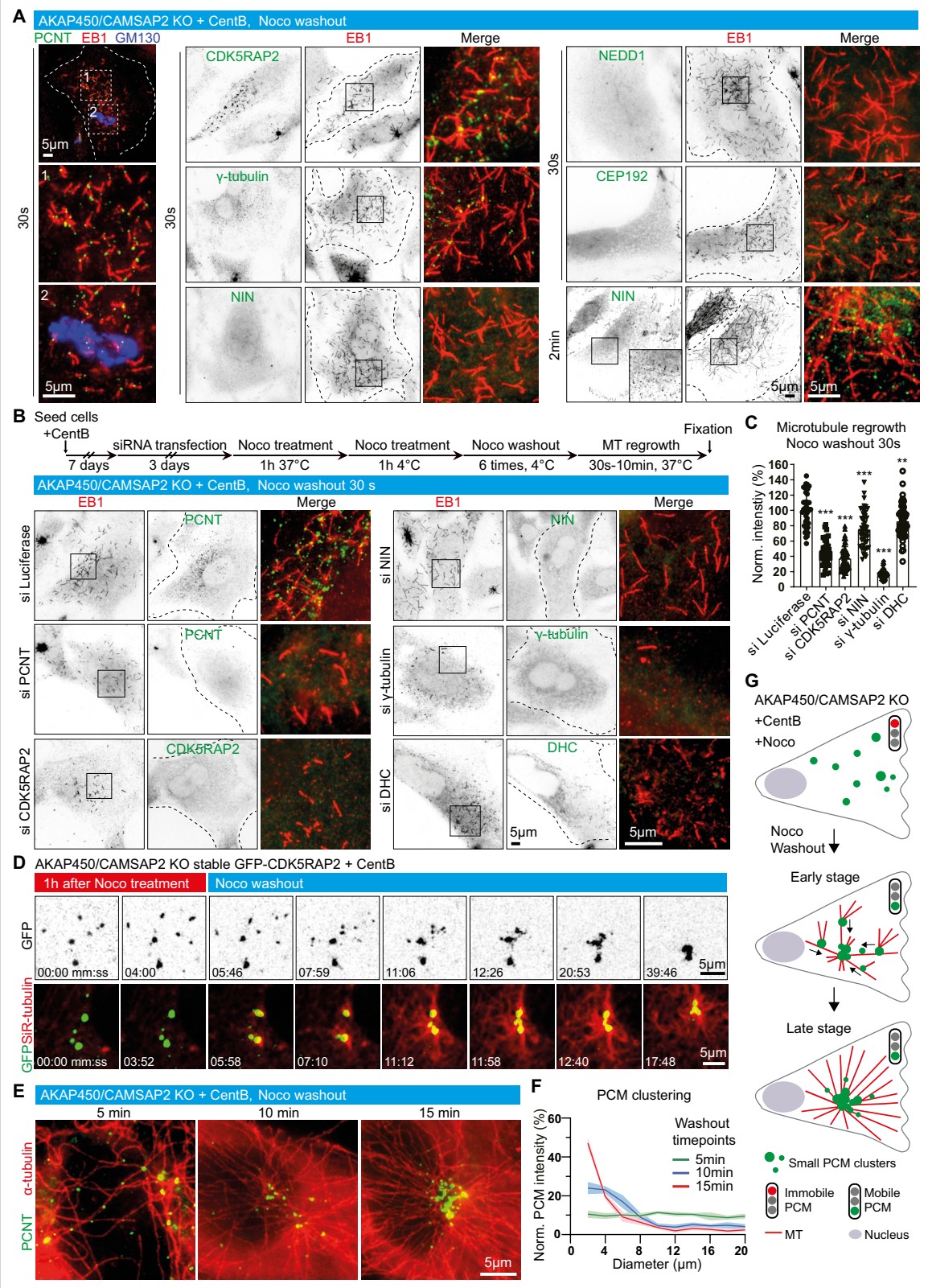

**Figure 5.** Dynamics of microtubule nucleation and caMTOC re-assembly in acentriolar cells. (**A**) Immunofluorescence images of microtubule regrowth after nocodazole washout at the indicated timepoints in centrinone-treated AKAP450/CAMSAP2 knockout RPE1 cells stained for PCM components (green) and newly nucleated microtubules (EB1, red). A Golgi marker, GM130 (blue), is included in the left row, and zooms of the boxed regions (numbered 1 and 2) show that microtubules nucleate from PCM clusters but not from the Golgi membranes. Dashed lines show cell boundaries. (**B**) (Top)

*Figure 5 continued on next page*

*Figure 5 continued*

Timeline shows the time course of protein depletion (siRNA transfection), nocodazole treatment, nocodazole washout and microtubule regrowth. (Bottom) Immunofluorescence images of microtubule regrowth experiments after depletion of the indicated proteins in centrinone-treated AKAP450/CAMSAP2 knockout RPE1 cells stained for the indicated PCM markers (green) and EB1 as a marker of nascent microtubules (red). Cell outlines are indicated with dashed lines and enlargements of the merged channels of the boxed areas are shown on the right. (C) Quantification of normalized microtubule intensity at 30 s after nocodazole washout in control cells and cells depleted of the indicated PCM proteins. n=40 (siLuci, siPCNT), 57 (siCDK5RAP2), 48 (siNIN), 45 (siγ-tubulin), and 50 (siDHC) cells analyzed for each measurement in three independent experiments. The statistical significance was determined by unpaired two-tailed Mann-Whitney test in Prism 9.1 (**p<0.002; ***p<0.001). Values are represented as mean ± SD. (D) Time-lapse images of centrinone-treated AKAP450/CAMSAP2 knockout RPE1 cells stably expressing GFP-CDK5RAP2 before and after nocodazole washout. Dispersed GFP-CDK5RAP2-positive PCM clusters (GFP, green) serve as microtubule nucleation sites (SiR-tubulin, red) and coalesce into a big cluster after nocodazole washout. Time is min: s. (E) Immunofluorescence images of centrinone-treated AKAP450/CAMSAP2 knockout RPE1 cells stained for pericentrin (green) and microtubules (α-tubulin, red) at the indicated timepoints after nocodazole washout. (F) Measurements of normalized fluorescence intensity of PCM clusters at the indicated distances in relation to the brightest point, as described in Materials and methods. The biggest PCM cluster (which normally also had the highest fluorescence intensity) was selected as the center, around which 10 concentric circles with 2 µm width were drawn. Fluorescence intensity of PCM clusters in these concentric circles was measured automatically and normalized by the sum of the total PCM intensity in each cell per condition. n=12 cells per plot per timepoint. Values represent mean ± SD. (G) Summarizing diagram illustrating microtubule organization and motility of GFP-CDK5RAP2-positive PCM clusters during nocodazole washout.

The online version of this article includes the following source data and figure supplement(s) for figure 5:

**Source data 1.** An Excel sheet with numerical data on the quantifications shown in panels C and F.

**Figure supplement 1.** PCM dynamics and microtubule regrowth during nocodazole washout in AKAP450/CAMSAP2 knockout cells.

**Figure supplement 1—source data 1.** An Excel sheet with numerical data on the quantifications shown in panel E.

**Figure supplement 2.** Pericentrin is required for PCM clustering in acentriolar AKAP450/CAMSAP2 knockout RPE1 cells.

PCM clusters and thus prevents their assembly into a single caMTOC, likely because PCM interactions are not sufficient to trigger their stable association (*Figure 6C*).

If the geometry of the CAMSAP2-stabilized microtubule network determines PCM distribution, focusing CAMSAP2-bound minus ends is expected to bring PCM together. To test this idea, we linked CAMSAP2-stabilized minus ends to a minus-end-directed motor. In order to avoid potential cell toxicity associated with manipulating cytoplasmic dynein, we used the motor-containing part of a moss kinesin-14, type VI kinesin-14b from the spreading earthmoss *Physcomitrella patens* (termed here ppKin14). The C-terminal motor-containing part of this protein can efficiently induce minus-end-directed motility of different cargoes in mammalian cells when it is tetramerized through a fusion with the leucine zipper domain of GCN4 (GCN4-ppKin14-VIb (861–1321)) and recruited to cargoes using inducible protein heterodimerization (*Jonsson et al., 2015*; *Nijenhuis et al., 2020*). We employed a chemical heterodimerization system that is based on inducible binding of two protein domains, FRB and FKBP, upon the addition of a rapamycin analog (rapalog AP21967, also known as A/C heterodimerizer) (*Clackson et al., 1998*; *Pollock et al., 2000*). To ensure that all CAMSAP2-decorated microtubule minus ends were linked to kinesin-14, we rescued centrinone-treated AKAP450/CAMSAP2 knockout cells by expressing CAMSAP2 fused to a tandemly repeated FKBP domain (2FKBP-mCherry-CAMSAP2) (*Figure 6D–F*). This construct was co-expressed with the FRB-GCN4-tagBFP-ppKin14 fusion, which by itself localized quite diffusely, with only a weak enrichment along

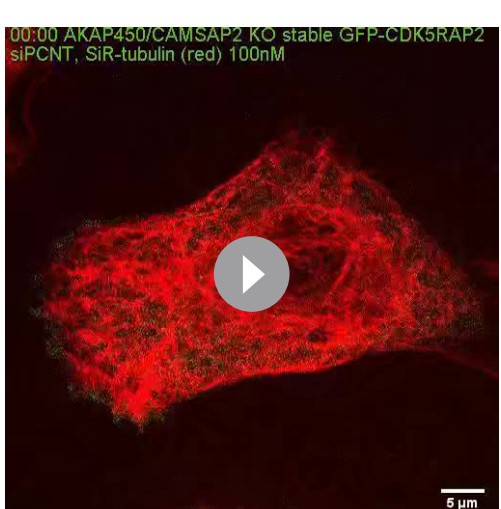

**Video 3.** Depletion of pericentrin inhibits PCM clustering in acentriolar AKAP450/CAMSAP2 knockout cells. A pericentrin-depleted acentriolar AKAP450/CAMSAP2 knockout cell stably expressing GFP-CDK5RAP2 (green) and labeled overnight with 100 nM SiR-tubulin (red) was imaged for ~4.5 min (140 frames, 2 s time interval) prior to treatment with 10 µM nocodazole. Nocodazole was washed out at ~20 min (frame 591), when all microtubules were depolymerized. Time is hr: min:s.

https://elifesciences.org/articles/77892/figures#video3

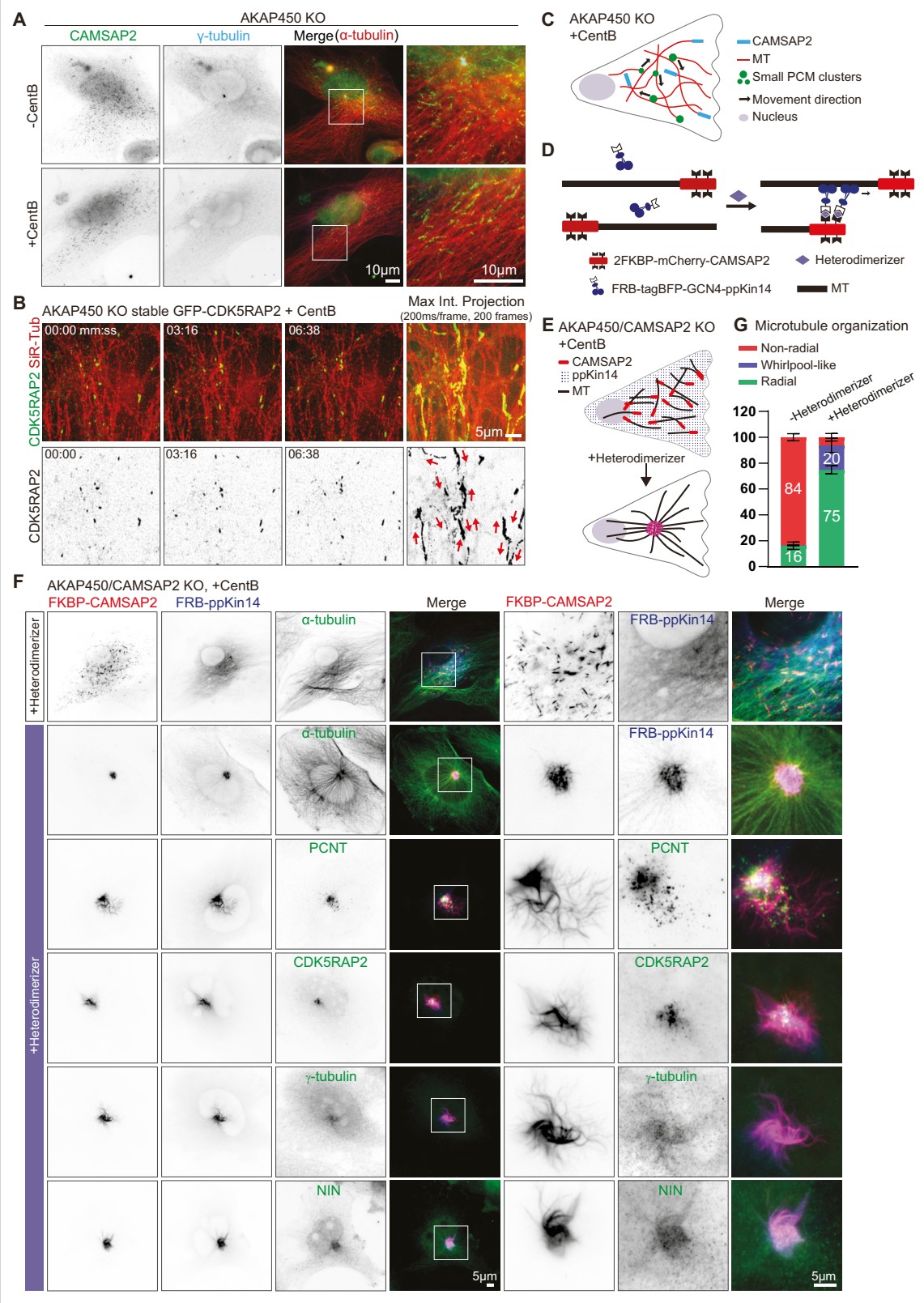

**Figure 6.** caMTOC formation in the presence of CAMSAP2 using inducible motor recruitment. (**A**) Immunofluorescence images of control or centrinone treated AKAP450 knockout RPE1 cells stained for CAMSAP2 (green), PCM protein (γ-tubulin, cyan) and microtubules (α-tubulin, red). Enlargements show the boxed regions of the merged images. (**B**) Time lapse images of centrinone treated AKAP450 knockout RPE1 cells stably expressing GFP-CDK5RAP2 (green). Microtubules were labeled with 100 nM SiR-tubulin overnight (red, top row). The maximum intensity projection includes 200

*Figure 6 continued on next page*

*Figure 6 continued*

frames, 200ms/frame. Red arrows show the motion directions of GFP-CDK5RAP2-positive PCM clusters. Time is min: s. (**C**) A diagram of microtubule organization and PCM motility in AKAP450 knockout cells. (**D, E**) Diagram of the inducible heterodimerization assay with ppKin14 and CAMSAP2. (**D**) CAMSAP2 was tagged with mCherry and fused to a tandemly repeated FKBP domain; tetramerized ppKin14 was tagged with TagBFP and fused to FRB. Heterodimerizer induces the binding of CAMSAP2 and ppKin14 by linking FKBP to FRB. (**E**) Heterodimerizer treatment induces the binding of CAMSAP2 (red) and ppKin14 (blue) and the formation of radial microtubule network. In this scheme, PCM-anchored microtubules are not shown. (**F**) Immunofluorescence images of centrinone-treated AKAP450/CAMSAP2 knockout RPE1 cells co-transfected with 2FKBP-mCherry-CAMSAP2 and FRB-TagBFP-GCN4-ppKin14 and stained for the indicated proteins in cells treated with DMSO or heterodimerizer. Zooms show the magnifications of boxed areas. (**G**) Quantification of the proportion of cells with a radial, whirlpool-like or non-radial microtubule organization with and without heterodimerizer treatment. Numbers on the histogram show the percentages. A total of 414 cells treated with DMSO (-Heterodimerizer) and 385 cells treated with heterodimerizer (+Heterodimerizer) analyzed for each measurement in three independent experiments (n=3). Values represent mean ± SD.

The online version of this article includes the following source data and figure supplement(s) for figure 6:

**Source data 1.** An Excel sheet with numerical data on the quantifications shown in panel G.

**Figure supplement 1.** PCM dynamics and the effects of CAMSAP2 clustering on PCM organization in centrinone-treated RPE1 cells lacking AKAP450.

**Figure supplement 1—source data 1.** An Excel sheet with numerical data on the quantifications shown in panel E.

**Figure supplement 2.** Pericentrin is required for PCM clustering in acentriolar AKAP450 knockout RPE1 cells.

---

microtubules, as described previously (*Nijenhuis et al., 2020*; *Figure 6D and F*). In the absence of heterodimerizer, CAMSAP2-decorated microtubule minus ends were distributed randomly, similar to endogenous CAMSAP2 in AKAP450 knockout cells (*Figure 6F*). However, upon heterodimerizer addition, ppKin14 was rapidly recruited to CAMSAP2-decorated microtubule ends, and after 2 hr, more than 90% of cells acquired a radial microtubule organization (*Figure 6E–G*). In heterodimerizer-treated cells, CAMSAP2-bound microtubule minus ends formed either a tight cluster or a 'whirlpool-like' ring in the cell center (*Figure 6D–G*, *Figure 6—figure supplement 1F*). The whirlpool-like arrangement likely comes about when CAMSAP2-stretches are a bit longer and continue to slide against each other, forming a nematic circular bundle. The major caMTOC components, pericentrin, CDK5RAP2, γ-tubulin, and ninein were also concentrated within the CAMSAP2 cluster (*Figure 6F*, *Figure 6—figure supplement 1F*). These data indicate that the positioning of stabilized minus-ends is an important determinant of the overall microtubule organization and PCM localization in interphase cells.

## The role of PCM in CAMSAP2-driven microtubule organization

Presence of PCM in the caMTOC induced by minus-end-directed transport of CAMSAP2 might be a passive consequence of microtubule reorganization but might also play an active role in forming this caMTOC. To distinguish between these possibilities, we attached CAMSAP2 to

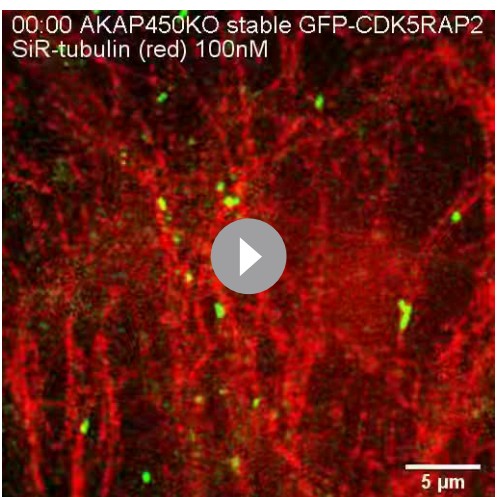

**Video 5.** PCM dynamics during nocodazole treatment and washout in acentriolar AKAP450 knockout cells. An acentriolar AKAP450 knockout RPE1 cell stably expressing GFP-CDK5RAP2 (green) and labeled overnight with 100 nM SiR-tubulin (red) was imaged for ~7 min (200 frames, 2 s interval) prior to treatment with 10 μM nocodazole. Nocodazole was washed out at ~27 min (frame 801) when all microtubules were depolymerized. Time is min: s.
https://elifesciences.org/articles/77892/figures#video5

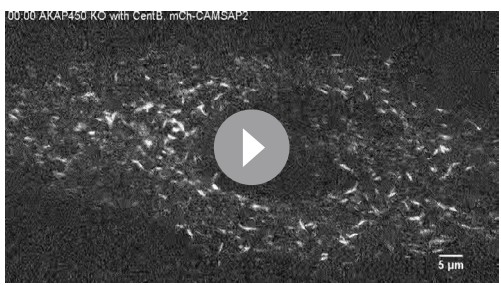

**Video 4.** CAMSAP2 distribution and dynamics in acentriolar AKAP450 knockout cells. An acentriolar AKAP450 knockout RPE1 cell transiently expressing mCherry-CAMSAP2 was imaged for 15 hrs (900 frames, 1 min interval). Time is hr:min.
https://elifesciences.org/articles/77892/figures#video4

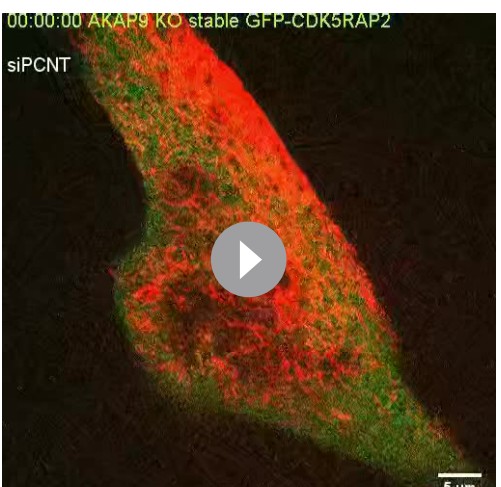

**Video 6.** PCM dynamics in acentriolar AKAP450 knockout cells are inhibited by dynapyrazole. An acentriolar AKAP450 knockout RPE1 cell stably expressing GFP-CDK5RAP2 (green) and labeled overnight with 100 nM SiR-tubulin (red) was imaged for ~12 min (350 frames, 2 s interval) prior to the dynapyrazole treatment, treated with 5 μM Dynapyrazole A for 1 hr, and then the same cell was imaged for ~12 min. Time is min: s.
https://elifesciences.org/articles/77892/figures#video6

ppKin14 in centrinone-treated cells where both AKAP450 and pericentrin were knocked out (AKAP450/CAMSAP2/p53/pericentrin knockout). In the absence of heterodimerizer, CAMSAP2-stabilized minus ends and the whole microtubule network were disorganized, as expected, and the same was true when the two constructs were expressed separately, with or without heterodimerizer (*Figure 7A and D*, *Figure 7—figure supplement 1*). After heterodimerizer addition, microtubules in cells expressing both constructs acquired a radial organization, but their minus ends usually did not converge in a single spot but rather accumulated in a~30–70 μm-large ring-like structure (*Figure 7A and D*, *Figure 7—figure supplement 2*, *Video 8*). Staining for PCM markers showed that CDK5RAP2 and γ-tubulin were enriched in the vicinity of CAMSAP2-positive microtubule minus ends, whereas ninein appeared rather diffuse (*Figure 7A*). To determine the nature of the structure 'corralled' by the ring of CAMSAP2-decorated minus ends in heterodimerizer -treated cells, we stained for different membrane organelles and found that although there was no strong correlation with the nucleus, Golgi membranes, or lysosomes, the majority of mitochondria were found within the CAMSAP2 ring, and the endoplasmic reticulum (ER) displayed increased density overlapping with the CAMSAP2 ring (*Figure 7B*, *Figure 7—figure supplement 2B*). It therefore appeared that in the absence of pericentrin, CAMSAP2-decorated minus ends were brought together by ppKin14, but their convergence was inefficient and possibly impeded by membrane organelles enriched in the central, thicker part of the cell before heterodimerizer addition (see the upper panel of *Figure 7—figure supplement 2B*). Transient transfection of centrinone-treated AKAP450/CAMSAP2/p53/pericentrin knockout cells with GFP-pericentrin rescued the formation of a tight CAMSAP2 cluster upon heterodimerizer treatment (*Figure 7C*). Our data show that pericentrin-containing PCM contributes to the formation of a caMTOC driven by minus-end-directed transport of CAMSAP2-stabilized minus-ends.

To support this notion further, we also generated cells that were knockout for AKAP450, CAMSAP2, CDK5RAP2, myomegalin (MMG, homologue of CDK5RAP2), p53 and pericentrin. To achieve this, we used the previously described RPE1 cell line knockout for AKAP450, CAMSAP2, CDK5RAP2, and MMG (*Wu et al., 2016*), in which we sequentially knocked out p53 and pericentrin (*Figure 7—figure supplement 3A-G*). While it was not possible to induce centriole loss by centrinone B treatment in AKAP450/CAMSAP2/CDK5RAP2/MMG knockout cells because the proliferation of these cells was arrested in the absence of centrioles (*Wu et al., 2016*), centriole removal was successful in AKAP450/CAMSAP2/CDK5RAP2/MMG/p53/pericentrin knockout cells due to the absence of p53 and led to microtubule disorganization (*Figure 7—figure supplement 3H*). Interestingly, when these cells were co-transfected with FKBP-linked CAMSAP2 and FRB-linked ppKin14 and treated with heterodimerizer (*Figure 6D*), we observed that CAMSAP2 clustering was even less efficient than in AKAP450/CAMSAP2/p53/

**Video 7.** Depletion of pericentrin inhibits PCM clustering in acentriolar AKAP450 knockout cells. A pericentrin-depleted acentriolar AKAP450 knockout cell stably expressing GFP-CDK5RAP2 (green) and labeled overnight with 100 nM SiR-tubulin (red) was imaged for ~3 min (90 frames, 2 s time interval) prior to treatment with 10 μM nocodazole. Nocodazole was washed out at ~23 min (frame 701) when all microtubules were depolymerized. Time is hr: min:s.
https://elifesciences.org/articles/77892/figures#video7

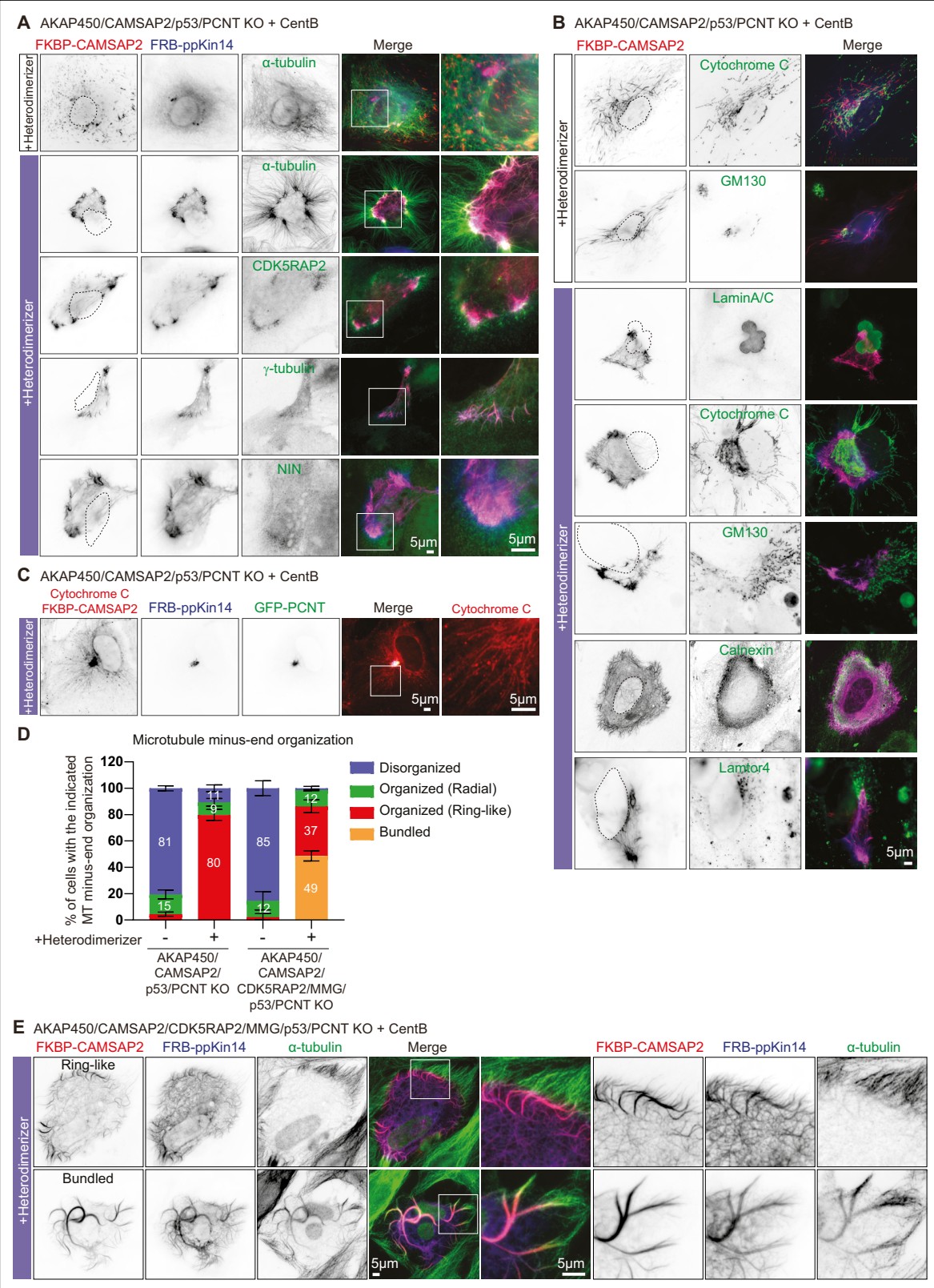

**Figure 7.** The role of the PCM in CAMSAP2-driven formation of caMTOCs. (**A,B**) Immunofluorescence images of centrinone-treated AKAP450/CAMSAP2/p53/pericentrin knockout RPE1 cells transfected with 2FKBP-mCherry-CAMSAP2 and FRB-TagBFP-GCN4-ppKin14 and stained for the indicated components before (top) or after an overnight heterodimerizer treatment. Zooms show magnifications of boxed areas. Black dashed lines show the position of the nucleus. (**C**) Cells treated as described for panel A were co-transfected with GFP-pericentrin and stained for mitochondria

*Figure 7 continued on next page*

*Figure 7 continued*

(cytochrome C, red) and CAMSAP2 (red) in same channel overnight after heterodimerizer addition. Zooms show magnifications of boxed areas. (**D**) Quantification of the proportion of cells with different types of microtubule minus end organization before and after overnight heterodimerizer treatment. Numbers on the histogram show the percentages. 334 (-Heterodimerizer), 424(+Heterodimerizer) cells of AKAP450/CAMSAP2/p53/ pericentrin knockout RPE1 cells, 206(-Heterodimerizer), and 239(+Heterodimerizer) of AKAP450/CAMSAP2/CDK5RAP2/MMG/p53/pericentrin knockout RPE1 cells analyzed for each measurement in three independent experiments (n=3). Values represent mean ± SD. (**E**) Immunofluorescence images of centrinone-treated AKAP450/CAMSAP2/CDK5RAP2/MMG/p53/pericentrin knockout RPE1 cells transfected with 2FKBP-mCherry-CAMSAP2 and FRB-TagBFP-GCN4-ppKin14 and stained for microtubules (α-tubulin, green) after an overnight heterodimerizer treatment. Zooms show magnifications of boxed areas.

The online version of this article includes the following source data and figure supplement(s) for figure 7:

**Source data 1.** An Excel sheet with numerical data on the quantifications shown in panel D.

**Figure supplement 1.** Inducible CAMSAP2-driven microtubule rearrangement in AKAP450/CAMSAP2/p53/pericentrin knockout cell.

**Figure supplement 2.** Inducible CAMSAP2-driven radial microtubule rearrangement in AKAP450/CAMSAP2/p53/pericentrin knockout cell.

**Figure supplement 3.** Generation of the AKAP450/CAMSAP2/CDK5RAP2/MMG/ p53/pericentrin knockout RPE1 cell line.

**Figure supplement 3—source data 1.** Full raw unedited western blots shown in panels F and G.

pericentrin knockout cells (**Figure 7D and E**). Forty-nine percent of AKAP450/CAMSAP2/CDK5RAP2/ MMG/p53/pericentrin knockout cells had small bundles of CAMSAP2 stretches dispersed throughout the cytoplasm, and only 37% of these cells formed a ring of CAMSAP2-decorated minus ends, whereas 80% of AKAP450/CAMSAP2/p53/pericentrin knockout cells formed such a ring. We examined the ER and mitochondria in these cells and found that in cells that did form a CAMSAP2 ring, the ER displayed an overlapping ring-like density, although the mitochondria inside the CAMSAP2 ring were more scattered compared to those of AKAP450/CAMSAP2/p53/pericentrin knockout cells (**Figure 7—figure supplement 2B**). In cells with dispersed CAMSAP2-positive bundles, no increased ER density or central accumulation of mitochondria were observed (**Figure 7—figure supplement 2B**). These data further support the notion that minus-end-directed transport of stable minus ends alone is insufficient to generate a caMTOC, and that synergy with PCM is required.

## Recapitulation of PCM self-organization by computer simulations

To rationalize the appearance of the different microtubule arrangements and to find a minimal set of interactions between filaments and motors that would lead to self-organization into structures that we observed experimentally, we set up agent-based computer simulations with Cytosim (**Nedelec and Foethke, 2007**). The numerical values for the biophysical parameters of the agents in our simulations have been taken from literature or reasonably estimated otherwise (see **Table 1** for details). The self-organized structures that form in a simulation also depend on the number of certain components of the system, such as microtubules. Because these numbers can vary and are not easy to precisely determine experimentally, we systematically explored their variation in silico (**Figure 8**, **Figure 8—figure supplement 1**). For simplicity, we considered a two-dimensional circular cell with a radius of 10 µm. We described a mobile PCM complex as a bead with a radius of 50 nm from which one microtubule plus end could grow. Microtubule growth and shrinkage were simulated with the classical microtubule model from Cytosim. When the microtubule reached a maximal length of 7.5 µm, its growth was stopped. In this way, we limited the microtubule length to avoid long microtubules that push their minus end to the periphery of the cell. Additionally, one dynein molecule was attached to a PCM complex with its cargo-binding domain. With this configuration, dynein molecules could transport PCM complexes along microtubules growing from other PCM complexes. Once they were bound to a microtubule, they walked

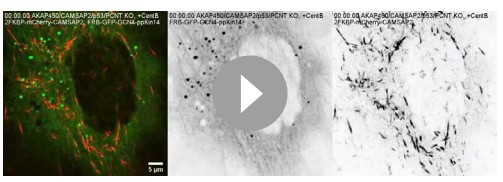

**Video 8.** Inducible CAMSAP2-driven radial microtubule rearrangement in an AKAP450/ CAMSAP2/p53/pericentrin knockout cell. An acentriolar AKAP450/ CAMSAP2/p53/pericentrin knockout cell transiently expressing 2FKBP-mCherry-CAMSAP2 (red) and FRB-GFP-GCN4-ppKin14 (green) was imaged for 10 min (100 frames, 6 s interval) prior to treatment with 100 nM heterodimerizer. Subsequently, the cell was imaged for ~1 hr and 35 min after heterodimerizer addition. Time is hr: min: s.

https://elifesciences.org/articles/77892/figures#video8

**Table 1.** Parameters with numerical values used in the Cytosim simulations.

| | Symbol | Value | Comment | Reference |
|---|---|---|---|---|
| **Cell** | | | | |
| Cell radius | | 10 µm | A typical RPE cell is ~50 µm wide. Therefore, the radius that we use is by a factor of ~2 smaller. We chose a smaller size to reduce the computational costs. | *Letort et al., 2016* |
| Viscosity | | 1 pN s/ µm² | Typical value for Cytosim simulations based on measurements in *C. elegans* embryos. | *Kole et al., 2005*; *Letort et al., 2016*; *Daniels et al., 2006* |
| Thermal energy | $kT$ | 4.2 pN nm | Thermal energy at room temperature | |
| **Dynein** | | | To parametrize dynein motors, we use the consensus numerical parameters discussed in Ohashi et al. | |
| Binding rate | $k_{on}$ | 5 s⁻¹ | | *Ohashi et al., 2019* |
| Force-free unbinding rate | $k^0_{off}$ | 0.1 s⁻¹ | | *Ohashi et al., 2019* |
| Detachment force | $F_d$ | 2 | | *Ohashi et al., 2019* |
| Binding range | | 75 nm | Estimated value based on the length of the motor, typically used in Cytosim simulations. | *Letort et al., 2016* |
| Force-free velocity | $v_0$ | 500 nm/s | | *Brenner et al., 2020* |
| Stall force | $F_s$ | 4 pN | | *Belyy et al., 2016* |
| **Kin14** | | | Because the biophysical parameters that describe a single kinesin-14 molecule are unknown, we assume numerical values as typical for kinesin-1 | |
| Binding rate | $k_{on}$ | 1 s⁻¹ | | *Klumpp et al., 2015* |
| Force-free unbinding rate | $k^0_{off}$ | 1 s⁻¹ | | *Berger et al., 2019* |
| Detachment force | $F_d$ | 3 pN | | *Pyrpassopoulos et al., 2020* |
| Binding range | | 75 nm | Typical length of a molecular motor | *Letort et al., 2016* |
| Force-free velocity | $v_0$ | 800 nm/s | | *Carter and Cross, 2005* |
| Stall force | $F_s$ | 7 pN | | *Carter and Cross, 2005* |

*Table 1 continued on next page*

*Table 1 continued*

| | Symbol | Value | Comment | Reference |
|---|---|---|---|---|
| **Microtubules** | | | The numerical values that we use to describe microtubule dynamics are typically used in Cytosim simulations and based on experimental measurements. The only new parameter that we introduced is the maximum length of a microtubule. | |
| Rigidity | | 20 pN µm$^2$ | Typical value used for Cytosim simulations based on experiments. | *Gibeaux et al., 2017*; *Gittes et al., 1993* |
| Catastrophe rate | $k_{cat}$ | 0.026 s$^{-1}$ | Published value | *Gibeaux et al., 2017* |
| Rescue rate | | 0 | We ignore rescues for simplicity | |
| Growing force | | 1.7 pN | | *Dogterom and Yurke, 1997* |
| Growing speed | $v_g$ | 0.13 µm/s | | *Burakov et al., 2003*; *Letort et al., 2016* |
| Shrinkage speed | | 0.272 µm/s | | *Burakov et al., 2003*; *Letort et al., 2016* |
| Maximum length | | 7.5 µm | To avoid boundary effects exerted by long microtubules pushing, we restrict microtubules to a maximum length. | |
| **Adhesive interactions of PCM complexes** | | | The biophysical parameters describing the adhesive interaction of PCM complexes are unknown. We introduced an effective model based on the implemented agents in Cytosim, and therefore the used values don't have a physical meaning. Overall, they generate an adhesive interaction between the PCM complexes which is not too weak and not too strong compared to the other forces that arise in the system. | |
| Binding rate | | 10 s$^{-1}$ | | |
| Binding range | | 100 nm | | |
| Unbinding rate | | 0.01 s$^{-1}$ | | |
| Detachment force | | 3 pN | | |
| **General parameters** | | | | |
| Stiffness of all linking elements | | 100 pN/ µm | Typical value for molecular motors | *Gros et al., 2021*; *Letort et al., 2016* |
| Total simulated time | | 18,000 s | Typical time scale of the experiments | |

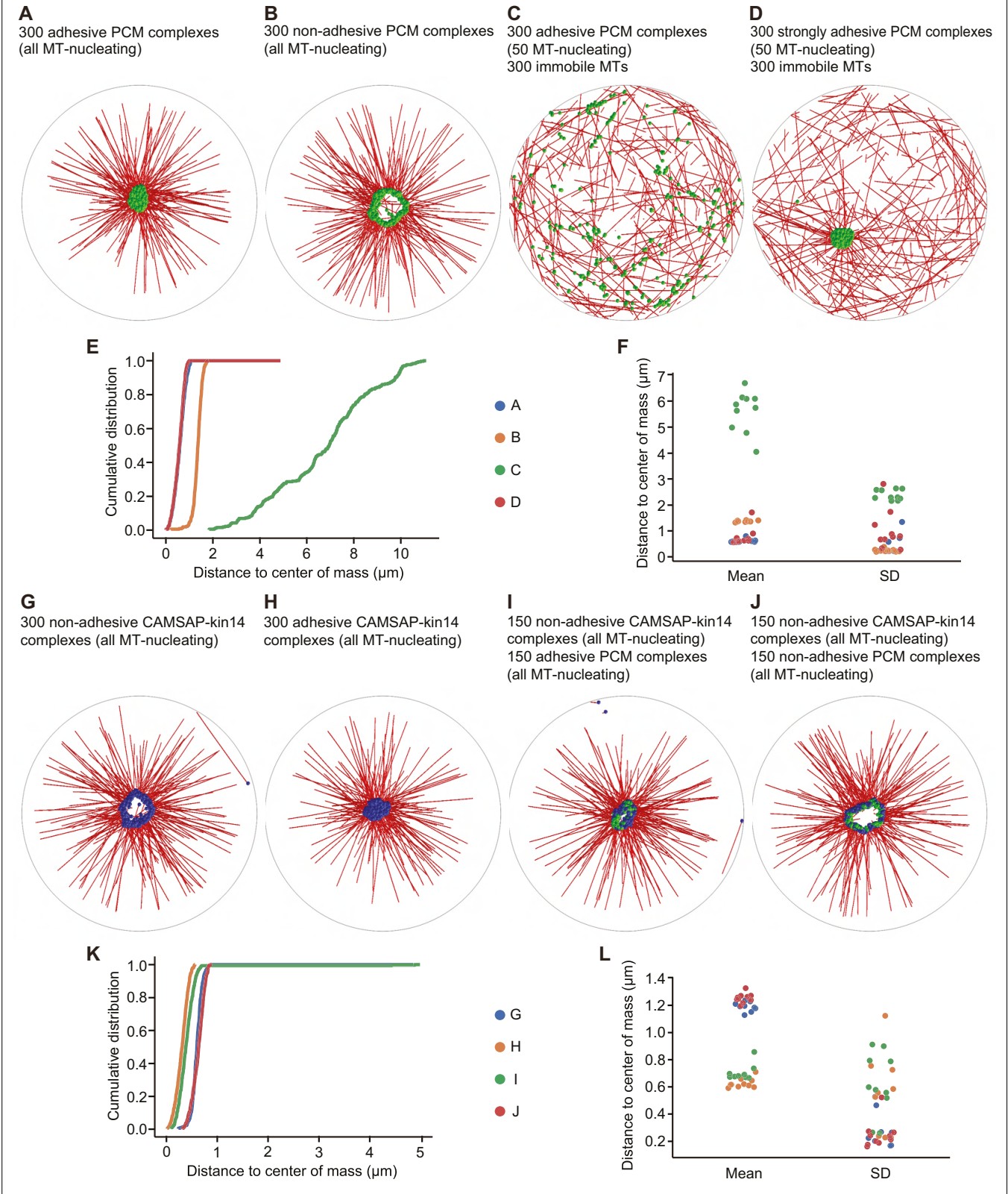

**Figure 8.** Cytosim simulations of PCM and microtubule self-organization. In all simulations, we considered a circular cell with a radius of 10 µm (gray line), containing microtubules (red), PCM complexes (green), and CAMSAP-kin14 complexes (blue). (**A**) Simulation of 300 PCM complexes; each PCM complex is attached to a dynein motor, and from each complex, a MT can grow. A weak adhesive interaction is introduced between the PCM complexes. A compact PCM cluster is observed. (**B**) The same simulation as in (**A**), but with the adhesive interaction disabled; PCM is organized in a

*Figure 8 continued on next page*

*Figure 8 continued*

ring-like structure. (**C**) The simulation in (**A**) modified in two ways: a randomly organized, dynamic microtubule network with 300 filaments is added, and only 50 of the 300 PCM complexes can nucleate microtubules. PCM cluster formation is disrupted. (**D**) The same simulation as in (**C**), but the strength of the adhesive interaction is increased. The formation of a compact cluster is recovered. (**E**) Cumulative distributions of distances of the PCM complexes to the average position (the center of mass), determined from the last frame of the simulations. (**F**) The mean and standard deviation (SD) of the distances of the PCM complexes to the center of mass from different repetitions of the simulations. Each system was simulated 10 times. Small values for the mean and for the standard deviation indicate a compact cluster. For a ring-like structure, the mean values are larger and describe the radius of the ring. A dispersed localization of PCM complexes is characterized by large values for the mean and the standard deviation. (**G**) Simulation of 300 CAMSAP-kin14 complexes, which consist of a microtubule nucleation site and five kinesin-14 motors, but do not adhere to each other. The complexes form a ring-like structure. (**H**) The same simulation as in (**G**), but with complexes that have the same weak adhesive interaction as for the PCM complexes in (**A**). The complexes form a central compact cluster. (**I**) Simulation of a mixed system with 150 CAMSAP-kin14 complexes that do not adhere to each other and 150 weakly adhesive PCM complexes. All complexes associate in a central compact cluster. (**J**) The same simulation as in (**I**), but with non-adhesive PCM complexes. A ring-like structure is formed. (**K**) Cumulative distributions of the distances from all complexes to the center of mass of the last frame of the simulation. (**L**) The mean and standard deviation of the distances from all complexes to the center of mass from different repetitions of the simulations. Each system was simulated 10 times. A compact cluster is characterized by a small value for the mean and for the standard deviation. A ring-like structure has a larger mean value.

The online version of this article includes the following source data and figure supplement(s) for figure 8:

**Source data 1.** An Excel sheet with numerical data of the quantifications shown in panels E, F, K, and L.

**Figure supplement 1.** Systematic alteration of the composition of the Cytosim simulations of PCM and microtubule self-organization.

**Figure supplement 1—source data 1.** An Excel sheet with numerical data of the shown quantifications.

**Figure supplement 2.** Manipulation of pericentrin localization in acentriolar AKAP450 knockout RPE1 cells through pericentrin homodimerization and CAMSAP2 overexpression.

toward the minus end in a force-dependent manner and were able to stochastically unbind along the way. When a dynein motor reached a minus end, it detached. Furthermore, we implemented a reversible binding interaction between PCM complexes to make them adhere to each other. The details of such an adhesive interaction are unknown, therefore we assumed parameters in the range typical for the binding interaction of microtubules and motor proteins. Because we were interested in the spatial arrangement of the PCM complexes, we introduced steric interactions between them. However, we neglected steric interactions between microtubules to effectively account for the three-dimensional space that we projected to two dimensions.

First, we sought to recapitulate the formation of a single PCM cluster. Simulations of 300 of such PCM complexes reproducibly reached a steady state in which they formed a compact centrally located cluster (*Figure 8A*, *Video 9A*). Quantification of 10 simulations showed that the mean and standard deviation of the distance from each PCM complex to the average location of all complexes (the center of mass) were small compared to the radius of the cell (*Figure 8E and F*). However, if we disabled the adhesive interactions between PCM complexes, they formed a loose ring-like arrangement and not a compact cluster (*Figure 8B*, *Video 9B*), and the mean and standard deviation of the distance of PCM complexes to the center of mass were larger (*Figure 8E and F*). Systematic variation of the number of PCM complexes in the system indicated that approximately 150 PCM complexes were needed to observe robust clustering (*Figure 8—figure supplement 1A*), which fits with the fact that most cultured mammalian cells such as RPE1 typically contain several hundred microtubules. Increasing the number of microtubules nucleated from one PCM complex did not affect the properties of the system (*Figure 8—figure supplement 1B*). Taken

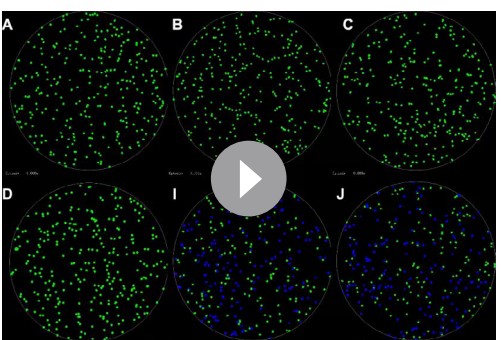

**Video 9.** Cytosim simulations of PCM and microtubule self-organization. Example videos of our Cytosim simulations. Each simulation represents 1800 seconds of real time, and the last frame is shown in the panels of Figure 8. In all simulations, we considered a circular cell with a radius of 10 μm (gray line), containing microtubules (red), PCM complexes (green), and CAMSAP-kin14 complexes (blue). The videos correspond to panels A, B,C, D, I, and J of Figure 8, as indicated.

https://elifesciences.org/articles/77892/figures#video9

together, our simulations support the idea that PCM components form a central cluster through positive feedback of dynein molecules carrying them toward the minus ends of microtubules attached to other PCM proteins, and that adhesive interactions between the PCM complexes promote cluster compaction. For such self-organization to emerge, a sufficient number of PCM complexes must be present in the system.

In our experiments, we saw that the presence of CAMSAP2-stabilized microtubule network prevented PCM clustering (*Figure 1D and E*, *Figure 6A–C*), and that CAMSAP2-bound minus ends appeared almost stationary on the scale of minutes (*Video 4*) compared to the rapid dynein-driven movement of small PCM clusters (*Video 5*). To simulate such a situation, we assumed that 300 microtubules could grow and shrink with random orientations from CAMSAP2-stabilized microtubule ends that were randomly distributed and stationary within the cell. Furthermore, we assumed that some of the moving PCM complexes were not efficient in nucleating microtubules, because the presence of a PCM-independent microtubule population would reduce the concentration of soluble tubulin and thus nucleation efficiency. To effectively account for such an effect, we allowed only 50 of the 300 PCM complexes to nucleate microtubules. In this system, the PCM cluster formation was disrupted, and PCM complexes randomly moved around the cell (*Figure 8C*, *Video 9C*). The distributions of the distances to the average position of the PCM complexes were broad and had a large mean and standard deviation, indicating the dispersed localization (*Figure 8E and F*). However, if we increased the strength of adhesive interactions between the PCM complexes, the randomly oriented microtubule network no longer suppressed the clustering of PCM complexes (*Figure 8D*, *Video 9D*). This outcome suggested that if self-association of pericentrin were increased, it would still form a cluster even in the presence of dispersed CAMSAP2-stabilized minus ends. To test this prediction, we have fused pericentrin to an inducible homodimerization domain FKBP, a variant of FKBP12 domain that homodimerizes in presence of rapamycin analog AP20187 (B/B homodimerizer) (*Clackson et al., 1998*; *Pollock et al., 2000*). We found that this modified pericentrin formed small dispersed clusters in centrinone-treated AKAP450 knockout protein, similar to endogenous pericentrin. However, the addition of the homodimerizer compound triggered strong pericentrin clustering, whereas CAMSAP2-decorated minus ends remained dispersed (*Figure 8—figure supplement 2A*). The results of this experiment were thus in line with our simulations.

Next, we varied the number of components in the system and found that when the number of immobile microtubules was less than 150, clustering of PCM complexes was still observed (*Figure 8— figure supplement 1C*). Likewise, when we increased the fraction of PCM complexes nucleating microtubules, 300 immobile microtubules were not sufficient any more to disperse the motion of PCM complexes. When 100 of the 300 PCM complexes could nucleate microtubules, the system became bistable: a cluster formed in some simulations, but not in others. A further increase of the fraction of PCM complexes nucleating microtubules led to robust clustering (*Figure 8—figure supplement 1D*). We conclude that the presence of a stationary randomly organized microtubule network can be sufficient to prevent the formation of a compact PCM cluster even though individual PCM complexes can still move with dynein and adhere to each other, provided that the interaction between PCM complexes is not too strong and that the number of immobile microtubules is sufficiently high.

We also explored a system where the localization of immobile minus ends was biased toward the cell periphery, by randomly placing their minus ends in a 1 μm-broad region adjacent to the cell boundary. Increasing the fraction of peripherally placed immobile microtubules led to the enrichment of the PCM complexes at the cell periphery (*Figure 8—figure supplement 1E*). To support this finding experimentally, we overexpressed mCherry-CAMSAP2 in centrinone-treated AKAP450 knockout cells. Overexpression of CAMSAP2 occasionally led to minus-end bundling and enrichment in certain cell areas, and in such cells, pericentrin clusters were enriched in the same cell regions (*Figure 8—figure supplement 2B*).

Next, we examined how CAMSAP2-stabilized microtubules would organize when minus-end-directed kinesin-14 motors could attach to them and carry the minus ends. CAMSAP2-stabilized microtubule ends bound to kinesin-14 motors (termed 'CAMSAP-kin14 complexes' in the text below) were modelled as a bead with a radius of 50 nm from which a microtubule could grow and to which 6 kinesin-14 molecules were attached. Because the biophysical properties of single plant kinesin-14 molecules used in our assays are poorly understood, we assumed typical kinesin-1 values with an opposite directionality. CAMSAP-kin14 complexes only interacted sterically but did not adhere to

each other. In a simulation of 300 CAMSAP-kin14 complexes, a loose ring-like arrangement appeared (*Figure 8G*), similar to the one observed in our simulations of PCM complexes that could not bind to each other (*Figure 8B*). Similar to adhesive PCM complexes, CAMSAP-kin14 complexes could not self-organize when their number was low (*Figure 8—figure supplement 1F*). Introducing adhesive interactions between CAMSAP-kin14 complexes was sufficient to promote compact cluster formation (*Figure 8H*).

Finally, to investigate if a compact cluster of CAMSAP-kin14 complexes would emerge in the presence of self-associating PCM complexes, we set up simulations with 150 CAMSAP-kin14 complexes together with 150 adherent PCM complexes. The steady state of such a system displayed a compact central cluster, in which both types of complexes were mixed (*Figure 8I, K and L*, *Video 9I*), while making PCM complexes non-adhesive prevented cluster compaction (*Figure 8J, K and L*, *Video 9J*). To induce compaction, at least a half of microtubules had to be attached to adhesive PCM components rather than non-adhesive CAMSAP-kin14 complexes (*Figure 8—figure supplement 1G*). Taken together, our simulations suggest that PCM complexes can provide enough adhesive interactions to compact the cluster of non-interacting CAMSAP-kin14 and PCM complexes, very similar to our experimental observations.

## Discussion

In this study, we explored the mechanisms of interphase PCM self-assembly in the absence of centrioles. Our experiments and agent-based Cytosim simulations support the idea that complexes of PCM proteins can form a single cluster through a positive feedback mechanism, whereby dynein motors carry microtubule minus-ends to other microtubule minus ends (*Cytrynbaum et al., 2004*). However, for a compact cluster to emerge, the minus ends must not only be able to move toward each other but also to bind to each other, and this idea is fully supported by our simulations. Interphase PCM is thus capable both of dynein binding and self-association sufficient to organize a compact microtubule-nucleating and anchoring structure in the absence of centrioles. However, the formation of a compact MTOC in acentriolar cells is slow and less robust than in centriole-containing cells, and the resulting structure is sensitive to the overall organization of microtubules and to dynein function. This means that interphase PCM self-association is by itself reversible and not sufficiently tight to resist dynein-driven forces. Centrioles can thus be regarded as catalysts of PCM assembly and stabilizers of interphase centrosomes, preventing PCM movement on microtubules oriented with their minus ends away from the centrosome.

The simulations that we developed allowed us not only to rationalize the emergence of the self-organized structures that we experimentally observed, but also to systematically change the relative numbers of the components of the system, a manipulation that is difficult or impossible in cells. We found that the emergence of specific organizations is robust if the number of components in the system is sufficiently high. For a fixed cell volume, a larger number of components implies a larger concentration and density. In low-density systems, the probability that components are close to each other and interact is small, and therefore structures cannot form or only form on very long time scales. All of the interactions that we defined were reversible, and therefore the observed structures were dynamic. Therefore, for persistent structures to emerge, the probability for reengagement between agents must be sufficiently large, which is only the case if the density of components is adequately high. This interplay of finding components, engaging in interactions, disengaging and finding again other components is the underlying process for the dynamic structures to form. It is challenging to quantitatively relate all parameters of the system to each other, because of their interdependence. Therefore, we expect that the absolute thresholds for structure formation that we report in this study depend on all other parameters of the system. Despite this complexity, we were able to identify robust parameter ranges, in which the experimentally observed structures could be recapitulated in computer simulations by only varying the number of components and keeping the specific interactions of components the same.

Our experimental system allowed us to examine which PCM components are capable of associating with each other and with dynein to promote microtubule nucleation and anchoring independently of centrioles. The major scaffold for interphase acentriolar PCM assembly is pericentrin, which can self-associate (*Jiang et al., 2021*) and form clusters that recruit CDK5RAP2 and ninein, two proteins known to bind to pericentrin (*Chen et al., 2014*; *Delaval and Doxsey, 2010*; *Kim et al.,*

*2014*; *Lawo et al., 2012*). CDK5RAP2 is important for efficient microtubule nucleation, consistent with its role as an activator of γ-TuRC (*Choi et al., 2010*). The same may be true for pericentrin itself (*Takahashi et al., 2002*) and, to a lesser extent, for ninein (*Delgehyr et al., 2005*; *Mogensen et al., 2000*), which was less important for microtubule nucleation in our assays. Ninein might be required for promoting minus-end anchoring, as proposed by previous studies (*Abal et al., 2002*; *Chong et al., 2020*; *Delgehyr et al., 2005*; *Goldspink et al., 2017*; *Lechler and Fuchs, 2007*; *Mogensen et al., 2000*; *Shinohara et al., 2013*; *Zheng et al., 2020*). The importance of ninein for the formation of caMTOCs highlights its function within the PCM independent of its role at the centriolar appendages, a major site of ninein localization at the centrosome (*Chong et al., 2020*; *Delgehyr et al., 2005*; *Sonnen et al., 2012*). Pericentrin can also directly interact with dynein through the dynein light intermediate chain (*Purohit et al., 1999*). Ninein was shown to form a triple complex with dynein and dynactin and activate dynein motility (*Redwine et al., 2017*), and an interaction between CDK5RAP2 and dynein has also been reported (*Jia et al., 2013*; *Lee and Rhee, 2010*). All these interactions likely contribute to dynein-dependent PCM coalescence in the absence of centrioles and also to the function of PCM in microtubule organization, as in the absence of PCM clustering, microtubule density is strongly reduced even though all PCM proteins are expressed.

Pericentrin-dependent MTOC assembly has also been observed in acentriolar mitotic cells (*Chinen et al., 2021*; *Watanabe et al., 2020*), but the interphase pathway displays some interesting differences. Most notably, CEP192 and its binding partners CEP152 and NEDD1 (*Gomez-Ferreria et al., 2012*; *Joukov et al., 2014*; *Kim et al., 2013*; *Sonnen et al., 2013*) were not enriched at caMTOCs, and their depletion appeared to have no impact on caMTOC formation. This is in line with earlier observations showing that microtubule-nucleating clusters of PCM components in acentriolar AKAP450 knockout cells contain pericentrin, CDK5RAP2 and ninein, but not CEP192 (*Gavilan et al., 2018*). In contrast, in mitotic acentriolar cells that rely on pericentrin and CDK5RAP2 for spindle pole formation, CEP192 is recruited to pericentrin clusters (*Chinen et al., 2021*; *Watanabe et al., 2020*). Moreover, although NEDD1 is targeted to centrosomes by CEP192, pericentrin can also contribute to the centrosomal targeting of NEDD1 independently of CEP192 during mitosis (*Chi et al., 2021*). The interactions between the components of pericentrin- and CEP192 pathways thus seem to be stronger in mitosis than in interphase, where these proteins are brought together by the centrioles. The relative importance of the two pathways is also different depending on the phase of the cell cycle and the presence of centrioles: unlike pericentrin or CDK5RAP2, CEP192 is essential for cell division (*Gomez-Ferreria et al., 2007*; *Joukov et al., 2014*; *Yang and Feldman, 2015*; *Zhu et al., 2008*), and it is more important than pericentrin or CDK5RAP2 for microtubule nucleation at interphase centrosomes (*Gavilan et al., 2018*). However, the situation is different in differentiated cells, where centrosome function is suppressed. Pericentrin, ninein, CDK5RAP2 and γ-TuRC, but not CEP192 are present in non-centrosomal MTOCs at the nuclear envelope and the Golgi membranes in muscle cells, where they are targeted by AKAP450 (*Gimpel et al., 2017*; *Oddoux et al., 2013*; *Vergarajauregui et al., 2020*). Recent work also demonstrated that at the ciliary base of certain types of worm neurons, there is an acentriolar PCM-dependent MTOC that is formed by the functional counterparts of CDK5RAP2 (SPD-5), pericentrin (PCMD-1) and γ-tubulin, but lacks CEP192/SPD-2 (*Garbrecht et al., 2021*; *Magescas et al., 2021*). The co-assembly of pericentrin, ninein and CDK5RAP2 counterparts into structures that can promote γ-TuRC-dependent microtubule nucleation and also anchor minus ends may thus be a general property of interphase acentriolar MTOCs. Additionally, the participation of dynein is likely to be a general feature of acentrosomal microtubule organization, as exemplified by MTOCs formed by the Golgi apparatus or endosomal membranes (*Liang et al., 2020*; *Zhu and Kaverina, 2013*).

Our simple cellular system allowed us to dissect the molecular details of acentriolar PCM assemblies. We found that caMTOCs recruited multiple MAPs and a subset of centriolar proteins. For example, caMTOCs accumulated several +TIPs, such as CLASPs, CLIP170 and chTOG, though the depletion of these proteins had no effect on caMTOC formation or function. In contrast, the core components of the +TIP complexes, EB1 and EB3, were neither enriched in caMTOCs nor required for their assembly. While the negative results on other +TIPs might be due to their incomplete depletion, EB1 and EB3 function was tested using genetic knockouts, indicating that interphase PCM function is EB-independent. Among the centriolar proteins, we detected CPAP, CP110 and CEP120 in caMTOCs, and it will be interesting to test whether any of these microtubule-binding factors contribute to microtubule organization independently of their participation in centriole and centrosome assembly.

Furthermore, our work provided insight into the self-assembly properties of interphase PCM components. Previous work showed that some PCM proteins, such as pericentrin, CDK5RAP2 and ninein, or their counterparts, can form mobile clusters that contribute to MTOC maintenance (*Dictenberg et al., 1998*; *Magescas et al., 2019*; *Megraw et al., 2002*; *Moss et al., 2007*), but the nature of these clusters may differ. For example, oligomerization and clustering of pericentrin molecules is important for MTOC formation both in interphase and mitosis; during mitotic entry, pericentrin forms condensates (*Jiang et al., 2021*), while our observations in interphase show that pericentrin clusters display no hallmarks of liquid droplets or condensates. Moreover, the compaction and shape of acentriolar clusters of PCM proteins depend on dynein and microtubules. Our experiments showed that while caMTOCs were stable when all microtubules were disassembled, indicating that attractive interactions between PCM components are sufficient to keep them together, cold treatment experiments suggested that the cylindrical shape of caMTOCs is likely due to their association with some stable microtubules. Furthermore, when the number of PCM-unattached stable minus-end-out microtubules was increased by the presence of CAMSAP2, PCM clusters could no longer form a compact structure but continued to move along microtubules. Therefore, an important outcome of our study is the key role of stabilized minus ends in determining interphase organization of PCM components in the absence of the centrioles. Our simulations showed that in order to suppress clustering of the PCM complexes, the number of stabilized immobile minus ends should be relatively high compared to the number of PCM-complex-anchored microtubules. This explains why caMTOC falls apart during nocodazole treatment: PCM-anchored microtubules are lost first, while the minus-end-out microtubules persist longer, become relatively more abundant and drive dispersion of PCM complexes.

Our findings may help to explain how acentrosomal MTOCs form in cells where both PCM proteins and PCM-independent microtubule stabilizers such as CAMSAPs are present. Two well studied examples of such systems are the MTOCs at the Golgi membranes and the apical cortex of epithelial cells, where CAMSAPs and PCM proteins are present simultaneously and may cooperate with each other or play redundant roles (*Goldspink et al., 2017*; *Nashchekin et al., 2016*; *Noordstra et al., 2016*; *Sanchez et al., 2021*; *Toya et al., 2016*; *Wang et al., 2015*; *Wu et al., 2016*; *Zhu and Kaverina, 2013*). We found that CAMSAP2-stabilized minus ends can exert a highly dominant effect on PCM organization, and biases in the distribution of minus ends lead to similar biases in the distribution of PCM proteins. This property helps to explain how the Golgi, which anchors CAMSAP2-stabilized microtubules, recruits pericentrin and becomes the major MTOC in cells lacking centrosomes. Furthermore, mobility of PCM clusters may contribute to centrosome disassembly after mitosis, when the effect of mitotic kinases driving PCM coalescence is abolished. This might help to explain the appearance of PCM 'packets' or 'fragments' accompanying centrosome disassembly during mitotic exit in different cell types (*Magescas et al., 2019*; *Rusan and Wadsworth, 2005*). Re-emergence of stable non-centrosomal microtubules, for example, due to post-mitotic dephosphorylation of CAMSAP proteins (*Jiang et al., 2014*), might contribute to this process.

Importantly, our experiments and simulations showed that coupling stable minus-ends to a minus-end directed motor is by itself insufficient to form a compact MTOC, but self-clustering PCM can contribute to this process, when the clusters of PCM proteins that can anchor microtubules are sufficiently abundant. Altogether, self-association of interphase PCM appears to be strong enough to promote its clustering but is sufficiently dynamic to allow PCM reorganization dependent on other microtubule regulators present in the cell.

An interesting question that remains unanswered by our work is the inhibitory role of PLK4 in interphase caMTOC formation. We did observe caMTOCs in cells depleted of PLK4, indicating that, unlike cells lacking TRIM37, which form PCM clusters containing catalytically inactive PLK4 (*Meitinger et al., 2020*; *Yeow et al., 2020*), interphase cells studied here do not rely on enzymatically inactive PLK4 for PCM assembly. PLK4 is known to phosphorylate NEDD1 (*Chi et al., 2021*), and it is possible that the lack of phosphorylation prevents this γ-TuRC-binding protein and its partners, such as CEP192, from participating in interphase caMTOC assembly. It is, of course, also possible that PLK4 phosphorylation inhibits the interactions or activities of some of the players driving caMTOC formation. The easy-to-manipulate cellular model that we have described here will allow these questions to be addressed and facilitate detailed studies of the interactions and functions of PCM components in nucleating and stabilizing interphase microtubule minus ends.

# Materials and methods

## Key resources table

| Reagent type (species) or resource | Designation | Source or reference | Identifiers | Additional information |
|---|---|---|---|---|
| Antibody | anti-Pericentrin (mouse monoclonal) | Abcam | Abcam Cat# ab28144, RRID:AB_2160664 | (1:500) for IF |
| Antibody | anti-Pericentrin (rabbit polyclonal) | Abcam | Abcam Cat# ab4448, RRID:AB_304461 | (1:500) for IF; (1:1000) for WB |
| Antibody | anti-CDK5RAP2 (rabbit polyclonal) | Bethyl Laboratories | Bethyl Cat# A300-554A, RRID:AB_477974 | (1:300) for IF; (1:1000) for WB |
| Antibody | anti-γ-tubulin (mouse monoclonal) | Sigma-Aldrich | Sigma-Aldrich: T6557; RRID:AB_477584 | (1:300) for IF; (1:2000) for WB |
| Antibody | anti-γ-tubulin (rabbit polyclonal) | Sigma-Aldrich | Sigma-Aldrich:T3559, RRID:AB_477575 | (1:300) for IF |
| Antibody | anti-NEDD1 (mouse monoclonal) | Abnova | Abnova Corporation Cat# H00121441-M05, RRID:AB_534956 | (1:300) for IF |
| Antibody | anti-NEDD1 (rabbit polyclonal) | Rockland | Rockland Cat# 109–401 C38S, RRID:AB_10893219 | (1:1000) for WB |
| Antibody | anti-Ninein (rabbit polyclonal) | BETHYL | Bethyl Cat# A301-504A, RRID:AB_999627 | (1:300) for IF; (1:2000) for WB |
| Antibody | anti-Ninein (mouse monoclonal) | Santa Cruz Biotechnology | Santa Cruz Biotechnology Cat# sc-376420, RRID:AB_11151570 | (1:300) for IF |
| Antibody | anti-Dynein HC (rabbit polyclonal) | Santa Cruz Biotechnology | Santa Cruz Biotechnology Cat# sc-9115, RRID:AB_2093483 | (1:300) for IF; (1:500) for WB |
| Antibody | anti-p150Glued (mouse monoclonal) | BD Biosciences | BD Biosciences Cat# 610473, RRID:AB_397845 | (1:100) for IF; (1:500) for WB |
| Antibody | anti-PCM1 (mouse monoclonal) | Santa Cruz Biotechnology | Santa Cruz Biotechnology Cat# sc-398365, RRID:AB_2827155 | (1:300) for IF |
| Antibody | anti-PCM1 (rabbit polyclonal) | Bethyl Laboratories | Bethyl Cat# A301-150A, RRID:AB_873100 | (1:300) for IF |
| Antibody | anti-AKAP450 (mouse monoclonal) | BD Biosciences | BD Biosciences Cat# 611518, RRID:AB_398978 | (1:500) for WB |
| Antibody | anti-CAMSAP2 (rabbit polyclonal) | Proteintech | Proteintech Cat# 17880–1-AP, RRID:AB_2068826 | (1:200) for IF; (1:1000) for WB |
| Antibody | anti-p53 (mouse monoclonal) | Santa Cruz Biotechnology | Santa Cruz Biotechnology Cat# sc-126, RRID:AB_628082 | (1:300) for IF; (1:1000) for WB |
| Antibody | anti-p53 (rabbit polyclonal) | BETHYL | Bethyl Cat# A300-248A, RRID:AB_263349 | (1:300) for IF |
| Antibody | anti-EB1 (mouse monoclonal) | BD Biosciences | BD Biosciences:610535; RRID:AB_397892 | (1:400) for IF |
| Antibody | anti-EB3 (rabbit polyclonal) | *Martin et al., 2018*; | | (1:300) for IF |
| Antibody | anti-Centrin (mouse monoclonal) | Millipore | Millipore Cat# 04–1624, RRID:AB_10563501 | (1:500) for IF |
| Antibody | anti-CEP120 (rabbit polyclonal) | Thermo Fisher Scientific | Thermo Fisher Scientific Cat# PA5-55985, RRID:AB_2639665 | (1:300) for IF |
| Antibody | anti-CEP135 (rabbit polyclonal) | Sigma-Aldrich | Sigma-Aldrich:SAB4503685; RRID:AB_10746232 | (1:300) for IF |
| Antibody | anti-CEP152 (rabbit polyclonal) | Abcam | Abcam, Cat # ab183911 | (1:300) for IF; (1:1000) for WB |
| Antibody | anti-CEP170 (mouse monoclonal) | Thermo Fisher Scientific | Thermo Fisher Scientific Cat# 41–3200, RRID:AB_2533502 | (1:200) for IF |
| Antibody | anti-CEP192 (rabbit polyclonal) | Bethyl Laboratories | Bethyl Cat# A302-324A, RRID:AB_1850234 | (1:300) for IF; (1:1000) for WB |
| Antibody | anti-GM130 (mouse monoclonal) | BD Biosciences | BD Biosciences:610823; RRID:AB_398142 | (1:300) for IF; (1:2000) for WB |
| Antibody | anti-α-tubulin YL1/2 (rat monoclonal) | Pierce | Pierce: MA1-80017; RRID:AB_2210201 | (1:300) for IF |
| Antibody | anti-α-tubulin (mouse monoclonal) | Sigma-Aldrich | Sigma-Aldrich:T5168; RRID:AB_477579 | (1:400) for IF |
| Antibody | anti-α-tubulin (rabbit monoclonal antibody) | Abcam | Abcam Cat# ab52866, RRID:AB_869989 | (1:800) for IF |
| Antibody | Anti-β-tubulin (mouse monoclonal) | Sigma-Aldrich | Sigma-Aldrich Cat# T8660, RRID:AB_477590 | (1:2000) for WB |

*Continued on next page*

*Continued*

| Reagent type (species) or resource | Designation | Source or reference | Identifiers | Additional information |
|---|---|---|---|---|
| Antibody | anti-CLASP1 (rabbit polyclonal) | *Akhmanova et al., 2001* | | (1:400) for IF |
| Antibody | anti-CLASP2 (rabbit polyclonal) | *Akhmanova et al., 2001* | | (1:400) for IF |
| Antibody | anti-CLIP-115 #2,238 (rabbit polyclonal) | *Akhmanova et al., 2001* | | (1:300) for IF |
| Antibody | anti-CLIP-170 #2,360 (rabbit polyclonal) | *Akhmanova et al., 2001* | | (1:300) for IF |
| Antibody | anti-ch-TOG (rabbit polyclonal) | *Charrasse et al., 1998* | Dr. Lynne Cassimeris (Lehigh University, USA) | (1:200) for IF |
| Antibody | anti-CPAP (rabbit polyclonal) | *Kohlmaier et al., 2009* | Dr. Pierre Gönczy (EPFL, Switzerland) | (1:200) for IF |
| Antibody | anti-CP110 (rabbit monoclonal) | Proteintech | Proteintech Cat# 12780–1-AP, RRID:AB_10638480 | (1:300) for IF |
| Antibody | anti-KIF1C (rabbit polyclonal) | Cytoskeleton | Cytoskeleton Cat# AKIN11-A, RRID:AB_10708792 | (1:300) for IF |
| Antibody | anti-KIF2A (rabbit polyclonal) | *Ganem and Compton, 2004* | Dr. Duane Compton (Geisel School of Medicine at Dartmouth, USA) | (1:300) for IF |
| Antibody | anti-HAUS2 (rabbit polyclonal) | *Lawo et al., 2009* | Dr. Laurence Pelletier (Lunenfeld-Tanenbaum Research Institute, Canada) | (1:200) for IF |
| Antibody | anti-BICD2 (rabbit polyclonal) | *Hoogenraad et al., 2003* | | (1:2500) for WB |
| Antibody | anti-Actin (mouse monoclonal) | Millipore | Millipore Cat# MAB1501, RRID:AB_2223041 | (1:4000) for WB |
| Antibody | anti-Ku80 (mouse monoclonal) | BD Biosciences | BD Biosciences Cat# 611360, RRID:AB_398882 | (1:2000) for WB |
| Antibody | anti-LaminA/C (mouse monoclonal) | BD Biosciences | BD Biosciences Cat# 612162, RRID:AB_399533 | (1:400) for IF |
| Antibody | anti-Cytochrome C (mouse monoclonal) | BD Biosciences | BD Biosciences Cat# 556432, RRID:AB_396416 | (1:300) for IF |
| Antibody | anti-Calnexin (rabbit polyclonal) | Abcam | Abcam Cat# ab22595, RRID:AB_2069006 | (1:300) for IF |
| Antibody | Anti-Lamtor4 (rabbit monoclonal) | Cell Signaling (CST)/Bioke | Cell Signaling Technology Cat# 12284, RRID:AB_2797870 | (1:800) for IF |
| Antibody | Anti-Tom20 (mouse monoclonal) | BD Biosciences | BD Biosciences Cat# 612278, RRID:AB_399595 | (1:200) for IF |
| Antibody | IRDye 800CW/680LT secondaries | Li-Cor Biosciences | LI-COR Biosciences Cat# 926–32219, RRID:AB_1850025' LI-COR Biosciences Cat# 926–68020, RRID:AB_10706161; LI-COR Biosciences Cat# 926–32211, RRID:AB_621843; LI-COR Biosciences Cat# 926–68021, RRID:AB_10706309 | (1:5000) for WB |
| Antibody | Alexa Fluor 405–, 488–, and 594–secondaries | Molecular Probes/ Thermo Fisher Scientific | Molecular Probes Cat# A-11007, RRID:AB_141374; Cat# A-11034, RRID:AB_2576217; Cat# A32723, RRID:AB_2633275; Cat# A-31553, RRID:AB_221604; Cat# A-11029, RRID:AB_138404; Cat# A-11032, RRID:AB_2534091; Cat# A-11006, RRID:AB_141373; Thermo Fisher Scientific Cat# A-11012, RRID:AB_2534079 | (1:500) for IF |
| Sequence-based reagent | siRNA against PCNT #1 | *Gavilan et al., 2018* | 5'-AAAAGCUCUGAUU UAUCAAAAGAAG-3' | |
| Sequence-based reagent | siRNA against PCNT #2 | *Gavilan et al., 2018* | 5'-UGAUUGGACGUCA UCCAAUGAGAAA-3' | |
| Sequence-based reagent | siRNA against PCNT #3 | *Tibelius et al., 2009* | 5'-GCAGCUGAGCUGAAGGAGA-3' | |
| Sequence-based reagent | siRNA against CDK5RAP2 | *Fong et al., 2008* | 5'-UGGAAGAUCUCCUAACUAA-3' | |
| Sequence-based reagent | siRNA against γ-tubulin #1 | *Lüders et al., 2006* | 5'-GGAGGACAUGUUCAAGGAA-3' | |
| Sequence-based reagent | siRNA against γ-tubulin #2 | *Vinopal et al., 2012* | 5'-CGCAUCUCUUUCUCAUAU-3' | |

*Continued on next page*

*Continued*

| Reagent type (species) or resource | Designation | Source or reference | Identifiers | Additional information |
|---|---|---|---|---|
| Sequence-based reagent | siRNA against Ninein | *Goldspink et al., 2017* | 5'-CGGUACAAUGAGUGUAGAAUU-3' | |
| Sequence-based reagent | siRNA against PCM1 | *Wang et al., 2013* | 5'-UCAGCUUCGUGAUUCUCAG-3' | |
| Sequence-based reagent | siRNA against CEP152 | *Cizmecioglu et al., 2010*; *Komarova et al., 2005* | 5'-GCGGAUCCA ACUGGAAAUCUA-3' | |
| Sequence-based reagent | siRNA against CEP120 | *Ganem et al., 2005*; *Lin et al., 2013* | 5'-AAUAUAUCUUCU UGCAUCUCCUUCC-3' | |
| Sequence-based reagent | siRNA against CEP192 | *Sonnen et al., 2013* | 5'-CAGAGGAAUCAAUAAUAAA –3' | |
| Sequence-based reagent | siRNA against NEDD1 #1 | *Lüders et al., 2006* | 5'-GCAGACAUGUGUCAAUUUA-3' | |
| Sequence-based reagent | siRNA against NEDD1 #2 | *Haren et al., 2006* | 5'-GGGCAAAAGCAGACAUGUG-3' | |
| Sequence-based reagent | siRNA against DHC #1 | *Splinter et al., 2010* | 5'-CGUACUCCCGUGAUUGAUG-3' | |
| Sequence-based reagent | siRNA against DHC #2 | *Splinter et al., 2010* | 5'-GCCAAAAGUUACAGACUUU-3' | |
| Sequence-based reagent | siRNA against CAMSAP2 | *Jiang et al., 2014* | 5'-GUACUGGAUAAAUAAGGUA-3' | |
| Sequence-based reagent | siRNA against CEP170 | *Stolz et al., 2015* | 5'-GAAGGAAUCCUCCAAGUCA-3' | |
| Sequence-based reagent | siRNA against CPAP | *Tang et al., 2009* | 5'-AGAAUUAGCUCGAAUAGAA-3' | |
| Sequence-based reagent | siRNA against CLIP170 #1 | *Lansbergen et al., 2004*; *Mimori-Kiyosue et al., 2005* | 5'-GGAGAAGCAGCAGCACAUU-3' | |
| Sequence-based reagent | siRNA against CLIP170 #2 | *Lansbergen et al., 2004*; *Mimori-Kiyosue et al., 2005* | 5'-UGAAGAUGUCAGGAGAUAA-3' | |
| Sequence-based reagent | siRNA against CLIP115 #1 | *Lansbergen et al., 2004* | 5'-GGCACAGCAUGAGCAGUAU-3' | |
| Sequence-based reagent | siRNA against CLIP115 #2 | *Lansbergen et al., 2004* | 5'-CUGGAAAUCCAAGCUGGAC-3' | |
| Sequence-based reagent | siRNA against ch-TOG: | *Cassimeris and Morabito, 2004*; *Lansbergen et al., 2004* | 5'-GAGCCCAGAGUGGUCCAAA-3' | |
| Sequence-based reagent | siRNA against EB1 | *Grigoriev et al., 2008*; *Lansbergen et al., 2004* | 5'-AUUCCAAGCUAAGCUAGAA-3' | |
| Sequence-based reagent | siRNA against EB3 | *Cassimeris and Morabito, 2004*; *Komarova et al., 2005* | 5'-CUAUGAUGGAAAGGAUUAC-3' | |
| Sequence-based reagent | siRNA against KIF2A | *Ganem et al., 2005*; *Grigoriev et al., 2008* | 5'-GGCAAAGAGAUUGACCUGG-3' | |
| Sequence-based reagent | siRNA against CP110 | *Cizmecioglu et al., 2010*; *Spektor et al., 2007* | 5'-AAGCAGCAUGAGUAUGCCAGU-3' | |
| Sequence-based reagent | siRNA against Luciferase | *Lansbergen et al., 2004*; *Lin et al., 2013* | 5'-CGUACGCGGAAUACUUCGA-3' | |
| Sequence-based reagent | sgRNA target CAMSAP2 | *Lansbergen et al., 2004*; *Wu et al., 2016* | 5'-gCATGATCGATACCCTCATGA-3 | |
| Sequence-based reagent | sgRNA target p53 e2 #1 | This study | 5'-gCGTCGAGCCCCCTCTGAGTC-3'; | |

*Continued on next page*

*Continued*

| Reagent type (species) or resource | Designation | Source or reference | Identifiers | Additional information |
|---|---|---|---|---|
| Sequence-based reagent | sgRNA target p53 e4 #2 | This study | 5'-gCCATTGTTCAATATCGTCCG-3'; | |
| Sequence-based reagent | sgRNA target PCNT e5-1 #1 | This study | 5'-gAGACGGCATTGACGGAGCTG-3'; | |
| Sequence-based reagent | sgRNA target PCNT e5-2 #2 | This study | 5'-GCTCAACAGCCGGCGTGCCC-3'; | |
| Sequence-based reagent | p53 KO sequencing primer F | This study | 5'-TCAGACACTGGCATGGTGTT-3'; | |
| Sequence-based reagent | p53 KO sequencing primer R | This study | 5'-AGAAATGCAGGGGGGATACGG-3'; | |
| Sequence-based reagent | PCNT KO sequencing primer F | This study | 5'-ATACAGCGAGGGAATTCGGG-3'; | |
| Sequence-based reagent | PCNT KO sequencing primer R | This study | 5'-TAGAATGCCCACACCGAGC-3'; | |
| Sequence-based reagent | Forward primer for PCR of tagBFP to generate pB80-FRB-TagBFP-GCN4-ppKin14 | This study | 5'-TCTCAAAGCAATTGT CGACAGGATCCGC TGGCTCCGCTGCTG GTTCTGGCGAATTCA GCGAGCTGATTA AGGAGAACA-3'; | |
| Sequence-based reagent | Reverse primer for PCR of tagBFP to generate pB80-FRB-TagBFP-GCN4-ppKin14 | This study | 5'-ATAGCGGAGCC TGCTTTTTTGTACA CATTAAGCTTGTG CCCCAGTTTG-3'; | |
| Sequence-based reagent | Forward primer for PCR of tagBFP to generate pB80-FRB-HA-GCN4-ppKin14 | This study | 5'-TCTCAAAGCAAT TGTCGACATACCCATA CGATGTTCCAGAT TACGCTGTGTAC AAAAAAGCAGGCTCC-3'; | |
| Sequence-based reagent | Reverse primer for PCR of tagBFP to generate pB80-FRB-HA-GCN4-ppKin14 | This study | 5'-GGAGCCTGCTTT TTTGTACACAG CGTAATCTGG AACATCGTATGG GTATGTCGACAA TTGCTTTGAGA-3'; | |
| Chemical compound | Centrinone B | Tocris Bioscience | Tocris Bioscience Cat # 5,690 | 125 nM |
| Chemical compound | Nocodazole | Sigma-Aldrich | Sigma-Aldrich, Cat # M1404-10MG | 10 µM |
| Chemical compound | Rapalog (A/C Heterodimerizer) | Takara | Takara, Cat # 635,056 | 50 nM (fixation), 100 nM (live imaging). |
| Chemical compound | Rapalog (B/B Homodimerizer) | Takara | Takara, Cat # 635,060 | 500 nM (fixation) |
| Chemical compound | Dynapyrazole A | Sigma-Aldrich | Sigma-Aldrich, Cat # SML2127-25MG | 5 µM |
| Chemical compound | BI2536 | Selleckchem | Selleckchem, Cat # S1109 | 500 nM |
| Chemical compound | Thymidine | Sigma-Aldrich | Sigma-Aldrich, Cat # T9250-25G | 5 mM |
| Chemical compound | proTAME | Boston Biochem | Boston Biochem, Cat # I-440 | 5 µM |
| Chemical compound | Brefeldin A | Peptrotech | Peptrotech, Cat # 2031560 | 5 µg/ml |
| Chemical compound | SiR-tubulin | Tebu-bio | Tebu-bio, Cat # SC002 | 100 nM |
| Software, algorithm | ImageJ radiality plugin | *Katrukha, 2019*; https://github.com/ ekatrukha/radialitymap | | |
| Recombinant DNA reagent | pLVX-IRES-puro (plasmid) | Clontech | | |

*Continued on next page*

*Continued*

| Reagent type (species) or resource | Designation | Source or reference | Identifiers | Additional information |
|---|---|---|---|---|
| Recombinant DNA reagent | pLVX-GFP-CDK5RAP2-IRES-puro (plasmid) | This work | | |
| Recombinant DNA reagent | pB80-FRB-TagBFP-GCN4-ppKin14 (plasmid) | This work | | |
| Recombinant DNA reagent | pB80-FRB-GFP-GCN4-ppKin14 (plasmid) | This work | | |
| Recombinant DNA reagent | pB80-FRB-HA-GCN4-ppKin14 (plasmid) | This work | | |
| Recombinant DNA reagent | 2FKBP-mCherry-CAMSAP2 (plasmid) | This work | | |
| Recombinant DNA reagent | GFP-PCNT (plasmid) | This work | | |
| Recombinant DNA reagent | 2homoFKBP-mCherry-PCNT (plasmid) | This work | | |
| Recombinant DNA reagent | GST-DmKHC(1-421)-mNeonGreen (plasmid) | This work | | |
| Cell line (*Homo sapiens*) | hTERT-RPE-1 | ATCC | CRL-4000 | |
| Cell line (*Homo sapiens*) | hTERT-RPE-1 AKAP450 knockout | *Wu et al., 2016* | | |
| Cell line (*Homo sapiens*) | hTERT-RPE-1 AKAP450/ CAMSAP2 knockout | *Wu et al., 2016* | | |
| Cell line (*Homo sapiens*) | hTERT-RPE-1 AKAP450/CAMSAP2/p53 knockout | This work | | |
| Cell line (*Homo sapiens*) | hTERT-RPE-1 AKAP450/CAMSAP2/ p53/Pericentrin knockout | This work | | |
| Cell line (*Homo sapiens*) | hTERT-RPE-1 AKAP450/ CAMSAP2/EB1/EB3 mutant | This work | | |
| Cell line (*Homo sapiens*) | hTERT-RPE-1 AKAP450/CAMSAP2/ CDK5RAP2/MMG/ p53/Pericentrin knockout | This work | | |
| Cell line (*Homo sapiens*) | HEK 293T | ATCC | CRL-11268 | |

## DNA constructs and protein purification

To generate the lentiviral vector pLVX-GFP-CDK5RAP2-IRES-Puro, pLVX-IRES-Puro plasmid (Clontech) was digested with AgeI and NotI (FastDigest, Thermo Fisher), and then Gibson Assembly (NEB) was performed with gel-purified PCR product of GFP-CDK5RAP2 (*Wu et al., 2016*). To generate pB80-FRB-TagBFP-GCN4-ppKin14 and pB80-FRB-HA-GCN4-ppKin14, pB80-FRB-GFP-GCN4-ppKin14-Vlb was digested with XbaI and BsrGI (FastDigest, Thermo Fisher), and then TagBFP and HA-tag encoding DNA fragments were subcloned into the linearized vector by Gibson Assembly. To generate 2FKBP-mCherry-CAMSAP2, a CAMSAP2-encoding DNA fragment was subcloned into a vector containing 2FKBP-mCherry digested by SalI and BamHI. To generate 2homoFKBP-mCherry-Pericentrin, a fragment contains two repeats of homodimerizing version of FKBP (homoFKBP) and mCherry was subcloned into the vector containing pericentrin fragment after digestion with NheI and NotI.

To generate the PX459 with single guide RNA (sgRNA) sequences, pSpCas9(BB)–2A-Puro (PX459) V2.0 (*Ran et al., 2013*; purchased from Addgene) was digested with FastDigest BbsI (Thermo Fisher), and the annealing product of single-strand sgRNA-encoding oligonucleotides was inserted into the linear PX459 linear vector by T4 ligation (Thermo Fisher). The sgRNA sequences that were used in this study are: sgRNA targeting AKAP450 5'- gAGGGTTACCTATGGGACTGA –3'; sgRNA targeting

CAMSAP2 encoding gene 5'-gCATGATCGATACCCTCATGA-3'; sgRNA targeting p53-encoding gene exon 2 #1 5'-gCGTCGAGCCCCCTCTGAGTC-3'; sgRNA targeting p53 exon 4 #2 5'-gCCATTGT TCAATATCGTCCG-3'; sgRNA targeting pericentrin exon 5 #1 5'-gAGACGGCATTGACGGAGCTG-3'; sgRNA targeting pericentrin-encoding gene exon 5 #2 5'-GCTCAACAGCCGGCGTGCCC-3'.

To generate the GST-DmKHC(1-421)-mNeonGreen construct used for protein purification for motor-PAINT, the fragment containing amino acids 1–421 of the *Drosophila melanogaster* Kinesin Heavy Chain (DmKHC) was amplified from donor construct DmKHC(1-421)-GFP-6x-His with a C-terminal mNeonGreen tag by PCR and then cloned into a pGEX vector. The plasmid was transformed into *E. coli* BL21 cells for purification. Bacteria were cultured until OD600 ≈0.7 and cultures were cooled prior to inducing protein expression with 0.15 mM IPTG at 18 °C overnight. Cells were then pelleted by centrifugation, snap frozen in liquid nitrogen, and stored at –80 °C until use. Cells were rapidly thawed at 37 °C before being resuspended in chilled lysis buffer (phosphate buffered saline (PBS) supplemented with 5 mM $MgCl_2$, 0.5 mM ATP, 0.1% Tween 20, 250 mM NaCl, and 1 x complete protease inhibitor; pH 7.4). Bacteria were lysed by sonication (5 rounds of 30 s) and supplemented with 5 mM DTT and 2 mg/mL lysozyme and then incubated on ice for 45 min. The lysate was clarified by centrifuging at 26,000 xg for 30 min before being incubated with equilibrated Glutathione Sepharose 4B resin for 1.75 hrs. Beads were then pelleted, resuspended in wash buffer (PBS supplemented with 5 mM $MgCl_2$, 0.1% Tween 20, 250 mM NaCl, 1 mM DTT, and 0.5 mM ATP; pH 7.4), and transferred to a BioRad column. Once settled, the resin was washed with 2 × 10 column volumes (CV) wash buffer, followed by 1 × 10 CV PreScission buffer (50 mM Tris-HCl, 5 mM $MgCl_2$, 100 mM NaCl, 1 mM DTT, 0.5 mM ATP; pH 8.0). The resin was then incubated overnight in 4CV PreScission buffer with 80 U PreScission protease to cleave off the GST tag. The following morning, after allowing the resin to settle, the eluent was collected, concentrated by spinning through a 3000 kDa MWCO filter, supplemented with an additional 0.1 mM ATP, 1 mM DTT, and 20% w/v sucrose before flash freezing in liquid nitrogen, and finally stored at –80 °C. Concentration was determined using a Nanodrop. All steps from lysis onwards were performed at 4 °C.

Cell culture and drug treatment hTERT immortalized RPE-1 (RPE1) cell lines were grown in an 1:1 mix of DMEM and F-10 (Lonza) and Human Embryonic Kidney (HEK) 293T cells line were cultured in DMEM, both supplemented with 10% fetal bovine serum (FBS, GE Healthcare) and 1% penicillin and streptomycin (Sigma-Aldrich). All cells were grown in tissue culture polystyrene flasks (Corning) and were maintained in a humidified incubator at 37 °C with 5% $CO_2$. Mycoplasma contamination was routinely checked with LT07-518 Mycoalert assay (Lonza).

FuGENE 6 (Promega) was used to transfect RPE1 cells with plasmids for generating CRISPR/Cas9 knockouts, immunofluorescence staining and live cell imaging; RNAiMAX (Thermo Fisher Scientific) was used to transfect RPE1 cells with siRNAs at 20 nM; MaxPEI was used to transfect HEK293T cells for lentivirus packaging. Transfections were performed according to the manufacturer's instructions within the recommended reagent/DNA or reagent/siRNA ratio range.

We used the following drugs: centrinone B (Tocris Bioscience), nocodazole (Sigma), rapalogs (A/C heterodimerizer and B/B homodimerizer, Takara), dynapyrazole A (Sigma-Aldrich), BI2536 (Selleckchem), Thymidine (Sigma-Aldrich), proTAME (Boston Biochem), and Brefeldin A (Peptrotech).

To remove centrioles, RPE1 cells were treated with 125 nM centrinone B containing complete medium for ~10 days, and drug-containing medium was refreshed every 24 hr; cell confluence was maintained around ~50–80% during the treatment.

For the microtubule disassembly and regrowth assay, the acentriolar RPE1 cells were seeded onto coverslips in 24-well plates and incubated for 24 hr, then cells were treated with 10 µM nocodazole for 1 hr in an incubator (37 °C, 5% $CO_2$) and followed by another 1 hr treatment at 4 °C to achieve complete disassembly of stable microtubule fragments. Nocodazole washout was then carried out by at least six washes on ice with ice-cold complete medium; subsequently, plates were moved to a 37 °C water bath and pre-warmed medium was added to each well to allow microtubule regrowth.

For cell cycle synchronization, centrinone-treated AKAP450/CAMSAP2/P53 knockout cells were treated with 5 mM Thymidine (Sigma-Aldrich) overnight, released in centrinone containing medium for 4 hr and subsequently treated with 5 µM proTAME (Boston Biochem, I-440) for 2 hr before being released in centrinone containing medium for 1–4 hr followed by live imaging and fixation.

For the inducible ppKin14-CAMSAP2 heterodimerization experiment, acentriolar cells were seeded onto coverslips in 24-well plates, cultured with centrinone B containing medium and co-transfected

with 2FKBP-mCherry-CAMSAP2 and FRB-TagBFP-GCN4-ppKin14 vectors. Twenty-four hr after transfection, rapalog AP21967 A/C heterodimerizer was added into the medium at a final concentration of 50 nM and incubated overnight for preparation of fixed cells. For live imaging, heterodimerizer was used at 100 nM.

For the inducible pericentrin homodimerization experiment, acentriolar AKAP450 KO cells were seeded onto coverslips in 24-well plates, cultured with centrinone B containing medium and transfected with 2homoFKBP-mCherry-pericentrin vector. Twenty-four hr after transfection, rapamycin analogue AP20187 (B/B homodimerizer) was added into the medium at a final concentration of 500 nM and incubated 2 hr before fixation.

## Lentivirus packaging and generation of transgenic stable cell lines

Lentiviruses were produced by MaxPEI-based co-transfection of HEK293T cells with the transfer vectors together with the packaging vector psPAX2 and envelope vector pMD2.G (psPAX2 and pMD2.G were a gift from Didier Trono, Addgene plasmid #12,259 and #12260; RRID:Addgene_12259 and RRID:Addgene_12260). Supernatant of packaging cells was harvested 48–72 hr after transfection, filtered through a 0.45 µm filter, incubated with a polyethylene glycol (PEG)–6000-based precipitation solution overnight at 4 °C and centrifuged for 30 min at 1500 rpm to concentrate the virus. Lentiviral pellet was resuspended in PBS.

Wild type, AKAP450 and AKAP450/CAMSAP2 knockout RPE1 cells were infected with lentivirus and incubated in complete medium supplemented with 8 µg/ml polybrene (Sigma-Aldrich). After 24 hr, the cell medium was replaced with fresh medium. Starting 72 hr after viral transduction, cells were subjected to selection with puromycin at a concentration of 25 µg/ml for wild-type, 20 µg/ml for AKAP450 knockout and 15 µg/ml for AKAP450/CASMAP2 knockout for up to 3 days (until most of the untransduced control cells, treated with the same concentration of antibiotic, were dead). After selection, cells were grown in normal medium for 3 days and individual colonies expressing GFP were isolated into 96-well plates by fluorescence-activated cell sorting (FACS). Sorted single transgenic stable cell lines were further confirmed by immunofluorescence staining to check the expression level of GFP-CDK5RAP2 and its colocalization with other centrosomal proteins.

## Generation of CRIPSR/Cas9 knockout cell lines

The CRISPR/Cas9-mediated knockout of p53-, pericentrin-, AKAP450- and CAMSAP2-encoding genes was performed as described previously (*Ran et al., 2013*). In brief, AKAP450/CAMSAP2 knockout RPE1 cells (*Wu et al., 2016*) were transfected with the vectors bearing the appropriate targeting sequences using FuGENE 6. One day after transfection, the transfected AKAP450/CAMSAP2 knockout RPE1 cells were subjected to selection with 15 µg/ml puromycin for up to 3 days. After selection, cells were allowed to recover in normal medium for ~7 days, and knockout efficiency was checked by immunofluorescence staining. Depending on the efficiency, 50–500 individual clones were isolated and confirmed by immunofluorescence staining, and the resulted single colonies were characterized by Western blotting, immunostaining and genome sequencing. AKAP450/CAMSAP2/p53 and AKAP450/CAMSAP2/MMG/CDK5RAP2/p53 knockout cell lines were generated first and subsequently, each of them was used to knock out the gene encoding pericentrin. The mutated portions of the p53- and pericentrin-encoding genes were sequenced using gel-purified PCR products obtained with primers located in the vicinity of the corresponding sgRNA targeting sites.

## Antibodies, immunofluorescence staining, and western blotting

Antibodies used for immunostaining and Western blotting are listed in the Key Reagent or Resource table. For immunofluorescence cell staining, cultured cells were fixed with –20 °C methanol for 5 min or with 4% paraformaldehyde (PFA) for 12 min at room temperature, rinsed in PBS for 5 min, permeabilized with 0.15% Triton X-100 in PBS for 2 min, washed 3 times for 5 min with 0.05% Tween-20 in PBS, sequentially incubated for 20 min in the blocking buffer (2% BSA and 0.05% Tween-20 in PBS), 1 hr with primary antibodies in the blocking buffer, washed 3 times for 5 min with 0.05% Tween-20 in PBS, then for 1 hr in secondary antibodies in the blocking buffer, washed 3 times for 5 min with 0.05% Tween-20 in PBS, and air-dried after a quick wash in 96% ethanol. Cells were mounted in Vectashield mounting medium with or without DAPI (Vector laboratories, Burlingame, CA). Alexa Fluor

−405,−488, −594 and −647 conjugated goat antibodies against rabbit, rat and mouse IgG were used as secondary antibodies (Molecular Probes, Eugene, OR).

For Western blotting, cells were harvested from six-well plates or 10 cm dishes at 90% confluence and protein extracts were prepared using the lysis buffer containing 20 mM Tris-Cl, pH 7.5, 150 mM NaCl, 0.5 mM EDTA, 1 mM DTT, 1% Triton X-100 or RIPA buffer containing 50 mM Tris-HCl, pH 7.5, 150 mM NaCl, 1% Triton X-100, 0.5% Sodium Deoxycholate supplemented with protease inhibitor and phosphatase inhibitors (Roche). Samples were run on polyacrylamide gels, followed by transfer on 0.45 µm nitrocellulose membrane (Sigma-Aldrich). Blocking was performed in 2% BSA in PBS for 30 min at room temperature. The membrane was first incubated with the primary antibodies overnight at 4 °C and washed with 0.05% Tween-20 in PBS 3 times and subsequently incubated with secondary antibodies for 1 hr at room temperature and washed 3 times with 0.05% Tween-20 in PBS. IRDye 800CW/680 LT Goat anti-rabbit and anti-mouse were used as secondary antibodies (Li-Cor Biosciences, Lincoln, LE) and membranes were imaged on Odyssey CLx infrared imaging system (Image Studio version 5.2.5, Li-Cor Biosciences).

## Imaging and analysis of fixed cells

Images of fixed cells were collected with a Nikon Eclipse Ni upright fluorescence microscope equipped with a DS-Qi2 CMOS camera (Nikon), an Intensilight C-HGFI epi-fluorescence illuminator (Nikon), Plan Apo Lambda 100×NA 1.45 or Plan Apo Lambda 60 x N.A. 1.40 oil objectives (Nikon) and driven by NIS-Elements Br software (Nikon).

Gated STED imaging was performed with Leica TCS SP8 STED 3 X microscope driven by LAS X software using HC PL APO 100 x/1.4 oil STED WHITE objective, white laser (633 nm) for excitation and 775 nm pulsed lased for depletion. Images were acquired in 2D STED mode with vortex phase mask. Depletion laser power was equal to 90% of maximum power and an internal Leica HyD hybrid detector with a time gate of 1≤tg ≤ 8 ns was used.

ImageJ was used for adjustments of intensity levels and contrast, quantification of the immunofluorescence signal intensity and maximum intensity projections. To analyze PCM clustering after nocodazole washout in AKAP450/CAMSAP2 knockout RPE1 cells, images were separated into concentric circular areas using Concentric Circles plugin of ImageJ. The biggest PCM cluster (which normally also had the highest fluorescence intensity) was selected as the center, around which 20 circles with 2 µm inner radius and 20 µm outer radius were drawn. Fluorescence intensity of PCM clusters in these concentric circles was measured automatically and normalized by the sum of the total PCM intensity in each cell per condition. To quantify the areas occupied by PCM clusters, immunofluorescence images of fixed cells and time lapse images of live cells were analyzed by drawing the smallest circle that covered visible PCM clusters to indicate the area occupied by the PCM clusters, and the diameters of the circles were used for the quantification.

## Measurements of microtubule radiality

To analyze microtubule radiality, images of fluorescently labeled microtubules were separated into radial and non-radial components using a customized ImageJ macro (https://github.com/ekatrukha/radialitymap; *Katrukha, 2019*). First, a local orientation angle map was calculated for each pixel using the OrientationJ plugin. We used 'cubic spline gradient' method and tensor sigma parameter of 6 pixels (0.4 µm). The new origin of coordinates was specified by selecting the centrosome position in the corresponding channel, or the brightest spot in case of centrinone treatment. Radial local orientation angle was calculated as a difference between the local orientation angle and the angle of the vector drawn from the new origin of coordinates to the current pixel position. A radial map image was calculated as an absolute value of the cosine of the radial local orientation angle at each pixel providing values between zero and one. A non-radial map image was calculated as one minus the radial map. Both maps were multiplied with the original image to account for different signal intensities; the two maps illustrate separated radial and non-radial image components.

## Live cell imaging and analysis

Live fluorescent imaging was performed with spinning disk confocal microscopy on inverted research microscope Nikon EclipseTi-E (Nikon), equipped with the Perfect Focus System (Nikon), Nikon Plan Apo VC 60 x NA 1.4 and Nikon Plan Apo VC 100 x N.A. 1.40 oil objectives (Nikon) and spinning-disc

confocal scanner unit (CSU-X1-A1, Yokogawa). The system was also equipped with ASI motorized stage with the piezo top plate MS-2000-XYZ (ASI), Photometrics Evolve 512 EMCCD camera (Photometrics) and controlled by the MetaMorph 7.8 software (Molecular Devices). Vortran Stradus lasers (405 nm 100 mW, 488 nm 150 mW and 642 nm 165 mW) and Cobolt Jive 561 nm 110 mW laser were used as the light sources. System was equipped with ET-DAPI (49000), ET-GFP (49002), ET-mCherry (49008) and ET-Cy5 (49006) filter sets (Chroma). 16-bit images were projected onto the EMCCD chip with the intermediate lens 2.0 X (Edmund Optics) at a magnification of 110 nm per pixel (60 x objective) and 67 nm per pixel (100 x objective). To keep cells at 37 °C and 5% $CO_2$ we used stage top incubator (INUBG2E-ZILCS, Tokai Hit). Cells were plated on round 25 mm coverslips, which were mounted in Attofluor Cell Chamber (Thermo fisher). Cells were imaged with a 2 s interval and 200ms exposure for 1–3 hr at 10% laser power.

Phase-contrast live cell imaging was performed on a Nikon Ti equipped with a perfect focus system (Nikon), a super high pressure mercury lamp (C-SHG1, Nikon, Japan), a Plan Apo 60 x NA 1.4 (Ph3), a CoolSNAP HQ2 CCD camera (Photometrics, Tucson, AZ), a motorized stage MS-2000-XYZ with Piezo Top Plate (ASI, Eugene, OR) and a stage top incubator (Tokai Hit, Japan) for 37 °C/5% $CO_2$ incubation. The microscope setup was controlled by Micro-manager software. Cells were plated on round 25 mm coverslips, which were mounted in Attofluor Cell Chamber (Thermo fisher), and imaged with a 1 min interval for ~24 hr.

For live imaging of nocodazole treatment and washout experiments, cells were incubated with the medium containing 100 nM SiR-tubulin (Tebu-bio) overnight to image the microtubule network. Centrinone-treated cells were imaged for a desired period of time prior to the nocodazole treatment, and then nocodazole was added into the medium at a final concentration of 10 μM while imaging simultaneously. Culture medium was carefully removed when microtubules were completely depolymerized and washed with prewarmed medium six times to let microtubules regrow. GFP-CDK5RAP2 and SiR-tubulin imaging was performed with a 2 s interval with 200ms exposure for 1–3 hr in total, and maximum intensity projections, contrast adjustment and further processing was performed using ImageJ.

## FRAP

FRAP experiments were performed on the spinning disc microscope describe above, equipped with iLas platform and using Targeted Laser Action options of iLas and controlled with iLas software (Roper Scientific, now Gataca Systems). Photobleaching in the GFP channel was performed with the 488 nm laser. For the FRAP analysis, Polygon ROIs were set in photobleached and non-bleached regions as well as in the background. The average fluorescence intensity was measured using ImageJ for each frame, the background intensity was subtracted from the bleached and non-bleached areas and normalized to the average of the frames acquired prior to the bleach. The mean fluorescence intensities of the images before photobleaching were set as 100%, and the subsequent relative recovery percentages were calculated. Time lapse acquisitions were corrected for drift with the ImageJ plugins Template Matching.

## motor-PAINT and analysis

For motor-PAINT, a protocol published previously (*Tas et al., 2017*) was used, with minor adjustments. Cells were incubated with 50 nM SiR-tubulin and 500 nM verapamil overnight to allow fields of view suitable for imaging to be located before the addition of purified GST-DmKHC(1-421)-mNeonGreen. For nocodazole-treated samples, cells were first incubated with 10 μM nocodazole for 15 min at 37 °C. A single nocodazole-treated or control sample was then transferred to an imaging chamber, and cells were subjected to extraction for 1 min in extraction buffer (BRB80: 80 mM K-Pipes, 1 mM $MgCl_2$, 1 mM EGTA; pH 6.8, supplemented with 1 M sucrose and 0.15% TritonX-100) pre-warmed to 37 °C. Pre-warmed fixation buffer (BRB80 supplemented with 2% PFA) was added to this (i.e. final PFA concentration of 1%) and the solutions were mixed by gentle pipetting for 1 min. This buffer was removed and the chamber was washed for 4 times for 1 min in pre-warmed wash buffer (BRB80 supplemented with 1 μM Taxol) before adding imaging buffer (BRB80 supplemented with 583 μg/mL catalase, 42 μg/mL glucose oxidase, 1.7% w/v glucose, 1 mM DTT, 1 μM Taxol, and 5 mM ATP). An aliquot of GST-DmKHC(1-421)-mNeonGreen motors was warmed, spun in the Airfuge at 20 psi for

5 min in a pre-chilled rotor to remove any aggregates, and then transferred to a clean tube prior to use. Motors were kept on ice and added locally to cells in 0.3 µl increments.

Imaging was performed immediately after sample preparation at room temperature on a Nikon Ti-E microscope equipped with a 100 x Apo TIRF oil immersion objective (NA. 1.49) and Perfect Focus System 3 (Nikon). Excitation was achieved with a Lighthub-6 laser combiner (Omicron) containing a 647 nm laser (LuxX 140 mW, Omicron), a 488 nm laser (LuxX 200 mW, Omicron), and optics allowing for a tunable angle of incidence. Illumination was adjusted for (pseudo-) total internal reflection fluorescence (TIRF) microscopy. Emission light was separated from excitation light using a quad-band polychroic mirror (ZT405/488/561/640rpc, Chroma), a quad-band emission filter (ZET405/488/561/640 m, Chroma), and an additional single-band emission filter (ET525/50 m for mNeonGreen emission, Chroma). Detection was achieved using a Hamamatsu Flash 4.0v2 sCMOS camera. Image stacks were acquired with a 60ms exposure time, 7% laser power, and 15000–22000 images per field of view. Components were controlled using MicroManager (*Edelstein et al., 2014*).

Acquired stacks were pre-processed using the Faster Temporal Median ImageJ plugin (https://github.com/HohlbeinLab/FTM2; *Jabermoradi et al., 2021*) with a window size of 100 frames. These stacks were then analyzed using Detection of Molecules (DoM) plugin v.1.2.1 for ImageJ (https://github.com/ekatrukha/DoM_Utrecht), as has been described previously (*Chazeau et al., 2016*; *Tas et al., 2017*). Each image in an acquired stack is convoluted with a two-dimensional Mexican hat kernel. The resulting intensity histogram is used to create a thresholded mask based on a cut-off of three standard deviations above the mean. This mask is then subject to rounds of dilution and erosion to create a filtered mask used to calculate the centroids on the original image. These centroids are used as initial values to perform unweighted nonlinear least squares fitting with a Levenberg-Marquardt algorithm to an asymmetric two-dimensional Gaussian point spread function (PSF), allowing for the sub-pixel localization of particles.

Images were drift-corrected using DoM. The normalized cross-correlation between intermediate reconstructions of consecutive sub-stacks is used to calculate the drift in x and y between sub-stacks, which is then linearly interpolated to adjust each individual frame in the stack.

Detected particles were linked into tracks again using DoM, which performs a quicker variant of a nearest neighbor search, with a maximum distance of 5 pixels (~320 nm) between consecutive frames and no permitted frame gap. Tracks were later filtered to remove those shorter than 4 frames or longer than 200 frames, those in which an angle between parts of the trajectory exceeded 90 degrees, and those in which the speed of the motor was less than 100 nm/s or more than 1500 nm/s.

The particle table was then split into four particle tables corresponding to the four quadrants of the image with tracks sorted based on their net displacement (i.e., $\Delta x>0 \wedge \Delta y>0$; $\Delta x>0 \wedge \Delta y<0$; $\Delta x<0 \wedge \Delta y>0$; $\Delta x<0 \wedge \Delta y<0$), as described previously (*Tas et al., 2017*). These directionality-filtered particle tables were reconstructed using DoM, creating four super-resolved images of microtubule segments pointing in a similar direction. These were merged with the reconstructed image of all localizations to determine the direction of each microtubule segment. Each microtubule was manually assessed to assign it as being plus-end-in or plus-end-out. Microtubules were manually traced with lines 4 pixels (80 nm) wide, assigned a color based on their orientation, flattened onto the image, filtered with a Gaussian Blur of radius 2, and finally merged with the reconstructed image of all localizations.

To quantify the percentage of minus-end-out microtubule length to total microtubule length before and after nocodazole treatment, the length of each microtubule (determined from kinesin-1 trajectories) in the cell was measured by calculating the Euclidean distance between all subsequent pairs of points along the microtubule and summed. The ratio was calculated as the total minus-end-out microtubule length divided by the total microtubule length.

## Analysis of PCM cluster dynamics

To represent the motion of PCM clusters during nocodazole treatment, ImageJ plugin KymoResliceWide v.0.4 (https://github.com/ekatrukha/KymoResliceWide; *Katrukha, 2020*) was used for generating kymographs from the time lapse images. The velocity of PCM clusters was measured manually using kymographs starting from the time point when a small PCM cluster moved out of a caMTOC. Microtubule density around each PCM cluster was determined by measuring the mean fluorescence intensity of SiR-tubulin in a circular area with a 2 µm radius centered on the PCM cluster and normalizing it to the mean fluorescence intensity of 20 images prior to nocodazole addition (set as 100%).

The moment when a PCM cluster started to move out of the caMTOC was set as the initial time point (0 min), and the subsequent PCM cluster motion velocity and the relative local microtubule density at 43 time points were calculated and averaged.

The movement trajectories of PCM clusters were generated using ImageJ plugin TrackMate (version is 6.0.2). The parameters and the settings used were as following: LoG (Laplacian of Gaussian) detector with estimated blob diameter: 14.9 μm; thresholding value 12.25; sub-pixel localization was selected. HyperStack Displayer was selected to overlay the spots and tracks on the current hyperstack window. Simple LAP tracker was selected to track the distance and time with the linking max distance: 32.0 μm, gap-closing max distance: 55.0 μm and gap-closing max frame gap: 2. All other parameters and settings were used as the default.

## Computer simulations and analysis

Simulations were performed with Cytosim (version June 2019). Cytosim solves a set of Langevin-equations that describe the movement of flexible cytoskeletal filaments and associated proteins, such as molecular motors (*Nedelec and Foethke, 2007*). The numerical values for the parameters are given in *Table 1*. The configuration file is provided as *Supplementary file 1*.

We defined the following components in the simulation:

Cell shape: We considered a two-dimensional system with a circular cell with a radius of 10 μm. As commonly used in Cytosim simulations, we set the intracellular viscosity to 1 pN s/ μm². 

Molecular motors: The binding process of a molecular motor to a microtubule was described by a binding rate $k_{on}$ and the unbinding from the microtubule by a force-dependent unbinding rate $k_{off} = k^0_{off} \exp(F/F_d)$. When a motor was engaged with the microtubule it moved along the microtubule with a linear force-dependent velocity, characterized by $v(F) = v_0 (1 F/F_s)$. Dynein, as well as kinesin-14 motors moved to the minus end of microtubules.

Microtubule filaments: We used a classical model for microtubule dynamics which is described by a catastrophe rate, a growth speed, and a shrinkage speed. The growth speed is force-dependent with a characteristic growing force. For simplicity, we ignored rescue events. The catastrophe rate was set as $k_{cat} = v_g/L_{MT}$, in which the mean microtubule length $L_{MT}$ was 5 μm. To further restrict the microtubule length, we set a maximum of 7.5 μm. This limitation was necessary to avoid that long microtubules were pushing the minus ends to the periphery.

PCM complexes: We described a PCM complex as a bead with a radius of 50 nm. We randomly placed one microtubule nucleation site and one dynein on the bead. To effectively account for an unspecific adhesive interaction between PCM complexes, we introduced two molecules that can bind to each other. One was implemented as a 10 nm Cytosim fiber and the binding partner as a Cytosim hand with a binding range of 100 nm, binding rate of 10 s⁻¹, force-free unbinding rate of 0.01 s⁻¹, and characteristic unbinding force of 3 pN. We randomly placed one of each molecule on a PCM complex. In the simulations with strong adhesive interactions, we increased the number of adhesive binding molecules on the beads and kept all the other parameters the same. We defined five random attachment points on the beads and placed to each point five molecules. In this setup each PCM complex was covered with 25 Cytosim binding filaments and 25 Cytosim binding hands. Therefore, when two PCM complexes were close to each they formed multiple bonds between each other.

CAMSAP-kin14 complexes: We described a complex consisting of a CAMSAP-stabilized microtubule end with kinesin 14 motors attached as a Cytosim bead with a radius of 50 nm. We attached five kin14 motors and one microtubule nucleation site randomly on the bead. When we implemented adhesive interaction between CAMSAP-kin14 complexes, we used exactly the same binding molecules and arrangements as used for the PCM complexes.

Steric interactions: In all simulations, we considered steric interaction between the beads which either describe the PCM complex or the CAMSAP-kin14 complexes. All other steric interactions, except with the cell boundary were ignored.

Simulations and data analysis: We set the time step for the simulation to 0.01 s and simulated for a total of 30 min, after which a definite steady state was reached. For each configuration, we run 10 simulations and analyzed them afterward to obtain statistics on the emerging structures. From the last frame of the simulation, we obtained all positions of the complexes and calculated the mean position, which defines the center of mass. We subtracted the center of mass from all positions and derived all distances of the complexes to the center of mass. For a few examples, we determined the empirical

cumulative distribution of the distances to the center of mass. We used the standard NumPy functions to determine the mean and standard deviation of the distances to the center of mass for each simulation.

## Statistical analysis
All statistical analyses were performed using GraphPad Prism 9. Statistical details for each experiment can be found in the corresponding figure legends and supporting files.

## Data and software availability
All mentioned ImageJ plugins have source code available and are licensed under open-source GNU GPL v3 license. The source data for the original Western blots are available within the paper.

# Acknowledgements
We thank Lynne Cassimeris (Lehigh University, USA), Pierre Gönczy and Didier Trono (EPFL, Switzerland), Dr. Duane Compton (Geisel School of Medicine at Dartmouth, USA) and Dr. Laurence Pelletier (Lunenfeld-Tanenbaum Research Institute, Canada) for the gift of materials and Ilya Grigoriev and Eugene Katrukha (Biology Imaging Center, Utrecht University) for the help with imaging and image analysis. This work was supported by China Scholarship Council scholarships to Fangrui Chen, Jingchao Wu and Chao Yang, the Netherlands Organization for Scientific Research Spinoza prize to AA, as well as the European Research Council Consolidator Grant 819,219 to LCK and the Eindhoven-Wageningen-Utrecht Alliance (https://www.ewuu.nl) that supports the Center for Living Technologies.

# Additional information

## Competing interests
Anna Akhmanova: Senior editor, *eLife*. The other authors declare that no competing interests exist.

## Funding

| Funder | Grant reference number | Author |
|---|---|---|
| China Scholarship Council | Scholraship | Fangrui Chen |
| China Scholarship Council | Scholarship | Jingchao Wu |
| China Scholarship Council | Scholaship | Chao Yang |
| Nederlandse Organisatie voor Wetenschappelijk Onderzoek | Spinoza prize | Anna Akhmanova |
| European Research Council | Consolidator Grant 819219 | Lukas C Kapitein |
| Eindhoven-Wageningen-Utrecht Alliance | Support for Center for Living Technologies | Lukas C Kapitein |

The funders had no role in study design, data collection and interpretation, or the decision to submit the work for publication.

## Author contributions
Fangrui Chen, Conceptualization, Formal analysis, Funding acquisition, Investigation, Methodology, Resources, Validation, Visualization, Writing - original draft, Writing - review and editing; Jingchao Wu, Investigation, Methodology, Resources; Malina K Iwanski, Investigation, Methodology, Visualization, Writing - original draft, Writing - review and editing; Daphne Jurriens, Investigation, Methodology; Arianna Sandron, Investigation, Resources; Milena Pasolli, Gianmarco Puma, Jannes Z Kromhout, Investigation; Chao Yang, Resources; Wilco Nijenhuis, Resources, Writing - original draft; Lukas C Kapitein, Supervision, Writing - original draft, Writing - review and editing; Florian Berger, Investigation, Visualization, Writing - original draft, Writing - review and editing; Anna Akhmanova,

Conceptualization, Funding acquisition, Project administration, Supervision, Writing - original draft, Writing - review and editing

## Author ORCIDs
Fangrui Chen http://orcid.org/0000-0003-2830-7760
Jingchao Wu http://orcid.org/0000-0002-2958-3751
Malina K Iwanski http://orcid.org/0000-0002-4903-9796
Daphne Jurriens http://orcid.org/0000-0001-5123-3099
Milena Pasolli http://orcid.org/0000-0001-6079-4808
Wilco Nijenhuis http://orcid.org/0000-0002-7095-0955
Lukas C Kapitein http://orcid.org/0000-0001-9418-6739
Florian Berger http://orcid.org/0000-0003-3355-4336
Anna Akhmanova http://orcid.org/0000-0002-9048-8614

## Decision letter and Author response
Decision letter https://doi.org/10.7554/eLife.77892.sa1
Author response https://doi.org/10.7554/eLife.77892.sa2

---

# Additional files

## Supplementary files

• Supplementary file 1. Cytosim configuration file for the simulations. The configuration file was executed with a compiled Cytosim (https://www.cytosim.org) version June 2019.

• Transparent reporting form

• Source data 1. Uncropped Western blots shown in this manuscript. (A,B) Western blots showing that NEDD1 and CEP192 are present in centrinone-treated AKAP450/CAMSAP2 knockout cells shown in *Figure 2—figure supplement 1A*. (C-J) Western blots showing the depletion of indicated proteins in centrinone-treated AKAP450/CAMSAP2 knockout cells shown in *Figure 2—figure supplement 1B*. White dashed lines indicate where the blots were cut before incubation. (K) Western blot showing that 3 hrs treatment with dynapyrazole A does not affect the expression of the endogenous dynein heavy chain and the dynactin large subunit p150Glued in centrinone-treated AKAP450/CAMSAP2 knockout cells shown in *Figure 4D*.

• Source data 2. Uncropped Western blots shown in this manuscript. (A) Western blots showing the knockout of p53 from AKAP450/CAMSAP2 knockout cell line shown in *Figure 2—figure supplement 2F*. (B) Western blots showing the knockout of pericentrin from AKAP450/CAMSAP2/ p53 knockout cell line shown in *Figure 2—figure supplement 2K*. (C-D) Western blots showing expression levels of CDK5RAP2, γ-tub, and ninein (NIN) in control (-CentB) and centrinone-treated AKAP450/CAMSAP2/p53 knockout and AKAP450/CAMSAP2/p53/PCNT knockout cell lines shown in *Figure 2—figure supplement 4B*. (E) Western blots showing the knockout of AKAP450 and CAMSAP2 from EB1/EB3 mutant RPE1 cell line shown in *Figure 2—figure supplement 5B*. (F) Western blots showing the knockout of pericentrin in AKAP450/CAMSAP2/CDK5RAP2/MMG/p53 / knockout cell line shown in *Figure 7—figure supplement 3G*. White dashed lines indicate where the blots were cut before incubation.

## Data availability

The configuration file for Cytosim, the software used for simulations, is included as Supplementary file 1. All numerical data and all raw Western blot data are included as Source Data files.

---

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
