## [Editor Report]

Microtubules are organized by microtubule organizing centers (MTOCs) such as the centrosome, which is composed of two centrioles surrounded by pericentriolar material (PCM). Despite a century of investigation, the mechanisms by which the centrosome organizes microtubules remains incompletely understood. Here, using genetic and pharmacological manipulations, as well as computer simulations, Chen and colleagues generate interphase cells with centriole-less PCM to investigate mechanisms by which PCM proteins cluster and nucleate and anchor microtubules. This manuscript will be of interest to cell biologists studying microtubule organization.

---

## [Decision Letter]

**Decision letter after peer review:**

[Editors’ note: the authors submitted for reconsideration following the decision after peer review. What follows is the decision letter after the first round of review.]

Thank you for submitting the paper "Centriole-independent centrosome assembly in interphase mammalian cells" for consideration by *eLife*. Your article has been reviewed by 3 peer reviewers, and the evaluation has been overseen by a Reviewing Editor and a Senior Editor. The following individual involved in review of your submission has agreed to reveal their identity: Laurence Pelletier (Reviewer #3).

We are sorry to say that, after consultation with the reviewers, we have decided that this work will not be considered further for publication by *eLife* in its current form.

While the reviewers agree that the data presented are abundant and of high quality, there were general concerns that this paper did not provide a significant advance either conceptually or mechanistically beyond prior work in either understanding the roles of the proteins presented or how interphase microtubule arrays are physiologically organized. However, the reviewers felt that this work could be reconsidered as a Research Advance if additional data is added to bolster the impact of the findings and the manuscript is rewritten to make the premise clearer, as described below.

*Reviewer #1 (Recommendations for the authors):*

Here, Chen et al. report the presence of acentriolar PCM in interphase cells following removal of the centrioles, the PCM and Golgi microtubule scaffolding protein AKAP450, and the microtubule minus end protein CAMSAP2. This PCM is able to form a single cylindrical centralized MTOC and microtubule array, the establishment and maintenance of which is dependent on pericentrin, g-tubulin, CDK5RAP2, and ninein. Dynein mediated microtubule-based transport establishes and maintains this singular array, bringing and keeping together small PCM foci. Intriguingly, the presence of CAMSAP2 works against the establishment of this single centralized microtubule array as microtubule minus ends are distributed throughout the cytoplasm. However, microtubule minus ends can be forced into a centralized localization by tethering CAMSAP to a rapalog-inducible minus end directed motor. Overall, this paper contains abundant data that are well presented and the experiments are well controlled and documented. I think this study is important as we can learn a lot about how the centrosome functions by divorcing its many parts and asking what they can do on their own. That said, this paper is a tour de force or what cells can do rather than what they actually do. The authors should consider putting these findings into the context of what they teach us about the normal processes they represent.

1. As mentioned above, I think some of the main findings of this paper are too buried and can be easily missed. Conceptually, it is exciting to have a platform to test how PCM can organize microtubules divorced from the normal complication of studying the centrosome. There are many helpful cartoons that help the reader interpret the figures and similar attention should be paid to explaining the rationale behind each experiment, again considering what they might teach us about the way in which PCM is able to normally function.

2. I find the vocabulary in the paper off putting. First, the word centrosome should mean something, i.e. centrioles surrounded by PCM, or at least as a centriole bearing entity. This word has been often misused in the literature, for example in the early mouse embryo where cells do not have centrioles, yet the MTOCs in these cells have been mislabeled as centrosomes. Centriole-less or acentriolar PCM would be a better term as has been recently used in other systems (Garbrecht et al., 2021; Magescas et al., 2021). Although it is true that PCM is not the perfect term either as pericentriolar should be in relation to a centriole, this term on its own has meaning as evidenced by its usage throughout the text. Second, the term acentrosomal MTOC (aMTOC) or noncentrosomal MTOC (ncMTOC) has often been used to describe MTOCs that are not derived from the centrosome such as in many types of differentiated cells. Thus, the use of this term to describe the PCM here is similarly problematic. I recommended referring to this structure as PCM for consistency.

3. A key interesting point in this paper is that CAMSAP antagonizes radial MTOC formation by pericentrin. I wish this concept had been further explored either experimentally or conceptually. The authors allude to its significance in the discussion, but instead focus on how PCM might be "redeployed" in differentiated cells, which does not seem to be the normal mode of microtubule organization seen in organismal studies. Instead, non-centrosomal proteins like CAMSAP and other proteins function in building these decentralized arrays concomitant with the centrosome attenuating its microtubule organizing capacity.

4. Figure 7 is a nice experiment as it indicates that, although not surprising, driving microtubule minus ends together creates a radial microtubule array, albeit as a ring-like structure rather than a single condensed point. Addition of pericentrin helps to focus the microtubule array into a condensed point. Under other conditions presented elsewhere in the paper, pericentrin is not able to counteract the presence of CAMSAP2 which drives microtubules into a decentralized array. Thus, this experiment suggests that pericentrin and the focusing of minus ends have additive effects. A more appropriate control for this experiment however is to show the results with and without the addition of Rapalog to control for any effects of the FKBP, FRB, or Rapalog itself.

5. The microtubule orientations shown in Figure 4G and Figure 4S1B are nice, however, it is important to show the associated motor localization from which the psuedocoloring was drawn.

6. I am confused by the timeline of siRNA treatments in Figure 5 and associated supplement relative to nocodazole treatment and washout. The text and cartoons depict that siRNA knockdown of targets was achieved just prior to nocodazole washout, which seems unlikely. Can you please clarify the timeline. In addition, the phenotypes siRNA depletions during steady state and after nocodazole washout are not the same. Is this true or just the images shown? Is this teaching us something interesting about establishment vs. maintenance of PCM structures?

*Reviewer #2 (Recommendations for the authors):*

Centrosomes are the major microtubule-organizing centers that support formation of a radial microtubule array in interphase and catalyse spindle formation in mitosis. Centrosomes are built on a centriolar core that accumulates a stable matrix called the pericentriolar material; microtubule-nucleating and anchoring activities are concentrated in the pericentriolar material. When centrioles are removed from interphase cells, weaker MTOCs form at the Golgi in an AKAP450-dependent manner. In this manuscript, Chen et al. characterize MTOCs in the absence of centrioles and AKAP450; they build on prior work they had published showing that, while microtubules are disorganized in the absence of centrioles and AKAP450, additional inactivation of the microtubule minus end-binding protein CAMSAP2 leads to the spatial concentration of centrosomal components like pericentrin and CDK5RAP2, which leads to the formation of a more organized microtubule network. They refer to these structures as "centriole-independent centrosomes" and provide evidence for dynein-mediated transport as being important for their formation. Some of the statements, including in the title and abstract, are not well justified. Overall, the work does not yet lead to a significant advance in understanding of how interphase microtubule arrays are organized.

There are a number of issues with this manuscript that make it difficult to provide a clear set of recommendations. The term "centriole-independent centrosomes" is incorrect. Centrosomes are defined structures, which contain a pair of centrioles that promoted localized assembly of a stable pericentriolar material matrix independently of microtubules. This stable structure, in turn, serves as the nucleation site for microtubules. This is clearly not the case for the accumulations described which form by dynein-driven forces bringing components together and are microtubule-dependent. More troubling is that the manuscript lacks a clear purpose. Why do the authors study a highly artificial system in which PLK4 is inhibited and AKAP450 and CAMPSAP2 are knocked out? What is the relevance of this state? The fact that they can tether dynein to CAMSAP2 and drive organization is unsurprising, given the significant prior work (starting from Verde et al. 1991) on dynein-based organization of microtubules into organized arrays. There are potentially interesting mechanistic questions like how CAMSAP2 competes with potential dynein recruiters in the pericentriolar material but no analysis of depth is conducted here. Overall, the effort comes across as not being sufficiently clearly motivated with a direct line of experimentation that has yielded new insights into microtubule organization in interphase cells.

*Reviewer #3 (Recommendations for the authors):*

In this manuscript, Chen et al. investigated the prospect of forming a microtubule organizing centre (MTOC) without centrioles, the catalysts of centrosomal MTOC formation. Upon disabling non-centrosomal MTOC pathways and depleting centrioles from cells, the authors describe the formation of a single centrally located acentrosomal MTOC (aMTOC). The authors go on to describe that the self-assembly of this structure is primarily driven by dynein driven aggregation of pericentriolar material (PCM). Particularly, the authors describe key PCM components that are necessary in sustaining the aMTOC that include: PCNT, γ-tubulin, CDK5RAP2 and ninein. The findings of this paper could yield further insight into the PCM redistribution and microtubule reorganization that occurs in differentiated cells such as myotubes and neurons, where centrosomal MTOCs are abandoned for more efficient non-centrosomal microtubule arrays.

The conclusions of this paper are well supported by the data. These include elaborate but well controlled experiments using complexly engineered cellular models, treatment schemes, and convincing imaging. However, some aspects of the manuscript could be clarified and extended to provide more robust support for their findings.

1. The authors clearly showed the process of PCM assembly in AKAP450/CAMSAP2-deleted acentriolar cells and their characteristics. However, it was not clear whether microtubules were nucleated from the assembled PCM in Figure 5B-E as microtubules could be nucleated in the cytoplasm. Can the authors perform microtubule regrowth assays using two different conditions (cold treatment and nocodazole) and stain them with antibodies against a-Tub and PCNT?

2. The authors suggest that an aMTOC can be formed in the absence of microtubule nucleation activity from the centrosome and Golgi apparatus. Can the authors show the same phenotypes in the presence of both Centrinone B and Brefeldin A (to disassemble the Golgi apparatus) in wild type hTERT-RPE1 and CAMSAP2 KO cell lines?

3. Removal of centrosomes typically triggers a p53-dependent G1 arrest however the authors acknowledge this and seem to have created RPE-1 models with AKAP450, CAMPSAP2, and p53 KO (Line 156-160, Figure 2E). While this paper is focused on aMTOC formation in interphase, I am curious as to whether the cells are still cycling after the cylindrical aMTOC formation. If so, what does progression through mitosis look like?

4. The authors describe an interesting phenomenon in this manuscript; however, the significance of this finding is not well emphasized in the abstract or in the ending paragraph of the introduction. In the last paragraph of the discussion the authors describe the implications of their findings in the re-organization of microtubules during cellular differentiation, however it would have been nice to delve deeper into how their findings could apply to these different models besides the identification of the redistribution of common players. Though they do concede that much is left to be discovered when it comes to these differentiated models.

[Editors’ note: further revisions were suggested prior to acceptance, as described below.]

Thank you for resubmitting your work entitled "Self-assembly of pericentriolar material in interphase cells lacking centrioles." for further consideration by *eLife*. Your revised article has been reviewed by 3 peer reviewers, and the evaluation has been overseen by a Reviewing Editor and Suzanne Pfeffer as the Senior Editor. The following individual involved in review of your submission has agreed to reveal their identity: Laurence Pelletier (Reviewer #1).

The manuscript has been improved but there are some remaining issues that need to be addressed, as outlined below:

This new manuscript has been significantly improved based on the reviewers' input from a previous submission. Before we can accept the manuscript, we would like the authors to address two remaining points that we believe will further improve the story.

Essential revisions:

1) Considering the highly manipulated model used by the authors to analyse PCM protein assembly and function, the introduction and discussion could be further improved to contextualise this work and explain its relevance in a more physiological setting. e.g during differentiation and in different cell types. See also comments by reviewer 2 and 3 for further details.

2) The impact of the simulations should be improved. As conducted/presented they merely confirm the already existing experimental data. However, the full potential of simulations, e.g. testing different outcomes by varying parameters and manipulating the system in different ways, has not been explored.

– What is the rationale behind the choice of the parameters defined by the authors (e.g. absolute and relative number of MTs, number of PCM complexes, number of MTs nucleated per PCM complex?)

– What is the range of these parameters that result in the outcome observed in cells?

– What happens if the orientation of CAMSAP-bound MTs would be biased (as may be the case during cell differentiation).

Exploring these types of questions will add significant additional value to the simulations that go beyond what is already demonstrated in cells.

*Reviewer #2 (Recommendations for the authors):*

Here Chen et al. submit a revised manuscript characterizing the self-assembly of PCM in cells that lack centrosomes, Golgi derived microtubules, and CAMSAP. As in the original version of the paper, the data are abundant, but well-presented and rigorously executed. While my concerns still remain about the relevance of these studies in this highly manipulated in vitro cell context to what cells do in vivo, the new data and framing do alleviate some of my original concerns. In particular, the authors underscored the importance of the fact that the structure that forms when CAMSAP microtubules are driven together with dynein can be refined by the presence of PCNT1. This point was bolstered in the text and through the addition of a computer simulation. While the ability of motors to generate a radial array of microtubules on their own has been previously demonstrated in many contexts over several decades, the demonstration that PCM can refine this interaction does extend this concept. The question does still remain whether this highly manipulated cell culture system with a nuanced relationship to in vivo contexts will appeal to the broad readership of *eLife*.

1) While I like the new simulations added as Figure 8 to experimentally bolster the point that CAMSAP microtubules steal PCM from the caMTOC, I do wish there was a way to directly test the sufficiency of PCM cohesion to counteract this process (i.e. experimentally reinforcing the point in Figure 8D) perhaps with phosphomimetics or other mutations that might create more stable interactions between PCM molecules. Such mutations might not be available and so could be beyond the scope of this work.

2) I still take issue with the way in which these findings are related to actual in vivo contexts, especially their relation to how MTOCs might form in differentiated cells. There are really three ways in which this work might relate to MTOCs in differentiated cells:

1) It might have a direct bearing on structures that are clearly derived from PCM in vivo: These include at the base of cilia in *C. elegans* sensory neurons or PCM packets (*C. elegans*, Magescas et al., 2019), flares (*Drosophila*, Megraw, 2002), or fragments (Rusan and Wadsworth, 2005) seen at mitotic exit, some of which persist in interphase.

2) This work might relate to non-centrosomal/acentrosomal MTOCs seen in interphase cells, but there is little evidence that any of these structures are directly derived from PCM rather than sharing the same components as the PCM in some cases. Here we get into a bit of semantics again, but I think PCM should mean material surrounding centrioles (or that used to surround centrioles as is the case in the cases cited above). If PCM proteins assemble in different locations in the cell, these structures would no longer be called PCM. I like the idea that the CAMSAP-associated (or other non-centrosomal) microtubules could deploy a tug of war with the PCM proteins, but this idea is currently just speculative and should be deemed as such. If the non-centrosomal microtubules had a way to be biased asymmetrically (as in differentiated cells), the clustering of these PCM proteins would create a way to make positive feedback to further reinforce a non-centrosomal network.

3) This work might also give a mechanism for the way in which PCM is stripped from the centrosome following mitosis, a common occurrence across cell types and organisms upon mitotic exit. In this case, non-centrosomal microtubules could strip PCM from the centrosome once the matrix was crippled following the inactivation of mitotic kinases.

Some of these ideas are explored in the Discussion, but the Introduction is still imprecise in discussion of the potential relationship of the work to actual in vivo contexts. I would encourage the authors to be more precise with the potential implications in the Introduction and explore some of these concepts further in the Discussion.

For example:

Line 57-58: "These properties…are relevant because in most differentiated cell types…PCM forms acentrosomal MTOCs.": Please see point 2 above, but there is little evidence that PCM (meaning structures that derive from the centrosome) rather than PCM proteins forms acentrosomal MTOCs.

*Reviewer #3 (Recommendations for the authors):*

In this revised manuscript Chen et al. investigate the mechanisms underlying the self-assembly of PCM proteins into tight clusters that are able to function as MTOCs in interphase RPE1 cells, under conditions where these cells lack their two main MTOCs at centrioles and at the Golgi. These conditions, achieved by centrinone treatment to eliminate centrioles, and AKAP450 KO to eliminate the Golgi-associated MTOC, allow to study the MTOC formation properties of various proteins without interference by centrosomal or Golgi-associated MTOC activity. In this revised version the authors complemented an already comprehensive set of data with modelling studies, computationally confirming the observations made in cell-based experiments. Also, the relevance of the observations, made in an artificial situation of complete absence of centrosomal and Golgi MTOCs, has been addressed in the text.

Overall the study contains an impressive amount of information and useful insight regarding the ability of PCM proteins to self-assemble into structures that provide microtubule nucleation and anchoring sites, to control the shape of the cellular microtubule network. Some of this information would be difficult to obtain in the presence of the dominant centrosome and Golgi-associated microtubule arrays. On the other hand, an important criticism refers to the fact that the experimental setup is based on a highly artificial situation, raising concerns about how relevant the observations are in a physiological setting. Although improved, the additional data and rewriting still does not fully address this point. Also, limitations in the computational simulations need to be addressed.

Specific points:

1) Pericentrin and CDK5RAP2, which both are important for caMTOC formation and function, are not very important for the interphase centrosomal MTOC. This has been shown by Gavilan et al., 2018 using multiple knockout approaches similar to those in the current manuscript. This suggests that despite the importance of pericentrin and CDK5RAP2 for caMTOCs, they are not very important at the major physiological MTOC that they localize to, at least during interphase. This should be discussed and the above study cited (it was cited but not in this context). Do the authors envision a specific scenario in which their findings would help understanding MTOC assembly?

2) The Gavilan et al. study above also described cytoplasmic clusters of PCM proteins that form in centriole-lacking AKAP450 KO cells and that nucleate and organize microtubules (referred to a cytoplasmic or 'cMTOCs'). They probe these with a panel of antibodies and reveal their composition including the absence of CEP192 (similar to the analysis of caMTOCs in the current manuscript). This should be discussed/cited.

3) I wonder why the authors refer to some of the caMTOCs as 'cylindrical' – what is the evidence for this? It would imply some kind of geometry, but to my eye they rather appear to be clusters arranged in a roughly linear fashion. In the absence of data supporting a cylindrical shape, I would suggest changing this description.

4) As presented, the computer simulation data does not add much to the manuscript. It mainly confirms the experimental data, so what is the point of it, if one already has cell-based experimental evidence? It would be useful to make predictions and then design experiments in cells to test these, but this has not been done. At the very least, it would be useful to use the modelling to define parameter ranges within which the observed effects are true. This would give the modelling more meaning, since this may not be feasible to do in cells. For example, the authors make several assumptions regarding specific parameters included in the simulations such as the numbers of microtubules, PCM complexes, and CAMSAP-kin14 complexes, but there is no indication of how they came up with these numbers – are they related in any way to estimates in cells? In some simulations this does not seem to be the case. For example, to model the effect of PCM cluster dispersion, the authors assume 300 randomly oriented, CAMSAP-associated MTs, and only 50 PCM clusters each with one associated MT, but in cells this ratio seems to be the opposite – only 25% of all MTs have minus-end-out orientation (not cluster-associated).

Also, what is the outcome of simulations, if any of these parameters were to be gradually increased or decreased? In the absence of such data, it seems as if the authors have picked numbers that produce the desired outcome observed in cells. Indeed, it may be informative to ask under what conditions this outcome is not observed.

Regarding the more general simulation input parameters in Table 1, the authors have included references only for some of them. In the text they state "The numerical values for the parameters of our simulations have been taken from literature or reasonably chosen otherwise". What does 'reasonably chosen otherwise' mean? I understand that there may not be a reference for every parameter, but in these cases there should be at least a brief explanation describing how it was chosen/estimated.

Line 605: as far as I know, NEDD1 is not a gTuRC activator.

---

## [Author Response]

[Editors’ note: the authors resubmitted a revised version of the paper for consideration. What follows is the authors’ response to the first round of review.]

While the reviewers agree that the data presented are abundant and of high quality, there were general concerns that this paper did not provide a significant advance either conceptually or mechanistically beyond prior work in either understanding the roles of the proteins presented or how interphase microtubule arrays are physiologically organized. However, the reviewers felt that this work could be reconsidered as a Research Advance if additional data is added to bolster the impact of the findings and the manuscript is rewritten to make the premise clearer, as described below.

We thank the reviewers for the thoughtful and critical comments. We have profoundly revised our paper by adding new data and thoroughly re-writing the manuscript, including the Title, Abstract, Introduction and Discussion, to make the premise and the novelty of our study clearer. In short, the goal of our work was to investigate how centriole-independent processes – self-association of pericentriolar material (PCM) and dynein-mediated PCM transport – contribute to the assembly of interphase microtubule-organising centers (MTOCs). We also aimed to determine which molecular players participate in this process and how PCM assembly is affected by the presence of noncentrosomal microtubules. To conceptualize our experimental findings, we have now added a completely new Results section, where we used Cytosim simulations to model the distribution and self-assembly of PCM in different conditions, and we showed that our major experimental findings can be recapitulated *in silico* using simple assumptions. Thus, our experimental results complemented by our mechanistic simulations provide important new insights into the properties and functions of interphase regulators of microtubule minus-end organization.

As explained in detail below and in the revised manuscript, an important conclusion from our work is that interphase PCM can self-assemble in the absence of centrioles in a manner that requires a specific subset of PCM proteins. The presence of immobile randomly distributed non-centrosomal microtubules is sufficient to disrupt PCM self-assembly, a conclusion that we have developed further using simulations. We also explored whether a compact MTOC can be formed simply by minus-enddirected transport of stabilized microtubule minus ends and found that this is not the case – both simulations and experiments indicate that in this situation, microtubule minus ends form a more loose ring-like structure rather than a single compact MTOC. However, a compact MTOC does form when self-clustering PCM components are also present in the system. We think that these non-trivial findings are relevant for understanding MTOC assembly in interphase cells and MTOC organization during cell differentiation, when the centrosome is inactivated and PCM components are used to build non-centrosomal microtubule systems.

Concerning potential resubmission as a Research Advance, one reviewer wrote: “The eLife paper I was thinking of was Wang et al., 2015, which revealed that a ninein and g-tubulin pathway function in parallel to a patronin pathway in organizing non-centrosomal microtubule arrays in worms." That detail should have been included in the review as that paper is not from your team; I think that a rewriting to make the premise clear in relation to prior work could make a revised submission either a RESEARCH ADVANCE related to that story or if you prefer, an independent submission, but taking to heart, the comments of the reviewers.

Our paper is indeed related to the very nice study by Wang et al., 2015, which we cited, but the relationship is much more distant with respect to the model system, content and conclusions than what is normally expected from an *eLife* Research Advance. Therefore, we think that an independent submission would be more appropriate in this case.

Reviewer #1 (Recommendations for the authors):Here, Chen et al. report the presence of acentriolar PCM in interphase cells following removal of the centrioles, the PCM and Golgi microtubule scaffolding protein AKAP450, and the microtubule minus end protein CAMSAP2. This PCM is able to form a single cylindrical centralized MTOC and microtubule array, the establishment and maintenance of which is dependent on pericentrin, g-tubulin, CDK5RAP2, and ninein. Dynein mediated microtubule-based transport establishes and maintains this singular array, bringing and keeping together small PCM foci. Intriguingly, the presence of CAMSAP2 works against the establishment of this single centralized microtubule array as microtubule minus ends are distributed throughout the cytoplasm. However, microtubule minus ends can be forced into a centralized localization by tethering CAMSAP to a rapalog-inducible minus end directed motor. Overall, this paper contains abundant data that are well presented and the experiments are well controlled and documented. I think this study is important as we can learn a lot about how the centrosome functions by divorcing its many parts and asking what they can do on their own. That said, this paper is a tour de force or what cells can do rather than what they actually do. The authors should consider putting these findings into the context of what they teach us about the normal processes they represent.

We fully agree that we should have explained better what our findings teach us about the normal processes of MTOC assembly and microtubule organization. In the revised version of the paper, we outlined more clearly that the purpose of our experiments was to understand how two centriole-independent PCM properties – the propensity to self-associate (and potentially even form condensates, as proposed by the Hyman and other labs for mitotic cells) and dynein-dependent transport can contribute to the formation of an interphase MTOC. We also studied how the presence of CAMSAP-stabilized non-centrosomal microtubules affects PCM self-assembly, because such an effect is relevant for understanding how PCM is organized in differentiated cells, where noncentrosomal pathways of microtubule minus-end stabilization play a major role. Finally, we investigated whether minus-end-directed transport of stable microtubule minus-ends is by itself sufficient to form a single compact MTOC. In the revised version of the paper, we used simulations (new Figure 8) to conceptualise our experimental findings and showed that they could be recapitulated using simple assumptions, highlighting basic principles for self-organization.

In brief, we found that PCM clustering and dynein-mediated transport, acting together, can indeed generate a single compact acentriolar MTOC, and that this process requires a subset of PCM components, including pericentrin, ninein and CDK5RAP2, but not CEP192 or NEDD1 (Figures 1 and 2). These PCM components have different functions: pericentrin is required for PCM clustering and dynein transport, CDK5RAP2 for efficient γ-TURC mediated microtubule nucleation, and ninein likely contributes to PCM clustering and as well as microtubule minus-end anchoring but has a weaker impact on nucleation (Figure 5). PCM clustering increases microtubule density (Figure 3), suggesting that for efficient microtubule nucleation and minus-end stabilization, the proximity of multiple PCM components is an advantage. Strikingly, a similar set of PCM components is also present in noncentrosomal, PCM-based MTOCs such as those found in the nuclear envelope and the Golgi membranes in muscle cells where they are targeted by AKAP450 (Vergarajauregui et al., *eLife* 2020).

Importantly, although we did observe pericentrin-driven PCM clustering in our system, we saw no evidence that interphase PCM is subject to phase separation, because PCM clusters show no evidence of rapid protein exchange with the cytoplasmic pool (Figure 1H,I). Moreover, PCM clusters are sensitive to microtubule organization, and their formation is disrupted by the presence of CAMSAPstabilized minus-ends (Figures 4 and 6A-C, Videos 4-6). This again helps to explain how interphase PCM can be organized in differentiating cells when centrosome function is attenuated and CAMSAPstabilized minus ends become more abundant. In cells with inactivated centrosomes, targeting of CAMSAP-stabilized minus ends to a certain structure may be sufficient to organise the PCM, which in turn can strengthen MTOC properties of this structure. This notion matches the data on noncentrosomal microtubule organization at the Golgi membranes and the apical surface of epithelial cells, where CAMSAPs and PCM components cooperate.

When CAMSAP-stabilized minus ends were brought together by a minus-end-directed motor, a single compact MTOC containing PCM could form (Figure 6E,F). This result might seem trivial, but interestingly and in line with our newly added simulations, minus-end-directed transport of microtubule minus ends in the absence of attractive interactions between these ends is insufficient to form a compact MTOC (Figures 7 and 8). The presence of pericentrin (a factor inducing PCM clustering) is needed for the formation of a compact MTOC, as shown both by our experiments and simulations (Figures 7 and 8). Compact MTOC formation in the absence of centrioles thus requires that all components are subject to minus-end-directed transport. This underscores an important function of centrioles in bringing together proteins, some of which do and some of which don’t associate with dynein in order to generate a highly focused radial microtubule system, such as the one observed in immune cells.

All these points are now outlined better in the revised manuscript.

1. As mentioned above, I think some of the main findings of this paper are too buried and can be easily missed. Conceptually, it is exciting to have a platform to test how PCM can organize microtubules divorced from the normal complication of studying the centrosome. There are many helpful cartoons that help the reader interpret the figures and similar attention should be paid to explaining the rationale behind each experiment, again considering what they might teach us about the way in which PCM is able to normally function.

We agree with this comment, and in the revised version of the paper, we have explained better the rationale of each experiment.

2. I find the vocabulary in the paper off putting. First, the word centrosome should mean something, i.e. centrioles surrounded by PCM, or at least as a centriole bearing entity. This word has been often misused in the literature, for example in the early mouse embryo where cells do not have centrioles, yet the MTOCs in these cells have been mislabeled as centrosomes. Centriole-less or acentriolar PCM would be a better term as has been recently used in other systems (Garbrecht et al., 2021; Magescas et al., 2021). Although it is true that PCM is not the perfect term either as pericentriolar should be in relation to a centriole, this term on its own has meaning as evidenced by its usage throughout the text. Second, the term acentrosomal MTOC (aMTOC) or noncentrosomal MTOC (ncMTOC) has often been used to describe MTOCs that are not derived from the centrosome such as in many types of differentiated cells. Thus, the use of this term to describe the PCM here is similarly problematic. I recommended referring to this structure as PCM for consistency.

We fully agree that proper terminology and following the conventions in the field is important, though different researchers often use (and feel comfortable with) different terms. As correctly pointed out by the reviewer, acentriolar microtubule-organising structures have been called centrosomes previously. For example, the reviewer suggests that we should adopt the nomenclature from the recently published paper by Garbrecht et al., 2021, but the Galbrecht paper is in fact entitled “An acentriolar centrosome at the *C. elegans* ciliary base”, and the terms “acentriolar centrosome” and “acentriolar PCM” are used equally frequently in this manuscript, while the term “centriole-less PCM” is not used at all. This latter term was indeed introduced by Magescas et al. 2021, but we note that the terminology is not even uniform in the two manuscripts describing exactly the same acentriolar structure in *C. elegans*, making the choice difficult. Moreover, in our manuscript, we also describe CAMSAP2-dependent microtubule organization, and CAMSAP2 is not a PCM component.

So how should we call the structure we study? Importantly, in the previous version of the manuscript, we did not call this structure “acentriolar centrosome”, though we did discuss the process we investigated as “centrosome assembly in acentriolar cells”. Taking into account the opinions of Reviewers #1 and #2, who think that the term “centrosome” must be reserved for a centriole-bearing entity, we changed the title of the paper to “Self-assembly of pericentriolar material in interphase cells lacking centrioles”. Further, we think that the term *acentriolar MTOC*, which was very consistently used throughout the paper, accurately described the structure we are studying, because it is located centrally, lacks centrioles and potently organizes microtubules. However, we fully agree that this term can be confused with the term *acentrosomal MTOC*, which has been used to denote a variety of microtubule-nucleating and anchoring structures, containing or lacking PCM and located either centrally or peripherally. In the revised version of the manuscript, we, therefore, switched to the term *compact acentriolar MTOC* (caMTOC), because it is distinct from the previously used terms aMTOC and ncMTOC. This term identifies the major features of the studied structure – compact organization, absence of centrioles, the ability to organize a focused microtubule array, and most importantly, it can be applied to both a PCM-based structure and the structure induced by the minus-end-directed transport of CAMSAP2-decorated minus ends, even though CAMSAP2 is not a PCM component.

3. A key interesting point in this paper is that CAMSAP antagonizes radial MTOC formation by pericentrin. I wish this concept had been further explored either experimentally or conceptually. The authors allude to its significance in the discussion, but instead focus on how PCM might be "redeployed" in differentiated cells, which does not seem to be the normal mode of microtubule organization seen in organismal studies. Instead, non-centrosomal proteins like CAMSAP and other proteins function in building these decentralized arrays concomitant with the centrosome attenuating its microtubule organizing capacity.

In the revised version of the paper, we have explored how CAMSAP antagonizes radial MTOC formation by pericentrin in much more detail by adding new imaging data (new Video 4) and simulations (new Figure 8). We show that CAMSAP-decorated microtubule minus ends in AKAP450knockout cells are distributed throughout the cytoplasm and display very limited mobility on the timescale of hours (new Video 4), and our simulations demonstrate that random distribution of microtubules with immobile stable minus ends is sufficient to perturb self-assembly of PCM into a single MTOC (new Figure 8). We also describe better the significance of our findings for understanding MTOC organization in differentiated cells, where the same PCM components as identified in our work (pericentrin, CDK5RAP2, γ-tubulin and ninein) are used to nucleate and anchor microtubules at locations other than the centrosome.

4. Figure 7 is a nice experiment as it indicates that, although not surprising, driving microtubule minus ends together creates a radial microtubule array, albeit as a ring-like structure rather than a single condensed point. Addition of pericentrin helps to focus the microtubule array into a condensed point. Under other conditions presented elsewhere in the paper, pericentrin is not able to counteract the presence of CAMSAP2 which drives microtubules into a decentralized array. Thus, this experiment suggests that pericentrin and the focusing of minus ends have additive effects. A more appropriate control for this experiment however is to show the results with and without the addition of Rapalog to control for any effects of the FKBP, FRB, or Rapalog itself.

We agree that this is indeed an important point, which we have now strengthened by simulations in Cytosim that show that minus-end-directed transport of microtubule minus ends that do not associate with each other generates a ring rather than a compact MTOC (new Figure 8). Our simulations further showed that the addition of self-associating PCM in the system induces MTOC compaction.

We note that controls with and without rapalog were already included in the manuscript (see the original Figure 6F top panel and Figure 7A top panel), and we have now added additional images showing single and double transfections with the used FKBP and FRB fusions in AKAP450/CAMSAP2/p53/PCNT KO cells before and after rapalog treatment (new Figure 7 —figure supplement 1).

5. The microtubule orientations shown in Figure 4G and Figure 4S1B are nice, however, it is important to show the associated motor localization from which the psuedocoloring was drawn.

These images are now included in the Figure 4 —figure supplement 1B.

6. I am confused by the timeline of siRNA treatments in Figure 5 and associated supplement relative to nocodazole treatment and washout. The text and cartoons depict that siRNA knockdown of targets was achieved just prior to nocodazole washout, which seems unlikely. Can you please clarify the timeline. In addition, the phenotypes siRNA depletions during steady state and after nocodazole washout are not the same. Is this true or just the images shown? Is this teaching us something interesting about establishment vs. maintenance of PCM structures?

The timeline is now indicated more clearly in Figure 5B. Please note that nocodazole treatments and washouts are much shorter than siRNA treatment. Furthermore, it is indeed entirely correct that the morphology of the acentriolar MTOC before and after nocodazole treatment is not the same, because the very compact cylindrical structure forms slowly. We have now investigated this point in more detail by using cold treatment (new Figure 4I and Figure 4 —figure supplement 1C) and found that the cylindrical structure is formed by association with microtubules: cold treatment revealed the presence of short cold-stable microtubules associated with the cylindrical MTOC and partial deformation of the PCM cluster upon complete microtubule disassembly (Figure 4I and Figure 4 —figure supplement 1C).

Reviewer #2 (Recommendations for the authors):Centrosomes are the major microtubule-organizing centers that support formation of a radial microtubule array in interphase and catalyse spindle formation in mitosis. Centrosomes are built on a centriolar core that accumulates a stable matrix called the pericentriolar material; microtubule-nucleating and anchoring activities are concentrated in the pericentriolar material. When centrioles are removed from interphase cells, weaker MTOCs form at the Golgi in an AKAP450-dependent manner. In this manuscript, Chen et al. characterize MTOCs in the absence of centrioles and AKAP450; they build on prior work they had published showing that, while microtubules are disorganized in the absence of centrioles and AKAP450, additional inactivation of the microtubule minus end-binding protein CAMSAP2 leads to the spatial concentration of centrosomal components like pericentrin and CDK5RAP2, which leads to the formation of a more organized microtubule network. They refer to these structures as "centriole-independent centrosomes" and provide evidence for dynein-mediated transport as being important for their formation. Some of the statements, including in the title and abstract, are not well justified. Overall, the work does not yet lead to a significant advance in understanding of how interphase microtubule arrays are organized.

We agree with the reviewer that the conceptual contribution of our findings was not explained sufficiently well. We have rectified this in the revised manuscript both by improving the writing, and by including a set of simulations to conceptualise different properties of PCM and microtubule organization that either lead to or preclude PCM self-assembly into an MTOC.

To put it shortly, our paper provides new insights into the centriole-independent properties of PCM and the effect of non-centrosomal microtubules on PCM self-assembly.

1. We showed that clustering of PCM components and their dynein-mediated transport are both necessary and sufficient to form a single compact MTOC, provided that no other MTOCs or stable microtubules are present in the cell. This indicates that centrioles are not essential for generating a single compact interphase MTOC, but make PCM assembly more robust, resistant to microtubule perturbations and reorganization and keep PCM together by preventing its motility on minus-end-out microtubules.

2. We identified PCM components which are collectively able to generate a centrioleindependent MTOC, characterised their functions and showed that PCM clustering increases microtubule density.

3. We showed that the presence of a stable randomly organized microtubule network stabilized by CAMSAP is sufficient to disrupt PCM self-organization. This means that interphase PCM self-association and transport are tuned in a way that prevents strong PCM condensation. This finding is relevant and timely in light of recent work showing that mitotic centrosomes may form by phase separation of PCM components and is also important for understanding PCM organization in differentiating cells, where centrosomes are inactivated.

4. We showed that transport of CAMSAP-stabilized microtubule minus ends with a minus-enddirected motor is by itself not sufficient to form a compact MTOC, but cooperation with selfclustering PCM can help to achieve compaction, providing clues on how CAMSAP and PCM work together to organize different non-centrosomal MTOCs.

There are a number of issues with this manuscript that make it difficult to provide a clear set of recommendations. The term "centriole-independent centrosomes" is incorrect. Centrosomes are defined structures, which contain a pair of centrioles that promoted localized assembly of a stable pericentriolar material matrix independently of microtubules. This stable structure, in turn, serves as the nucleation site for microtubules. This is clearly not the case for the accumulations described which form by dynein-driven forces bringing components together and are microtubule-dependent. More troubling is that the manuscript lacks a clear purpose. Why do the authors study a highly artificial system in which PLK4 is inhibited and AKAP450 and CAMPSAP2 are knocked out? What is the relevance of this state? The fact that they can tether dynein to CAMSAP2 and drive organization is unsurprising, given the significant prior work (starting from Verde et al. 1991) on dynein-based organization of microtubules into organized arrays. There are potentially interesting mechanistic questions like how CAMSAP2 competes with potential dynein recruiters in the pericentriolar material but no analysis of depth is conducted here. Overall, the effort comes across as not being sufficiently clearly motivated with a direct line of experimentation that has yielded new insights into microtubule organization in interphase cells.

Our paper was indeed entitled “Centriole-independent centrosome assembly in interphase mammalian cells”, putting the emphasis on the centriole-independent pathways of centrosome assembly studied in this manuscript, but the structure that we studied was indeed consistently termed throughout the whole paper an “acentriolar MTOC” and not a “centrioleindependent centrosome”. As discussed in the response to Reviewer #1, opinions differ on whether or not a compact acentriolar MTOC can be called a centrosome. For example, a recent Current Biology paper by Galbrecht et al. (2021) describing an acentriolar MTOC was entitled “An acentriolar centrosome at the *C. elegans* ciliary base”. Furthermore, even if centrosomes are defined as strictly centriole-dependent structures, different pathways of their assembly may be both centrioledependent and independent (e.g. through phase separation of PCM, as proposed for mitotic centrosomes by the Hyman lab, Woodruff et al., Cell 2017). However, as discussed above, we respect the opinions of Reviewers #1 and #2 and changed the title of the paper to “Self-assembly of pericentriolar material in interphase cells lacking centrioles” and also renamed acentriolar MTOC into *compact acentriolar MTOC* (caMTOC) to avoid confusion with the term “acentrosomal MTOC”.

Notably, this reviewer also suggests that the term MTOC may be inappropriate to describe the structure we study, because it falls apart during nocodazole treatment and is thus sensitive to the state of the microtubule array, and only a microtubule-independent structure should be called an MTOC. We respectfully disagree for the following reasons:

1. Some MTOCs are microtubule-sensitive, with the Golgi apparatus providing the best-studied example of a microtubule- and dynein-dependent structure which nucleates and anchors microtubules. The Golgi complex forms an MTOC which tethers ~50% of all microtubules in wild type RPE1 cells at steady state, when cells are not treated with any inhibitors (Efimov et al., Dev Cell 2007).

The importance of this MTOC is supported by a large body of work in different systems. Yet the Golgi rapidly falls apart if cells are treated with nocodazole at 37°C.

2. In the revised version of the paper, we show that the compact acentriolar MTOC that we study is not fully dependent on microtubules – if microtubules are disassembled at 4°C, the PCM cluster stays together, even at 37°C (new Figure 4I) and can nucleate microtubules similar to the centrosome, when nocodazole is removed (new Figure 5 —figure supplement 1A). As explained in the paper, the fact that caMTOC is driven apart when cells are treated with nocodazole at 37°C, is due to the fact that the cells pass through a transient stage where the more stable minus-end-out microtubules become relatively abundant and serve as rails for dynein-mediated PCM transport out of the cluster. These experiments teach us two important things: 1/ the interactions between a subset of PCM components can keep them together, so that they form a compact microtubule-nucleating and anchoring structures even in the absence of a centriole template or microtubules; 2/ these interactions are not sufficiently tight to prevent motor-dependent movement of PCM clusters, and this makes PCM sensitive to the position of stable minus ends. This property is likely important for understanding how PCM components are organized in differentiated cells lacking centrosomes.

To conclude, we think that the term compact acentriolar MTOC (caMTOC), is appropriate, because it is distinct from the previously used terms aMTOC or ncMTOC. This term identifies the major features of the studied structure – compact organization, absence of centrioles and the capacity organized a focused microtubule array, and it can be applied to both a PCM-based structure and the structure induced by the minus-end-directed transport of CAMSAP2-decorated minus ends, even though CAMSAP2 is not a PCM component.

Purpose of this study and novel conclusions: our goal was to explore the mechanisms driving selfassembly of PCM into MTOCs. Three fundamentally different processes play a role in centrosome formation: 1/specific binding of PCM to the centriole wall; 2/self-association of PCM components or even their phase separation, whereby centrioles may serve as catalysts of condensate formation; 3/dynein-mediated transport of PCM and microtubule minus ends. In principle, one could argue that each of these processes alone or their combinations could be sufficient to form an MTOC, and in this paper, we generate a system where we can study the contributions and molecular components of the second and the third process in detail, independently of the first one. To achieve this, we inhibit PLK4 to remove centrioles, and we also remove AKAP450 so that we can study MTOC formation independently of PCM binding to Golgi membranes, to simplify the analysis. In this system, we then remove additional factors to study their impact on PCM self-assembly. We show that the presence of CAMSAP2-stabilized microtubules is sufficient to disrupt PCM self-assembly, and in the revised version of the paper, we analyse this phenomenon in more detail. Our simulations show that an immobile randomly organized microtubule network is sufficient to perturb PCM clustering due to the dyneindriven cluster motility, and one does not need to invoke any assumptions on specific competition of CAMSAP2 with dynein recruiters to explain the observed phenotype. Furthermore, the results on minus-end-directed motor tethering to CAMSAP2-stabilized minus ends are less trivial than one could have thought – both experiments and simulations show that by itself, such tethering is insufficient for microtubule minus-end focusing, which requires cooperation with self-clustering microtubule nucleators. The synergy between CAMSAPs and PCM components is a recurring theme in organizing microtubule arrays in differentiated cells, and our findings help to understand its basis. All these aspects of our work are motivated and emphasized better in the revised manuscript.

Reviewer #3 (Recommendations for the authors):In this manuscript, Chen et al. investigated the prospect of forming a microtubule organizing centre (MTOC) without centrioles, the catalysts of centrosomal MTOC formation. Upon disabling non-centrosomal MTOC pathways and depleting centrioles from cells, the authors describe the formation of a single centrally located acentrosomal MTOC (aMTOC). The authors go on to describe that the self-assembly of this structure is primarily driven by dynein driven aggregation of pericentriolar material (PCM). Particularly, the authors describe key PCM components that are necessary in sustaining the aMTOC that include: PCNT, γ-tubulin, CDK5RAP2 and ninein. The findings of this paper could yield further insight into the PCM redistribution and microtubule reorganization that occurs in differentiated cells such as myotubes and neurons, where centrosomal MTOCs are abandoned for more efficient non-centrosomal microtubule arrays.The conclusions of this paper are well supported by the data. These include elaborate but well controlled experiments using complexly engineered cellular models, treatment schemes, and convincing imaging. However, some aspects of the manuscript could be clarified and extended to provide more robust support for their findings.1. The authors clearly showed the process of PCM assembly in AKAP450/CAMSAP2-deleted acentriolar cells and their characteristics. However, it was not clear whether microtubules were nucleated from the assembled PCM in Figure 5B-E as microtubules could be nucleated in the cytoplasm. Can the authors perform microtubule regrowth assays using two different conditions (cold treatment and nocodazole) and stain them with antibodies against a-Tub and PCNT?

This is an excellent suggestion which we have followed. Using cold treatment with and without nocodazole, we show that the PCM cluster indeed represents the major microtubulenucleating structure in the cell (new Figure 5 —figure supplement 1A). We note that at early stages of microtubule re-growth, EB1 is a better marker of nascent microtubules than α-tubulin, because EB1 stains all microtubules but does not associate with the pool of free tubulin, abundant in such cells.

2. The authors suggest that an aMTOC can be formed in the absence of microtubule nucleation activity from the centrosome and Golgi apparatus. Can the authors show the same phenotypes in the presence of both Centrinone B and Brefeldin A (to disassemble the Golgi apparatus) in wild type hTERT-RPE1 and CAMSAP2 KO cell lines?

We have generated the requested data (new Figure 1 —figure supplement 1C-F). Interestingly, the disruption of the Golgi apparatus with Brefeldin A combined with centrinone B treatment could indeed trigger formation of compact acentriolar MTOCs in CAMSAP2 KO but not in wild type cells. The efficiency of this process was much lower than in AKAP450/CAMSAP2 KO cells, possibly because the treatment was relatively short (2 hr, due to the toxicity of Brefeldin A), whereas compact acentriolar MTOCs form slowly. Furthermore, AKAP450 attached to the remnants of the Golgi matrix might still perturb PCM organization and compaction in Brefeldin A-treated cells.

3. Removal of centrosomes typically triggers a p53-dependent G1 arrest however the authors acknowledge this and seem to have created RPE-1 models with AKAP450, CAMPSAP2, and p53 KO (Line 156-160, Figure 2E). While this paper is focused on aMTOC formation in interphase, I am curious as to whether the cells are still cycling after the cylindrical aMTOC formation. If so, what does progression through mitosis look like?

To address the reviewer’s suggestion, we have added a new Supplemental figure (Figure 2 —figure supplement 3 of the revised paper), where we show that AKAP450, CAMPSAP2 and p53 KO cells indeed divide also in presence of centrinone B and form bipolar spindles with unfocused poles. The duration of mitosis is variable, which is not surprising given previous extensive work showing that the presence of centrosomes makes cell division more robust. The fact that these cells keep cycling in spite of prolongated mitosis is also not surprising as it fits with previous work showing that elimination of p53 prevents cell cycle arrest in such conditions. Furthermore, we note that our paper is focused on interphase cells, and a more detailed analysis of mitosis and its abnormalities in the knockout cell models we have generated goes beyond its scope.

4. The authors describe an interesting phenomenon in this manuscript; however, the significance of this finding is not well emphasized in the abstract or in the ending paragraph of the introduction. In the last paragraph of the discussion the authors describe the implications of their findings in the re-organization of microtubules during cellular differentiation, however it would have been nice to delve deeper into how their findings could apply to these different models besides the identification of the redistribution of common players. Though they do concede that much is left to be discovered when it comes to these differentiated models.

We fully agree that the significance of our findings should have been explained better, and we have done our best to rectify this in the revised manuscript, as is also explained in responses to the comments of Reviewers #1 and #2.

[Editors’ note: what follows is the authors’ response to the second round of review.]

Essential revisions:1) Considering the highly manipulated model used by the authors to analyse PCM protein assembly and function, the introduction and discussion could be further improved to contextualise this work and explain its relevance in a more physiological setting. e.g during differentiation and in different cell types. See also comments by reviewer 2 and 3 for further details.

We have edited the Introduction and Discussion to better contextualize our work and explain its physiological relevance. The changes are highlighted in blue.

We further note that conceptually, there is no fundamental difference between knocking out one, two, three or more proteins to explore cellular mechanisms (all these experiments can be regarded as artificial, and the impact of different knockouts depends on the relative importance of the studied proteins). Generation of multiple simultaneous gene knockouts is an important way to move forward in characterizing molecular mechanisms in the mammalian system, which, unlike flies or worms, is hallmarked by a high degree of functional redundancy.

2) The impact of the simulations should be improved. As conducted/presented they merely confirm the already existing experimental data. However, the full potential of simulations, e.g. testing different outcomes by varying parameters and manipulating the system in different ways, has not been explored.– What is the rationale behind the choice of the parameters defined by the authors (e.g. absolute and relative number of MTs, number of PCM complexes, number of MTs nucleated per PCM complex?)– What is the range of these parameters that result in the outcome observed in cells?

We have improved the impact of simulations by adding experimental data confirming two specific predictions of the modelling: that increasing PCM-self-interaction will allow PCM protein clustering even in the presence of CAMSAP2-stabilized microtubule minus ends, and that biases in the distribution of CAMSAP2-stabilized minus ends can lead to biases in the distribution of PCM complexes. These data are included in the new Figure 8 —figure supplement 2.

Furthermore, in the previous version of the manuscript, we have made choices based on our estimates of these parameters. In the revised version of the manuscript, we have systematically varied the number of components in the system, explored their effect on microtubule and PCM organisation and obtained parameter ranges which match those observed in cells. These new simulations are shown in Figure 8 —figure supplement 1 and discussed in the Results and Discussion. We also improved the Supplemental Table S1, where the other parameters used for simulations are listed and explained.

– What happens if the orientation of CAMSAP-bound MTs would be biased (as may be the case during cell differentiation).

We have included a new set of simulations (Figure 8 —figure supplement 1E), where we biased the distribution of stabilized and immotile microtubule minus ends to the cell margin and showed that this causes a similar bias in the distribution of PCM complexes. This mimics the situation found in epithelial cells, where CAMSAP/Patronin-stabilized microtubule minus ends are attached to the cell cortex (for example, due to binding to the cortical actin cytoskeleton through spectraplakin), and where γ-TuRC-containing protein complexes are also located at the cortex through mechanisms that are not yet clear. Furthermore, we added a new Supplemental Figure (Figure 8 —figure supplement 2B) where we show images of AKAP450 knockout cells expressing mCherry-CAMSAP2, where we observed that a partially biased distribution of CAMSAP2-stabilized minus-ends triggers a bias in the distribution of the PCM component pericentrin.

Exploring these types of questions will add significant additional value to the simulations that go beyond what is already demonstrated in cells.

We fully agree, and we hope that the reviewers find the new additions useful.

Reviewer #2 (Recommendations for the authors):Here Chen et al. submit a revised manuscript characterizing the self-assembly of PCM in cells that lack centrosomes, Golgi derived microtubules, and CAMSAP. As in the original version of the paper, the data are abundant, but well-presented and rigorously executed. While my concerns still remain about the relevance of these studies in this highly manipulated in vitro cell context to what cells do in vivo, the new data and framing do alleviate some of my original concerns. In particular, the authors underscored the importance of the fact that the structure that forms when CAMSAP microtubules are driven together with dynein can be refined by the presence of PCNT1. This point was bolstered in the text and through the addition of a computer simulation. While the ability of motors to generate a radial array of microtubules on their own has been previously demonstrated in many contexts over several decades, the demonstration that PCM can refine this interaction does extend this concept. The question does still remain whether this highly manipulated cell culture system with a nuanced relationship to in vivo contexts will appeal to the broad readership of eLife.1) While I like the new simulations added as Figure 8 to experimentally bolster the point that CAMSAP microtubules steal PCM from the caMTOC, I do wish there was a way to directly test the sufficiency of PCM cohesion to counteract this process (i.e. experimentally reinforcing the point in Figure 8D) perhaps with phosphomimetics or other mutations that might create more stable interactions between PCM molecules. Such mutations might not be available and so could be beyond the scope of this work.

To experimentally bolster the point that the dispersion of PCM proteins in CAMSAP2-expressing cells can be counteracted by their increased self-association, we have now attached inducible homodimerization FKBP domains to pericentrin. We showed that this fusion protein is dispersed in AKAP450 knockout cells treated with centrinone but forms a cluster upon the addition of the compound triggering FKBP homodimerization, while CAMSAP2-stabilized microtubule minus ends remain dispersed (new Figure 8 —figure supplement 2A). This outcome was predicted by our modelling.

2) I still take issue with the way in which these findings are related to actual in vivo contexts, especially their relation to how MTOCs might form in differentiated cells. There are really three ways in which this work might relate to MTOCs in differentiated cells:1) It might have a direct bearing on structures that are clearly derived from PCM in vivo: These include at the base of cilia in *C. elegans* sensory neurons or PCM packets (C. elegans, Magescas et al., 2019), flares (Drosophila, Megraw, 2002), or fragments (Rusan and Wadsworth, 2005) seen at mitotic exit, some of which persist in interphase.2) This work might relate to non-centrosomal/acentrosomal MTOCs seen in interphase cells, but there is little evidence that any of these structures are directly derived from PCM rather than sharing the same components as the PCM in some cases. Here we get into a bit of semantics again, but I think PCM should mean material surrounding centrioles (or that used to surround centrioles as is the case in the cases cited above). If PCM proteins assemble in different locations in the cell, these structures would no longer be called PCM. I like the idea that the CAMSAP-associated (or other non-centrosomal) microtubules could deploy a tug of war with the PCM proteins, but this idea is currently just speculative and should be deemed as such. If the non-centrosomal microtubules had a way to be biased asymmetrically (as in differentiated cells), the clustering of these PCM proteins would create a way to make positive feedback to further reinforce a non-centrosomal network.3) This work might also give a mechanism for the way in which PCM is stripped from the centrosome following mitosis, a common occurrence across cell types and organisms upon mitotic exit. In this case, non-centrosomal microtubules could strip PCM from the centrosome once the matrix was crippled following the inactivation of mitotic kinases.Some of these ideas are explored in the Discussion, but the Introduction is still imprecise in discussion of the potential relationship of the work to actual in vivo contexts. I would encourage the authors to be more precise with the potential implications in the Introduction and explore some of these concepts further in the Discussion.For example:Line 57-58: "These properties…are relevant because in most differentiated cell types…PCM forms acentrosomal MTOCs.": Please see point 2 above, but there is little evidence that PCM (meaning structures that derive from the centrosome) rather than PCM proteins forms acentrosomal MTOCs.

We understand the semantic point of the reviewer, though we note that there is no consensus in the literature on the terminology describing centrosomal and non-centrosomal MTOCs and their components. For example, in the previous round of review, Reviewer #1, who is an expert in the field, has encouraged us to term the acentriolar structure that we study “acentriolar PCM”. We made a different choice, for reasons outlined in our previous rebuttal, but this clearly illustrates that the perception of terminology in this field varies greatly even among the few experts directly involved in these studies. To address the point of this reviewer, we edited the paper in some places to indicate more clearly that we are investigating the behavior of “PCM components” or “PCM proteins” when discussing acentriolar structures. However, using the term “PCM” seems to be by far the most practical way of collectively describing the molecules studied in this paper, especially when they collectively form a compact MTOC. Importantly, in the revised version of the manuscript, we took care to avoid the use of the term “PCM” and instead talk about PCM components or proteins when discussing naturally occurring acentrosomal MTOCs (e.g. line 57).

Furthermore, we have included more explicitly in the Introduction (lines 58-63) and Discussion (lines 648-654 and 680-685) the points and the papers suggested by the reviewer to better relate our results to the physiological situation. We further note that we did show in the paper that PCM components follow CAMSAP organisation (when CAMSAP-decorated ends are fully dispersed, PCM proteins are dispersed, and when minus ends are clustered, PCM proteins accumulate at the same site). In the revised version of the paper, we extended this conclusion by showing that in simulations, peripheral enrichment of stable minus ends relocalizes PCM proteins to the cell margin (Figure 8 —figure supplement 1E). Furthermore, we included experimental data showing that biases in the distribution of CAMSAP-stabilized microtubule minus ends cause a similar bias in the distribution of pericentrin clusters (new Figure 8 —figure supplement 2B).

Reviewer #3 (Recommendations for the authors):In this revised manuscript Chen et al. investigate the mechanisms underlying the self-assembly of PCM proteins into tight clusters that are able to function as MTOCs in interphase RPE1 cells, under conditions where these cells lack their two main MTOCs at centrioles and at the Golgi. These conditions, achieved by centrinone treatment to eliminate centrioles, and AKAP450 KO to eliminate the Golgi-associated MTOC, allow to study the MTOC formation properties of various proteins without interference by centrosomal or Golgi-associated MTOC activity. In this revised version the authors complemented an already comprehensive set of data with modelling studies, computationally confirming the observations made in cell-based experiments. Also, the relevance of the observations, made in an artificial situation of complete absence of centrosomal and Golgi MTOCs, has been addressed in the text.Overall the study contains an impressive amount of information and useful insight regarding the ability of PCM proteins to self-assemble into structures that provide microtubule nucleation and anchoring sites, to control the shape of the cellular microtubule network. Some of this information would be difficult to obtain in the presence of the dominant centrosome and Golgi-associated microtubule arrays. On the other hand, an important criticism refers to the fact that the experimental setup is based on a highly artificial situation, raising concerns about how relevant the observations are in a physiological setting. Although improved, the additional data and rewriting still does not fully address this point. Also, limitations in the computational simulations need to be addressed.Specific points:1) Pericentrin and CDK5RAP2, which both are important for caMTOC formation and function, are not very important for the interphase centrosomal MTOC. This has been shown by Gavilan et al., 2018 using multiple knockout approaches similar to those in the current manuscript. This suggests that despite the importance of pericentrin and CDK5RAP2 for caMTOCs, they are not very important at the major physiological MTOC that they localize to, at least during interphase. This should be discussed and the above study cited (it was cited but not in this context). Do the authors envision a specific scenario in which their findings would help understanding MTOC assembly?

We have improved the discussion of this point and cited the Gavilan et al., 2018 study in this context (lines 624-626 of the revised paper). Our work adds to a body of data indicating that the centrosome, one type of a physiological MTOC in interphase cells, relies for its function on several redundant pathways, one which depends on pericentrin and CDK5RAP2 and another one on CEP192 and NEDD1. The second pathway is highly important in mitosis (and therefore hard to eliminate using constitutive knockouts) but does not seem to contribute much to centriole-independent microtubule organization in any interphase setting. Precise measurements of the relative importance of the two pathways at interphase centrosomes requires inducible knockout studies combined with quantification of steady-state centrosomal microtubules, centrosome-dependent microtubule nucleation and anchoring, preferably in the absence of the confounding contribution of the AKAP450 and CAMSAP-dependent pathways (which generate a major part of microtubule density). Such experiments would extend the work by Gavilan et al., who performed counting of EB1 comets emerging from the centrosome with/without nocodazole treatment and would require extensive analyses that go beyond the scope of the current study.

2) The Gavilan et al. study above also described cytoplasmic clusters of PCM proteins that form in centriole-lacking AKAP450 KO cells and that nucleate and organize microtubules (referred to a cytoplasmic or 'cMTOCs'). They probe these with a panel of antibodies and reveal their composition including the absence of CEP192 (similar to the analysis of caMTOCs in the current manuscript). This should be discussed/cited.

We now cite these findings in the Discussion (lines 613-615).

3) I wonder why the authors refer to some of the caMTOCs as 'cylindrical' – what is the evidence for this? It would imply some kind of geometry, but to my eye they rather appear to be clusters arranged in a roughly linear fashion. In the absence of data supporting a cylindrical shape, I would suggest changing this description.

This is an interesting semantic point. Initially, we also called caMTOCs “linear”, but then a biophysicist pointed out to us that since these structures exist in a three-dimensional cellular space, one should assume that they represent a cylinder rather than a line (i.e. a one- or two-dimensional structure).

4) As presented, the computer simulation data does not add much to the manuscript. It mainly confirms the experimental data, so what is the point of it, if one already has cell-based experimental evidence? It would be useful to make predictions and then design experiments in cells to test these, but this has not been done. At the very least, it would be useful to use the modelling to define parameter ranges within which the observed effects are true. This would give the modelling more meaning, since this may not be feasible to do in cells. For example, the authors make several assumptions regarding specific parameters included in the simulations such as the numbers of microtubules, PCM complexes, and CAMSAP-kin14 complexes, but there is no indication of how they came up with these numbers – are they related in any way to estimates in cells? In some simulations this does not seem to be the case. For example, to model the effect of PCM cluster dispersion, the authors assume 300 randomly oriented, CAMSAP-associated MTs, and only 50 PCM clusters each with one associated MT, but in cells this ratio seems to be the opposite – only 25% of all MTs have minus-end-out orientation (not cluster-associated).Also, what is the outcome of simulations, if any of these parameters were to be gradually increased or decreased? In the absence of such data, it seems as if the authors have picked numbers that produce the desired outcome observed in cells. Indeed, it may be informative to ask under what conditions this outcome is not observed.

The most important point of our simulations is to show that simple assumptions about the minus-end-directed motility and self-association of PCM components are sufficient to explain why a caMTOC forms in the absence, but not in presence of CAMSAP-stabilized microtubule minus ends. Another important conclusion is the demonstration that the same assumptions are sufficient to explain how PCM and CAMSAP-mediated minus end stabilization can cooperate during MTOC formation.

In the revised version of the paper, we have experimentally tested two specific predictions:

1. Our simulations predicted that if self-association of pericentrin would be increased, it would be able to form a single cluster in acentriolar AKAP450 knockout cells expressing CAMSAP2 (cells with dispersed stable minus ends). We have now tested this prediction by adding an inducible homodimerization domain to pericentrin and showed that in the presence, but not the absence of a homodimerizer compound, this modified pericentrin protein strongly clusters (new Figure 8 —figure supplement 2A), exactly as our simulations predict (Figure 8D).

2. We added a new set of simulations (new Figure 8 —figure supplement 1E), where we show that a bias in the distribution of stable microtubule minus ends leads to a similar bias in the distribution of PCM complexes. To test this prediction, we mildly overexpressed CAMSAP2 in acentriolar AKAP450 knockout cells. While endogenous CAMSAP2-stabilized minus ends in such cells are randomly distributed, CAMSAP2 overexpression causes partial minus end bundling in some cells, resulting in their enrichment in certain cell areas. We observed that also endogenous pericentrin clusters tend to be enriched in the same areas (Figure 8 —figure supplement 2B)

We agree that varying specific parameters of the simulations can be very informative because it allows exploring conditions that cannot be achieved in cells. We have therefore strongly extended the simulations and their description in Figure 8 —figure supplement 1 and the Results. We also added a paragraph to the Discussion.

From a systematic variation of the number of components in the simulations, we were able to draw very general conclusions about the self-organized structures that we observe. We see this behaviour in all of our additional experiments: if the number of components is large enough, the global organization emerges. For smaller numbers of components, the system becomes more noisy, components interact only sporadically and no global organization emerges. In conclusion, if the concentration or density of the components is not large enough, we are not able to recapitulate the experimental findings in our simulations. In general, all the interactions that we defined were reversible, and therefore the observed structures were dynamic. Therefore, the emergence of these dynamic structures not only depend on the specific interactions, but also on the dynamics and the crowdedness of the environment. Because of this non-trivial interplay of components, it is difficult to systematically relate all parameters to each other. Nonetheless with our additional systematic simulations, we identified conditions in which the experimentally observed structures can be recapitulated in our simulations. Specifically, we found that when two types of stabilized microtubule minus ends are present – minus ends attached to motile complexes that can self-associate and minus ends that cannot self-associate and can be either immobile or move on other microtubules, the ratio between different types of ends is important for the final outcome. This helps to understand how global patterns of microtubule organization can be controlled by relatively simple parameters, such as the abundance of CAMSAP proteins during cell differentiation or PCM self-clustering during mitotic onset.

Regarding the more general simulation input parameters in Table 1, the authors have included references only for some of them. In the text they state "The numerical values for the parameters of our simulations have been taken from literature or reasonably chosen otherwise". What does 'reasonably chosen otherwise' mean? I understand that there may not be a reference for every parameter, but in these cases there should be at least a brief explanation describing how it was chosen/estimated.

We have amended Supplemental Table S1 to better explain the choice of simulation parameters.

Line 605: as far as I know, NEDD1 is not a gTuRC activator.

We agree that this point remains unclear and amended the text to state that NEDD1 is a γ-TuRC binding protein (line 697 of the revised manuscript).